# Genetic algorithm-based personalized models of human cardiac action potential

**Dmitrii Smirnov**[1☯], **Andrey Pikunov**[1], **Roman Syunyaev**[1,2,3☯]*, **Ruslan Deviatiiarov**[4], **Oleg Gusev**[4], **Kedar Aras**[2], **Anna Gams**[2], **Aaron Koppel**[2], **Igor R. Efimov**[1,2]*

**1** Moscow Institute of Physics and Technology, Dolgoprudny, Russia, **2** The George Washington University, Washington, DC, United States of America, **3** Sechenov University, Moscow, Russia, **4** Kazan Federal University, Kazan, Russia

☯ These authors contributed equally to this work.
* siuniaev.ra@mipt.ru (RS); efimov@gwu.edu (IRE)

**Data Availability Statement:** All optical mapping, CAGE and RNA-seq data used in the study are available using the following DOI: https://doi.org/10.5061/dryad.stqjq2c0.

## Abstract

We present a novel modification of genetic algorithm (GA) which determines personalized parameters of cardiomyocyte electrophysiology model based on set of experimental human action potential (AP) recorded at different heart rates. In order to find the steady state solution, the optimized algorithm performs simultaneous search in the parametric and slow variables spaces. We demonstrate that several GA modifications are required for effective convergence. Firstly, we used Cauchy mutation along a random direction in the parametric space. Secondly, relatively large number of elite organisms (6–10% of the population passed on to new generation) was required for effective convergence. Test runs with synthetic AP as input data indicate that algorithm error is low for high amplitude ionic currents (1.6±1.6% for IKr, 3.2±3.5% for IK1, 3.9±3.5% for INa, 8.2±6.3% for ICaL). Experimental signal-to-noise ratio above 28 dB was required for high quality GA performance. GA was validated against optical mapping recordings of human ventricular AP and mRNA expression profile of donor hearts. In particular, GA output parameters were rescaled proportionally to mRNA levels ratio between patients. We have demonstrated that mRNA-based models predict the AP waveform dependence on heart rate with high precision. The latter also provides a novel technique of model personalization that makes it possible to map gene expression profile to cardiac function.

## Introduction

Over the past half century, mathematical models of cardiac electrophysiology came a long way in terms of complexity, precision, and the area of application. Recent advances in computational cardiac electrophysiology make clinical application of computer models possible due to personalization of tissue geometry and fibers orientation [1]. However, while tissue-specific, person-specific and pathology-specific gene expression profiles affect AP waveform and propagation, these differences are usually not accounted for. The tissue-level simulations are usually based upon the same averaged single-cell elements of the model.

**Funding:** The research was supported by Russian Foundation for Basic Research (https://www.rfbr.ru/rffi/eng) grants 18-07-01480 (to RS and DS), 19-29-04111 (to RS), 18-00-01524 (to RS) and Leducq Foundation (https://www.fondationleducq.org/) project RHYTHM (to IE and KA). RNA-based model development study was supported by Russian Scientific Foundation (https://rscf.ru/en/) grant 18-71-10058 (to AP). The funders had no role in study design, data collection and analysis, decision to publish, or preparation of the manuscript.

**Competing interests:** The authors have declared that no competing interests exist.

A number of publications utilized genetic algorithms (GA) to determine a set of cell model parameters reproducing experimental AP [2–5]. GA apply evolutionary principles to computational models aiming to find the optimal solution fitting experimental data. Initially, a number of model "*organisms*" with random parameter values are generated. After that, the "*selection*" operator is applied to the first *generation* of models, passing the models with higher values of *fitness function* to the "*mating pool*". Usually, a fitness function is based on the Euclidean distance, as a squared difference between model and experiment. *Mutation* and *crossover* operators are then applied to the models in the mating pool. The former modifies model parameters according to a probability distribution function. In the simplest GA setting, the crossover operator exchanges the parameter values between organisms with a fixed probability, however more complicated modifications of crossover operators, such as Simulated Binary Crossover (SBX) were shown to improve algorithm performance [6]. Modified models are then passed to the next generation, new fitness function values are calculated, and the same set of genetic operators is applied iteratively until desirable goodness of fit is reached.

An obvious advantage of GAs is that these algorithms are *perfectly parallel*, making its parallelization straightforward: the slowest part of the algorithm, fitness function, could be performed independently for different organisms, while communication between tasks is limited to a small array of parameters. However, effective implementation of GA requires modification of genetic operators for each particular set of problems, which is often referred to as "*no free lunch theorem*" [7]. One of the goals of the current study was to develop robust GA implementation making it possible to find the set of cardiac electrophysiology model parameters without premature convergence to sub-optimal solution.

Another limitation of optimization algorithms as applied to electrophysiological models is the absence of a unique solution. As was noted previously [8–10] same AP waveforms could be reproduced by computer models with different sets of parameters, in other words, model parameters are unidentifiable from the AP. Also, techniques combining stochastic pacing and complicated voltage-clamp protocols have been recently proposed to overcome the problem [4,8]. However, these techniques are limited to single-cell voltage-clamp recordings, which are not feasible in clinical electrophysiology or whole heart and cardiac tissue measurements. The aim of the current study was to develop a technique that would allow finding a unique solution using optical, microelectrode, or monophasic AP recordings from cardiac tissue or whole heart. To the best of our knowledge, this is a first study providing a technique suitable for optical recordings, where only normalized AP waveform is known, but not the exact transmembrane potential values. Arbitrary rescaling and shift of input AP waveform introduces a new dimension to parameters identifiability problem mentioned above. A possible approach to address this problem is to utilize so-called restitution property, which is AP dependence on heart rate or pacing cycle length (PCL). For example, a reduction of the sodium current would result not only in the reduction of the amplitude of AP, but also in change of the steady state intracellular ionic concentrations and consequently, in changes of the restitution curve. Several previous publications [2, 5] utilized restitution property for optimization of GA-based cardiac models. For example, Syed et. al. [2] have used atrial AP waveform at several PCLs as input for GA and paced every organism for 10 seconds before fitness function evaluation. As we demonstrate below in the Final Algorithm subsection of the Results section, this approach results in a poor convergence. We have identified two reasons behind this fact: firstly, intracellular concentrations require much more than 10 seconds to converge to a steady state; secondly, a model with the particular set of parameters may converge to different steady states depending on the initial conditions. In order to address these issues, we implemented a modification of GA allowing optimizing parameters and searching for steady state in the slow variables space

simultaneously. After each short simulation, variables are saved, modified by *genetic operators* and reused as initial states for a new generation.

Finally, we verified the algorithm against the experimental optical AP recordings from the human heart. Since we could not measure ionic channel conductivities directly, we used the following assumption instead (Fig 1B): these conductivities should be proportional to corresponding proteins mRNA level of expression as measured by either Cap Analysis of Gene Expression (CAGE) [11] or RNA-seq. Thus, given that GA output model parameters represent actual ionic channels conductivities, the model rescaled in correspondence with mRNA expression profile differences between two patients, would reproduce AP restitution properties of both patients. Moreover, we have to note that this approach (*i.e.* combining GA with transcription profile) could be regarded as another technique of model personalization. As we show below, GA signal-to-noise ratio (SNR) requirements are rather strict and hard to accomplish in a clinical setting, while mRNA expression profile is possible to measure from tissue biopsy.

## Materials and methods

### Computer simulations

We used O'Hara-Rudy model [12] to simulate human ventricular cell electrophysiology. The genetic algorithm (GA) is based on Bot et al. [3] (Fig 2A). Briefly: tournament selection was used for selection operator, *i.e.* two individuals are selected at random from two copies of the previous generation and the one with higher fitness function goes into the mating pool. Then random organisms are selected from the mating pool to modify their parameter by *crossover* (with a 0.9 probability) and *mutation* (with a 0.1 probability in the original algorithm, 0.9 probability in the modified algorithm) operators. Simulated binary crossover (SBX) [6] was used with polynomial probability density function (PDF) of 10th order and 0.5 genewise swap probability. Polynomial mutation with order 20 PDF was used in the original algorithm (see below the modifications to mutation operator, that we used in this study). After that, worst organisms in the mating pool are replaced by *elite* organisms (i.e. best organisms from the previous generation) organisms and the same sequence is repeated for the next generation.

We introduced modifications to the algorithm (green-tinted boxes on Fig 1A) as described below, while their beneficial effects on the algorithm are discussed in the "Results" section.

1. **Input data.** is steady state AP waveforms recorded at several PCL. The algorithm is optimized for optical AP recordings, where absolute transmembrane potential (TP) values are not known, and thus input data (both synthetic and experimental) is renormalized by the algorithm prior to every fitness function calculation. The following technique was used for renormalization. Firstly, input AP waveform is shifted along the time axis in order to superimpose compared waveforms; in particular, half-maximum depolarization of the waveforms to be compared is used as a reference point. After that, input AP is rescaled: $V_{rescaled} = \alpha V + \beta$, where $\alpha$ and $\beta$ coefficients are determined by the least-squares technique to minimize the deviation between input and output AP. In order to discard subthreshold depolarizations some large error value was assigned to low-amplitude APs (see below "Fitness function" subsection).

2. **AP calculations.** O'Hara-Rudy [12] model was simulated with a custom C++ code using the Rush-Larsen integration technique [13] with an adaptive step as described in [12]. The minimal time step was set to 5e-3 ms.
   Since cell-to-cell interactions affect AP waveform (S1A Fig), we simulated 1D tissue instead of a single cell when GA was applied to experimental data recorded from tissue. On the

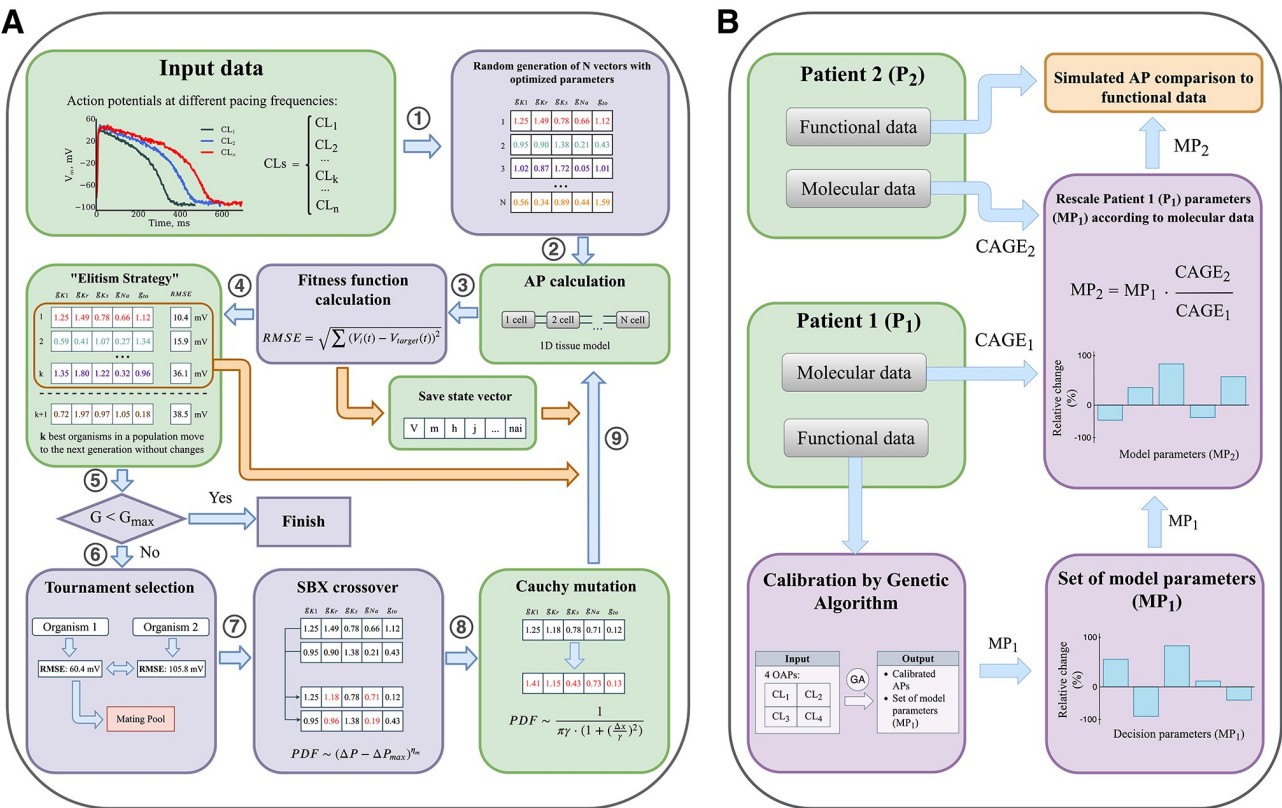

**Fig 1. Genetic Algorithm (GA) and CAGE-based personalization block diagrams.** (**A**) Genetic algorithm schematic diagram. Initially a set of organisms is generated, each of which is determined by a random vector of scaling factors for optimized model parameters (step 1). For each organism AP waveforms are calculated at several pacing frequencies (cycle lengths) and compared with the input APs (steps 3, 4). Organisms with the lowest Root Mean Square Error (RMSE) value are saved and directly copied into the subsequent generation, replacing the worst organisms (orange arrow). State vectors (intracellular concentration, gating variables etc) are also saved after each short simulation and reused as initial state during simulation in the next generation. After selection (step 6) the fittest organisms form the mating pool are modified by SBX crossover and Cauchy mutation (step 7, 8). Modified organisms move to the next generation (9). Process of AP and fitness function calculation, selection, crossover and mutation and elite replacement is repeated until the stop criterion is fulfilled. The algorithm is based on [3], modifications of the original algorithm are highlighted by green. (**B**) Algorithm verification with molecular (mRNA expression) and functional data (optical mapping). Patient 1 model parameters (MP₁) were determined by GA. Patient 1 parameters (MP₁) were rescaled proportional to Patient 2/Patient 1 mRNA expression level ratio (CAGE₂/ CAGE₁) and verified against functional data.

other hand, our simulations have shown that within the physiological range of conduction velocities (CV), 20–100 cm/s, exact gap junction conductivity value does not affect AP (S1B Fig). Therefore, during a GA run each model was simulated as 1D-tissue with 5 mS/μF conductivity between cells resulting in CV of 27 cm/s. The AP was recorded from the central cell of 30-cells long 1D tissue. We have found 30-cells long tissue to be sufficient to exclude boundary effects on the central cell of the tissue in case of CV slower than 27 cm/s (S1A Fig). 1D model simulations, while being less computationally expensive than 2D or 3D models, correspond to a plane wave propagating in a 3D tissue at a significant distance from the pacing electrode. Moreover, given that in a wide range of conductivities (S1B Fig) exact AP waveform was essentially independent from gap junctions conductivity, we can conclude that in case of a minor 2D or 3D wavefront curvature, additional perturbations by diffusion operator would not affect AP waveform as well.

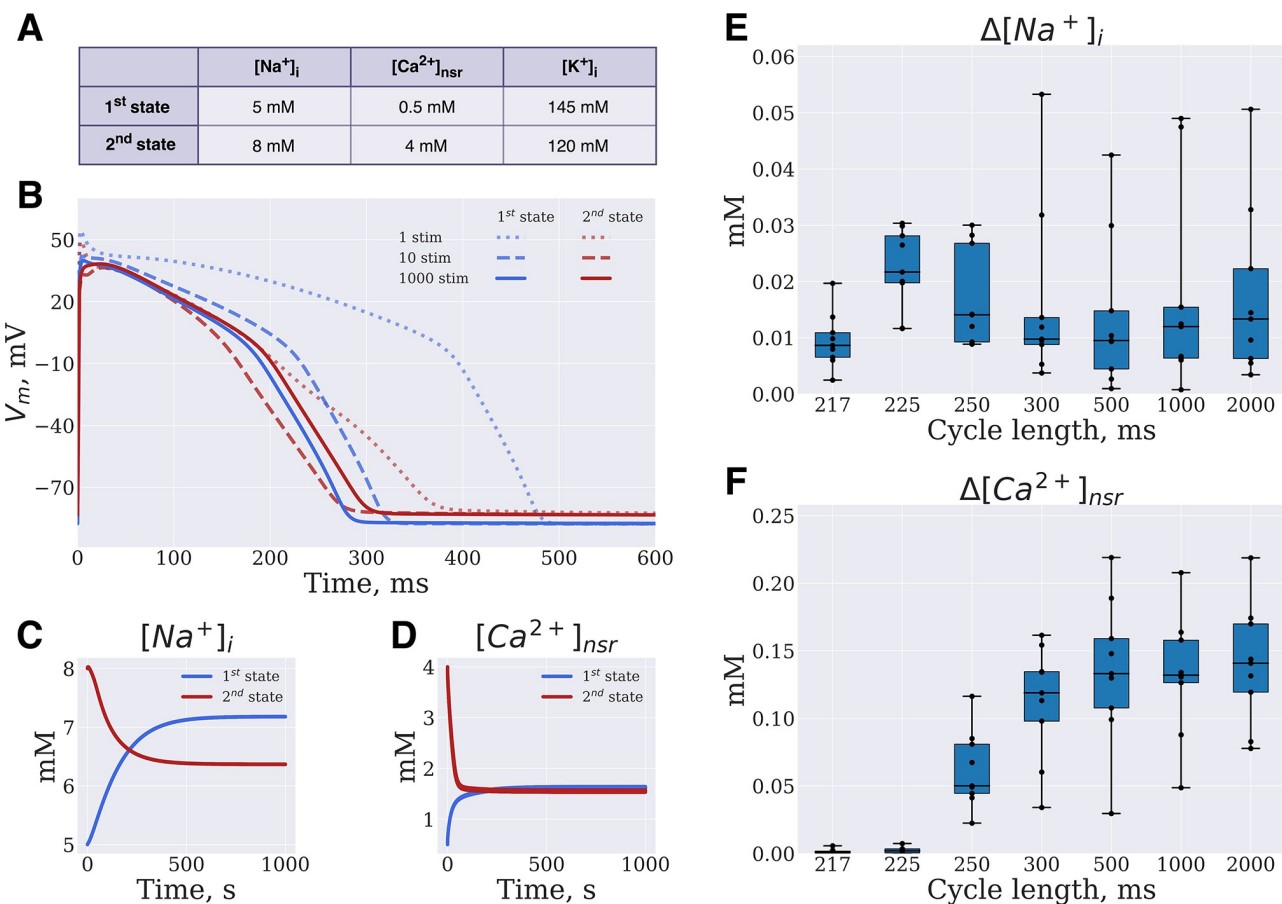

**Fig 2. Model convergence to steady state.** (**A,B**) The single-cell O'Hara-Rudy model was paced at 1Hz frequency for 1000 s, from different initial intracellular concentrations (initial concentrations are listed on panel A). AP waveform on the 1st, 10th and 1000th beats are depicted by dotted, dashed and solid lines correspondingly. (**C**, **D**) $[Na^+]_i$ and $[Ca^{2+}]_{nsr}$ concentrations change during the simulations. (**E**, **F**) Box-and-whiskers plots depict GA output $[Na^+]_i$ and $[Ca^{2+}]_{nsr}$ concentrations distance from steady state values.

3. **Fitness function.** We used a Root-Mean-Square Error (RMSE) to evaluate how close is a given organism AP to input data at particular PCL:

$$RMSE^i = \sqrt{\frac{1}{N} \sum_{t=0}^{N} \left[ V_{ref}^i(t) - V_{mod}^i(t) \right]^2},$$

where $V_{ref}^i$ is a baseline TP, $V_{mod}^i$ - is a TP of simulated AP, $N$—is a number of samples. The fitness function was calculated as a weighted sum of $SD^i$ corresponding to different PCLs:

$$RMSE_{tot} = \sum_{i=0}^{n} w_i \cdot RMSE^i$$

where $w_i$ is a weight coefficient. Weights were taken equal for all PCLs unless otherwise noted. In order to eliminate subthreshold depolarizations, APs with an amplitude below 30 mV were discarded, *i.e.* large RMSE value was assigned to these organisms. Since photon scattering in the optical mapping setting is known to distort depolarization [14,15], initial

depolarization phase (below -20 mV) was removed from compared AP prior to RMSE calculation.

4. **Save state vector.** Computational cost of pacing every organism at every generation until reaching steady state during a GA run is prohibitively high. Thus, we saved each AP after a short simulation. We used 9 stimulations before fitness function evaluation since we have found odd number of stimulations helpful to exclude possible 1:1 alternans in the output model: in the case of alternating APs, the waveform (and, consequently, RMSE as well) was different every other generation. Thus, if alternating AP have a low RMSE value on generation N, the RMSE is going to increase on generation N+1. After that state vector (ionic concentrations, gating variables, *etc.*) are saved for each organism. These state vectors are used as initial state in the next GA generation. As a result, each organism approaches closer to steady state variables with every generation.

5. **"Elitism strategy".** Since genetic operators tend to spoil a "good" solution, the best organism is passed to the next generation without any changes replacing the worst [16]. More precisely: elite organisms do participate in *mutation* and *crossover* as usual, but an "unspoiled" copy is saved to replace the worst organisms in every generation. However, final state of elite organisms on generation N are still reused as initial state on generation N+1, thus slow variables get closer to steady state, while AP waveform and RMSE changes correspondingly (see "save state variables" above). We have found that using a high number of elite organisms (about 6–10% of the whole population) is optimal for our GA modification.

6. **Cauchy mutation.**

   a. As was noted previously [17] Cauchy mutation in general results in better convergence for functions with many local minima. Therefore, we modified the mutation operator to use Cauchy distribution:

   $$f_X(x) = \frac{c}{\pi}\left[\frac{\gamma}{(x - x_0)^2 + \gamma^2}\right]$$

   where $f_X$ is a probability density of the distribution; $x_0$ corresponds to unmutated parameter value; $\gamma = 0.18^*x_{O'Hara-Rudy}$ is the half-width at half-maximum of the distribution (*i. e.* PDF is half the maximum value, when $x - x_0 = \gamma$); $\frac{c}{\pi}$ is a normalization constant resulting from a limited range of parameter values, varied between $0.01 \cdot x_{O'Hara-Rudy}$ and $4.0 \cdot x_{O'Hara-Rudy}$; $x_{O'Hara-Rudy}$ is original O'Hara-Rudy model parameter value. The particular value of $\gamma$ was chosen since test runs indicated best algorithm convergence in this case (see S3A and S3B Fig).

   b. Most commonly in genetic algorithms mutation operator is applied to each parameter separately with some fixed probability [18], we have found that it has adverse effects on algorithm convergence, because of the small probability to mutate several parameters simultaneously. Instead, we choose a random unit vector in multi-dimensional parameter-space and mutate the parameter vector in this direction. For example, in the case of the two-parameter problem: if $(\frac{\sqrt{2}}{2},\frac{\sqrt{2}}{2})$ unit vector was chosen, then both parameters are going to be increased by the same amount after mutation. See also Fig 3 and the corresponding Results section.

   c. The initial values of the slow variables at each PCL (intracellular $Na^+$ and network sarcoplasmic reticulum $Ca^{2+}$ concentrations) are included in parameters vector and mutated as usual model parameters. The $[Ca^{2+}]_{NSR}$ concentration was chosen, because significant

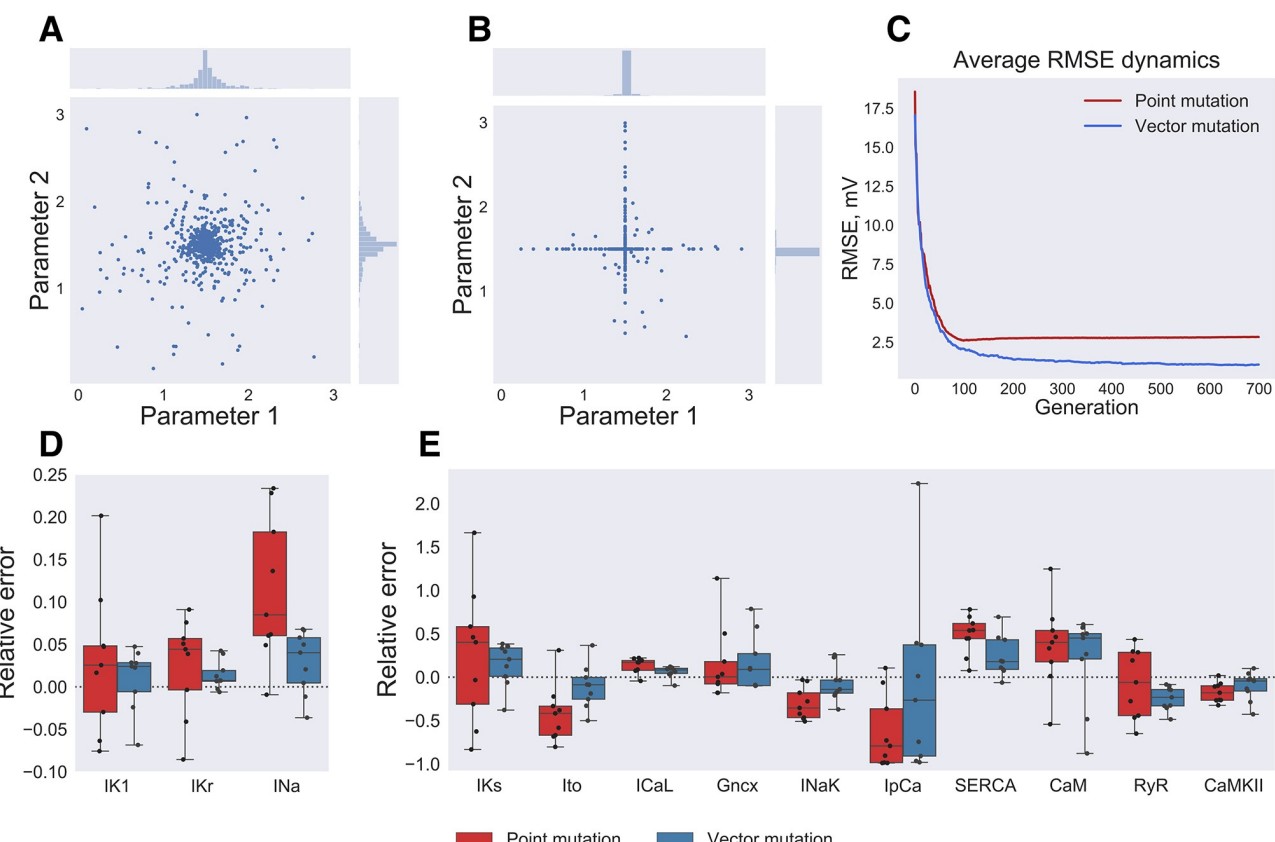

**Fig 3. Random mutation direction in multi-dimensional parameter space.** (**A**) Vector mutation: random mutation direction choice, blue points indicate parameter values after mutation. (**B**) Point mutation: each parameter value is mutated with fixed probability, which results in low probability to mutate parameter value in diagonal direction. (**C**) RMSE averaged over all organisms of 9 GA runs plotted against generation number. Red and blue lines correspond to point and vector mutations, respectively. (**D, E**) Objective parameters distribution for 9 GA runs with point mutation (red boxes) and vector mutation (blue boxes). Dashed line corresponds to the input model parameter values.

amount of calcium is stored within SR when the cell is at resting potential. We did not include potassium concentrations in the optimized parameters vector, since 10 mM $[K^+]_i$ concentration changes results only in approximately 2% Nernst potential change, and thus a major concentration changes have a minor immediate effect on AP waveform. This technique allows the algorithm to reach the model steady state more effectively (see corresponding part in the Results section). Note that intracellular concentrations are different for different PCL, consequently, there are separate values for each pacing frequency. Thus, when input data included 4 pacing frequencies the full set of parameters was: gNa, gKr, gK1, gKs, PCaL, gto, gNaK, gNCX, gpCa, Jrel, Jup, CMDN, CaMKII, $[Na^+_i]_{300\ ms}$, $[Na^+_i]_{500\ ms}$, $[Na^+_i]_{1000\ ms}$, $[Na^+_i]_{2000\ ms}$, $[Ca^{2+}_{NSR}]_{300\ ms}$, $[Ca^{2+}_{NSR}]_{500\ ms}$, $[Ca^{2+}_{NSR}]_{1000\ ms}$, $[Ca^{2+}_{NSR}]_{2000\ ms}$.

In order to verify the algorithm precision against synthetic data, simulated APs with an arbitrary set of model parameters were used as input data. The input model was paced until reaching steady state (1000 seconds) at several PCLs: 217 ms, 225 ms, 250 ms, 300 ms, 500 ms, 1000 ms, 2000 ms, unless noted otherwise.

## Multidimensional data visualization

1. **Principal Component Analysis** technique was used to visualize convergence in multidimensional parametric space (note, that slow variables where not used for PCA analysis). Organisms parameters of $p$ compared generations form a matrix $X$ of size $m \times np$, where $m$ is the number of organisms and n is the number of optimized parameters. According to the principal component method, matrix $X$ decomposed into a multiplication of two matrices $T$ (scores matrix) and $P$ (loading matrix) plus residual matrix $E$:

$$X = TP^T + E = t_1 p_1^T + t_2 p_2^T + E,$$

where $p_i$, $(i = 1, 2)$ correspond to loading matrix rows, columns $t_i$, $(i = 1, 2)$ of matrix $T$ are Principal Components (PC) being the organisms parameters in the new coordinate system. $P$ is a transformation matrix from initial variables space to 2D space of principal components. The first two principal components explained 75% of parameters variance in the Fig 4 and S2 Fig.

   In order to estimate organism parameters convergence in the principal components space we used **Mean Cluster Error** (MCE) and **Standard Distance** (SDist) metrics (S2A Fig):

   a. Mean Cluster Error represent the distance between $(x_0, y_0)$ point corresponding to precise solution (input model value) and $(x_c, y_c)$ corresponding to cluster geometric center calculated within a single generation:

   $$MCE = \sqrt{(x_0 - x_c)^2 + (y_0 - y_c)^2}$$

   b. Standard Distance was used to estimate cluster size:

   $$SDist = \sqrt{\frac{\sum_{i=1}^{n} (x_i - x_c)^2}{n} + \frac{\sum_{i=1}^{n} (y_i - y_c)^2}{n}},$$

   Where $x_i$ and $y_i$ are principal components of a given organism, $(x_c, y_c)$ is the cluster geometric center and $n$ is the total number of organisms within a generation.

## Donor heart procurement

All studies using human heart tissue were approved by the Institutional Review Board (Office of Human Research) of the George Washington University. In total, for this study, we procured from Washington Regional Transplant Community in Washington, DC discarded ventricular tissues from 14 deidentified donor human hearts, which were unsuitable for transplantation. All hearts were arrested using the ice-cold cardioplegic solution in the operating room and tissue was transported to the laboratory for dissection and electrophysiological experiments.

For subsequent CAGE mRNA analysis left ventricular tissue samples were dissected, submerged in RNAlater (Invitrogen) for 24 hours at 4°C, and stored at -80° C until the extraction of RNA. Tissue for RNA-Seq analysis was collected from the right ventricular (RV) tissue close to the apical region. Total RNA was extracted from these samples using the RNeasy Fibrous Tissue Mini Kit (Qiagen) according to the manufacturer's instructions, from approximately 30

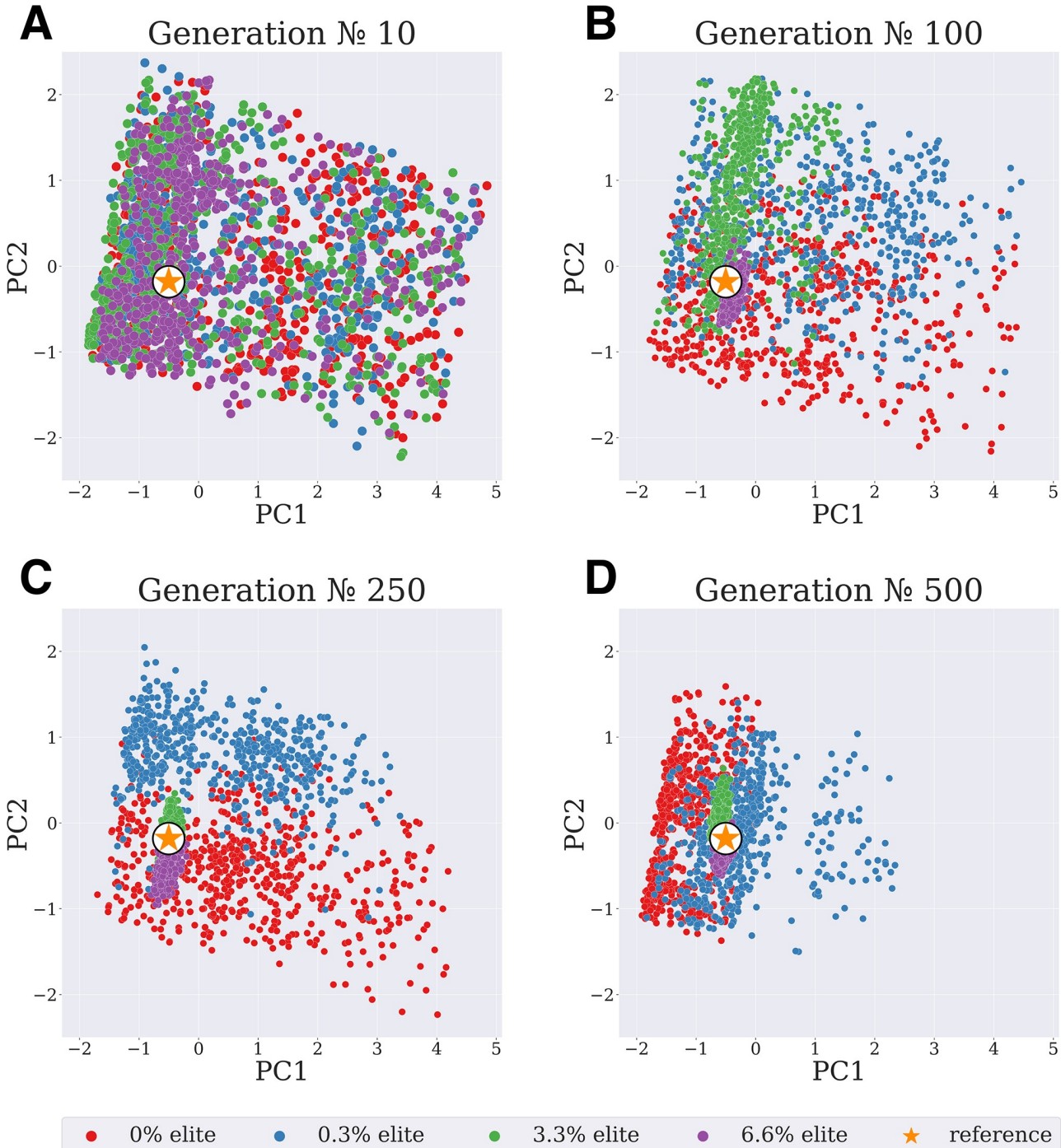

**Fig 4. Modified elitism strategy.** Principal component analysis of convergence dependence on the number of elite organisms: 0% (red points), 0.3% (blue points), 3.3% (green points) and 6.6% (purple points). Higher number of elite organisms results in the faster clusterization around the input model parameters.

mg samples. At the end of the extraction, spin columns were eluted with the eluate to increase the RNA yield. Total RNA concentration and purity were determined on an Eppendorf Bio-Photometer D30. Acceptable purity, as quantified by both the 260/280 and 260/230 absorbance ratios, was between 1.8 and 2.2. CAGE samples were also evaluated for integrity by electrophoresis on 1.2% agarose gels with 0.25 µg/ml ethidium bromide. RNA-seq samples were evaluated for integrity by Agilent Bioanalyzer, samples with RIN >7 were used for sequencing.

## Optical recordings of action potentials

Human left ventricular wedge or right ventricular outflow tract (RVOT) preparations were used for experiments, as described previously [19]. Briefly, wedges from the posterolateral LV free wall perfused via the left marginal artery were dissected, cannulated, and mounted in a tissue chamber with 4 surfaces (epicardium, endocardium, and the 2 transmural sides) facing 4 CMOS cameras of the optical apparatus [19]. For Human RVOT preparations, the left coronary artery (LCA) and the right coronary artery (RCA) were cannulated to enable RVOT perfusion and the tissue suspended vertically in a bath to enable simultaneous dual-sided optical mapping with 2 CMOS cameras. Preparations were perfused with oxygenated Tyrode solution maintained at 37°C, with a perfusion pressure of 60 to 80 mm Hg. The preparation was washed with 2 L of Tyrode solution to remove excess transplant solution and restore basal electrophysiology. Tissue was immobilized by blebbistatin (10–15 µM) to suppress motion artifacts in optical recordings, without adverse electrophysiological effects [20]. Di-4-ANBDQBS was used to map transmembrane potential as described previously [21]. Platinum-iridium tipped bipolar pacing electrode was placed on the epicardial surface. Optical action potentials were mapped from ≈5×5 cm field of view from the 4 (LV) or 2 (RVOT) surfaces using 4 or 2 MiCAM05 (SciMedia, CA) CMOS cameras with high spatial and temporal resolutions (100×100 pixels; sampling frequency, 1 KHz).

## Cap-Analysis of Gene Expression (CAGE)

This study was done on RNA extracted from human tissues procured as described above. After total RNA was extracted, as previously described, 5 µg of total RNA (260/280 2.0±0.01, 260/230 1.9±0.15, RIN 8.0±0.66) were submitted for 5'nAnT-iCAGE libraries preparation according to standard protocol [11], sequenced and demultiplexed on Illumina HiSeq2500 High throughput mode (50nt single end). In silico processing of sequenced CAGE tags was performed by using Moirai system [22]. This protocol includes quality control (fastx_trimmer: -Q33, -l 47), N base, and rRNA trimming (rRNAdust v1.0) with subsequent alignment to the human genome version hg19 through Burrows Wheeler Aligner (BWA). The median mapping ratio was 0.88±0.02 and median depth of 11.7M. CTSS (CAGE transcription start sites) and clusters of CAGE signal generated by applying python scripts: level1 and level2 [23]. The first script generates CTSS (CAGE tag starting sites) files, where 5' end of the mapped CAGE reads are counted at a single base pair resolution. The second script performs signal clustering on CTSS files with a minimum 10 TPM (tags per million) in at least one sample and minimum distance between clusters of 20 base pairs. These resulted in median 1.27M of CTSS and 13.1K of putative promoter regions, where ~12.1K overlap promoters from FANTOM5 [24]. Finally, 11612 predicted promoters were associated with 10355 genes through RefSeq and Ensembl transcripts obtained from UCSC [25] by extending the searching area of its 5' ends in ±500 base pairs. CAGE promoters for the key genes were manually curated by visualization in Zenbu browser [26]. TPM normalized CAGE counts were submitted to edgeR package for R for differential expression analysis according to the protocol [27].

## RNA-Seq analysis of gene expression

RNA sequencing was done at The George Washington Genomics Core facility. Library preparation was done with TruSeq Stranded mRNA Library Prep. Sequencing was performed on Illumina NextSeq 500 with a target of 30,000,000 reads per sample using 2x75 cycle High-Output kit. Raw reads were quality checked with FASTQC [28]. Trimming of the NextSeq sequencing adapters was done with Flexbar. Read alignment to the human genome 19 and transcript abundance were performed with Kallisto [29]. Summarization of the abundances into a matrix for the downstream analysis was done with tximport [30]. Transcript normalization was done with DESeq2 according to the published workflow [31].

## Optical mapping signals processing

In 2 left ventricular preparations and 7 right ventricular preparations after collecting tissue samples for CAGE optical APs were recorded from endocardial surface of the wedge. The tissue was paced until reaching steady state (100 seconds) at 4 different PCLs. Unless otherwise noted the PCLs used for recording AP waveform restitution were: 2000 ms, 1000 ms, 500 ms and 300 ms. We observed some variation in AP waveform over endocardial surface, therefore a single pixel recording with higher APD, upstroke velocity and overall signal-to-noise ratio was chosen manually and used as input data for GA. Low-pass filter was not used, to avoid AP waveform distortion. 60-Hz hum was removed with narrow band stop IIR Butterworth filter. Ensemble averaging over a series of APs recorded from the same pixel was used to improve signal to noise ratio (SNR).

## Algorithm verification against mRNA expression profile

When GA was applied to experimental recording, we could not directly measure ionic channel conductivities to verify the precision of GA output parameters. Instead, the following indirect approach based on transcription profiling was used (Fig 1B). Genome-wide transcription profile was measured *via* CAGE (for *Patients 1–7*) or RNA-Seq (for *Patients 8–14*) techniques as described above. We assumed ionic channels conductivity to be proportional to TPM counts (in case of the CAGE) or DESeq2-normalized counts (in case of RNA-Seq) of the mRNA encoding the corresponding pore-forming subunit protein. In particular, we considered differences in *SCN5A*, *KCNH2*, *KCNJ2*, *KCNQ1*, *CACNA1C*, *KCNA4*, *ATP1A1*, *SLC8A1*, *ATP2B4*, *RYR2*, *ATP2A2*, *CALM1*, *CAMK2D* genes to be represented in the model by INa, IKr, IK1, IKs, ICaL, Ito, INaK, INCX, IpCa, Jrel, Jup, CMDN, CaMKII parameters correspondingly (similar approach was used in [12]). Differences in these genes level of expression among *Patients 1–7* left ventricular preparations are shown in S4 Fig. In the case of CAGE group of patients, functional data was not available for *Patients 3–7*, thus only *Patients 1 and 2* were used for algorithm verification. *Patients 1* and *2* are highlighted by orange and grey colors in S4 Fig. *Patient 1* functional data from optical mapping was used as input for GA. Output *Patient 1* model parameters (ionic channels conductivities) were rescaled proportional to *Patient 2*/*Patient 1* ratio of corresponding mRNA TPM. The resulting *mRNA-based Patient 2* model was compared to *Patient 2* functional data as described below in "Results" section. Similarly *Patient 8*, *9 and 11* right ventricular AP waveform was used as input for GA, output parameters were rescaled proportional to mRNA profile as measured *via* RNA-Seq technique and compared to corresponding *Patients 9–14* AP waveform recordings. Due to computational limitations, after preliminary analysis several patients available were excluded and not used as input to GA. In particular, *Patient 10* was excluded because of a very long depolarization time (see "Experimental data" subsection of the "Results" section for a brief discussion of optical mapping artefacts affecting depolarization phase). *Patients 12* and *13* were excluded

because of the very short APD (below 300 ms, which indicated ischemia of the preparation). *Patients 12* and *14* were excluded due to low signal-to-noise ratio, while at high frequencies alternans was also observed for *Patient 14*.

## Results

### Intracellular concentrations

The immediate consequence of using steady state AP waveform dependence on PCL as GA input data is that output model AP should be steady state as well. The direct approach to the problem is to pace every organism for a long time during a GA run. However, this solution is computationally very expensive: successful GA convergence requires at least 100 organisms and 100 generations [2,4]. On the other hand, arbitrary initial state of the model requires at least 100 s to stabilize intracellular ionic concentrations. Moreover, given different initial state, intracellular concentrations converge to different values [32], although this issue is very often neglected in cardiac electrophysiology studies. Fig 2A–2D exemplifies the case. O'Hara-Rudy model was paced as 1000 ms PCL from the following initial states:

$1^{st}$ state: $[K^+]_i = 145$ mmol, $[Na^+]_i = 5$ mmol, $[Ca^{2+}]_{NSR} = 0.5$ mmol

$2^{nd}$ state: $[K^+]_i = 120$ mmol, $[Na^+]_i = 8$ mmol, $[Ca^{2+}]_{NSR} = 4$ mmol.

Fig 2B shows that these differences may have significant effects on the steady state AP waveform: resting membrane potential (RMP) difference is 4.8 mV, AP duration difference is 15 ms.

Instead of the direct approach, we have evaluated the fitness function after a short run (9 stimulations at each PCL). The final state variables at each PCL are saved and reused as initial state at the next generation (Fig 1A). However, given that the set of parameters minimizing RMSE is itself a function of intracellular concentrations, this approach results in an optimizer solving essentially a new problem every generation. Moreover, as shown above (Fig 2B–2D) "fixed" initial state for the organisms should result in a "fixed" steady state that might be different from input data. The solution that we propose in this study is to perform a simultaneous search in the parametric and slow variables space, *i.e.* the initial values of slow variables were added to parameters vector making them susceptible to mutation and crossover operators. Why would this technique eventually result in concentrations converging to a steady state? The rationale is the following: if given parameters vector results in an acceptable AP waveform (*i.e.* close to the input AP), but far from steady state, then this solution is going to be discarded eventually, since waveform is going to change after few beats. For example, if AP of particular organism on generation N minimizes RMSE, but corresponds to one of the dotted lines in Fig 2B, then RMSE is going to be large for this particular organism on generation N+1 (*i.e.* after 9 beats, one of the dashed lines).

In order to test if this technique indeed results in a steady state we have run 9 GA simulations. After that, the output models on generation 700 were paced for 1000 seconds. The observed difference between GA output state and steady state is depicted in Fig 2E and 2F. The output $[Na^+]_i$ was 0.016±0.012 mM from steady state, the $[Ca^{2+}]_{NSR}$ was 0.08±0.07 mM from steady state. S5 Fig demonstrates additional GA tests with different numbers of beats prior to fitness function. While using 5 or less beats increased output model distance to steady state, the increased number of beats resulted in minor improvement of the output model.

As we demonstrate below, this simultaneous optimization still might halt algorithm convergence after a number of generations; below we discuss several modifications to mutation operator and elitism strategy that improve optimizer performance.

## Mutation operator

The usual GA approach is to treat parameters separately by mutation operator, *i.e.* there is a fixed probability to mutate each parameter [18]. As demonstrated in Fig 3B this approach ("*point mutation*") results in a low probability to modify several parameters at the same time. RMSE dependence on generation number averaged on 9 GA runs (Fig 3C) demonstrate that slow convergence similar to coordinate descent algorithm [33] resulting from "*point mutation*" halts algorithm convergence after 100 generations. Therefore, in our GA implementation random direction in parameter space is chosen by mutation operation and the whole set of parameters is modified at the same time ("*vector mutation*", Fig 3A) resulting in better convergence (Fig 3C). Fig 3D shows that after 700 generations *vector mutation* estimates parameters much better than *point mutation*: for example, the error is 3±3% vs 7±8% for $I_{K1}$, 1.6±1.6% vs 5±5% for $I_{Kr}$, 4±3% vs 11±8% for $I_{Na}$.

Cauchy distribution is a "*pathological*" distribution with infinite variance and expected value. As was noted previously [17] Cauchy mutation tends to generate offspring far away from its parent. Consequently, it prohibits algorithm stagnation in local minima and generally results in better convergence for multimodal functions. Fig 5A compares two sample runs of GA with polynomial and Cauchy mutation. In case of polynomial mutation by generation 300 the whole population converged to a vicinity of a single solution that is different from input model set of parameters (red line). We observed slow convergence on subsequent generations simultaneous with slow intracellular concentration changes (compare to S6 Fig plotting corresponding intracellular concentration changes of the best organism). In the case of Cauchy mutation, every parameter except $I_{NaK}$ converged to some vicinity of input model value by generation 300. At the same time, we did not observe algorithm stagnation: note, for example, the wide variation of $I_{K1}$ and RyR parameters on generation 500. Consequently, we observed more robust algorithm convergence in case of Cauchy mutation for model fitting to synthetic AP data (Fig 5B). For example, the error of model parameters was 14±24% *vs* 17±75% for $I_{Ks}$, 11±25% *vs* 45 ±23% for $I_{to}$, 6±6% *vs* 13 ±15% for $I_{CaL}$ in GA runs with Cauchy and polynomial mutation correspondingly. These results might imply that wider polynomial mutation would result in better convergence as well. In order to test this assumption, we ran a number of GA tests with different distribution parameters. As shown by RMSE dependence on distribution parameters shown in S3 Fig, wider polynomial mutation did not improve the algorithm convergence.

## Elitism strategy

Genetic operators tend to spoil a "good" solution; therefore, best organisms are passed to the next generation without any modifications [3,4]. We have found that given the wide parametric variation by Cauchy mutation in our GA implementation a large number of elite organisms is required for fast GA convergence. While wide *exploration* of parametric space is required at the initial stage of algorithm convergence, it is more effective to *exploit* the global minimum once it was localized (see [34] for discussion on *exploration* and *exploitation*). This could be achieved by a large number of elite organisms passing their parameters *via crossover* operator to siblings after clustering around the same minimum.

Principal component analysis (PCA) comparison of sample GA runs (Fig 4A–4D) shows that in case of higher proportion of elite organisms (6.6%) solution tends to cluster around the precise solution at generation 100, while a GA run with 3.3% of elite organisms requires at least twice the number of generations to converge the population to the same cluster size. As further explained in S2 Fig high proportion (6.6% or 3.3%) of elite organisms results in the fast reduction of the cluster size (Standard Distance of the population, S2C Fig). This reduction, in

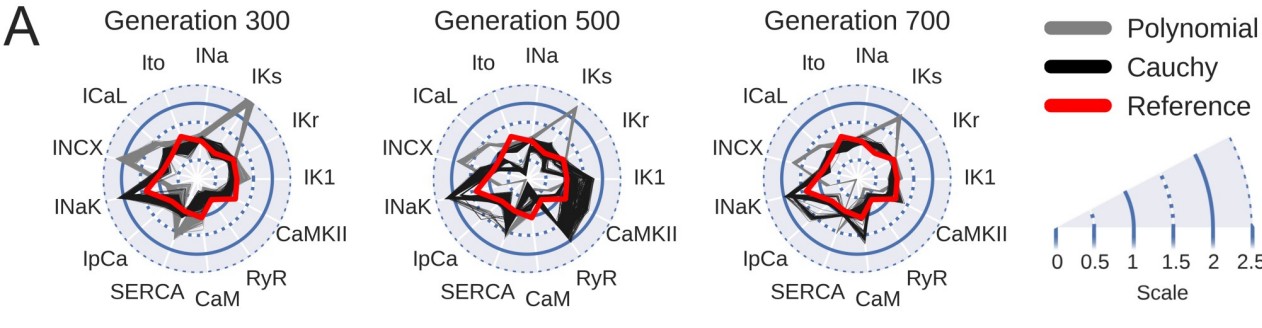

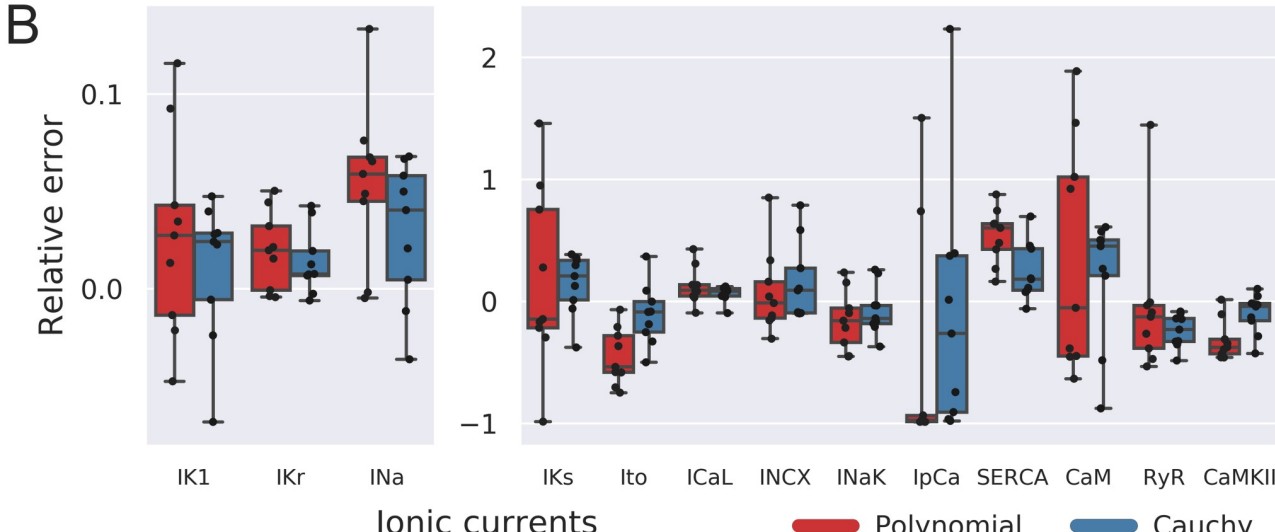

**Fig 5. Cauchy mutation.** (**A**) Radar plots illustrating algorithm convergence in case of Cauchy (black) vs Polynomial (grey) mutations. Synthetic AP with parameter values shown by red line was used as input data. Axes show parameters ratio to corresponding O'Hara-Rudy model [12] values. (**B**) Objective parameters distribution for 9 GA runs with Polynomial mutation (red boxes) and Cauchy mutation (blue boxes). Dashed line corresponds to the input model parameter values. All values shown correspond to the best organism on generation 700.

turn, allows to *exploit* the solution by the algorithm: mean cluster error (MCE) has a clear trend after the reduction of the cluster size (S2B Fig), while MCE for the low proportion of elite organisms (0% or 0.3%) follows random fluctuations. In other words, large number of *elite* organisms result in two-stage optimization: initially the whole parametric space is explored, but eventually a number of *elite* organisms converge to the same minimum attracting the whole population to its vicinity and resulting in effective local optimization. Although PCA plots did not account for intracellular concentrations (see Methods section) it could be seen from comparison of S6 Fig (Cauchy mutation) and S2 Fig (6.6% *elite*) that actually convergence to global parametric minimum (MCE and SDist reduction) is simultaneous with concentration changes to some vicinity of input model steady state value.

The requirement of a high number of *elite* organisms indicates that interbreeding *via crossover* operator does not necessarily result in improved results. In order to test whether *crossover* operator is indeed essential to algorithm convergence we have ran 7 GA runs without the *crossover*. Indeed, as seen in S7 Fig, leaving out the crossover still results in decent convergence, however some output parameters are much less precise, in particular the error is 100±50% vs 19±23% for IKs, 93±54% vs 32±25% for SERCA, 200±140% vs 26±14% for RyR.

The RMSE dependence on the proportion of elite organisms to the whole population (S8 Fig). We observed slower algorithm convergence, when the number of elite organisms was below 4% of the whole population. On the other hand, the increase of the proportion of elite organisms above 10% typically resulted in algorithm stagnation after generation 100. Due to random nature of the algorithm convergence, the convergence speed is susceptible to fluctuations, but optimal proportion of *elite* organisms could be estimated as 6–10% of the whole population from S8 Fig.

### Final algorithm

In order to test if the new algorithm narrowed down the solution range, we have compared our algorithm with the original one [3] (Fig 6). We have re-implemented the original algorithm by Bot et.al. using Sastry toolbox [35] with the following modifications to it. Firstly, each organism was paced for 50 stimulations at every PCL, *i.e.* quasi-steady-state was used because

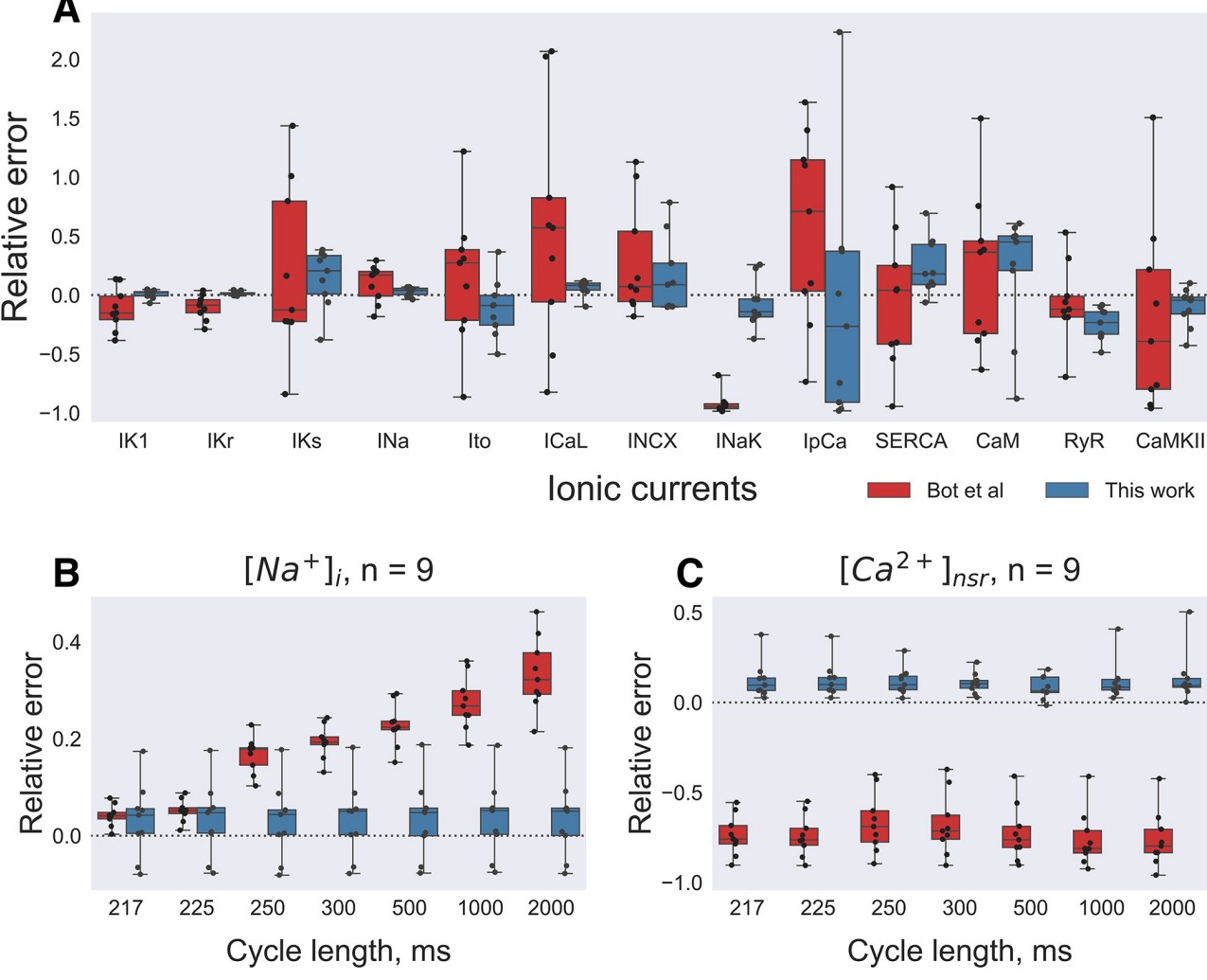

**Fig 6. Comparison of presented algorithm precision to Bot et al. (A)** Parameter scaling by the presented algorithm (blue boxes) as compared to implementation of [3] (red boxes) after a long evolutionary run for 700 generations. Dashed line corresponds to input model parameter value. **(B,C)** Output intracellular concentrations comparison between 2 algorithms.

it was computationally very expensive to reach actual steady state. Secondly, the normalized AP waveform at 7 PCLs was used as input data. Finally, the least squares technique was used to renormalize input data prior to *fitness function* evaluation as described in the Methods section. As shown in Fig 6 our algorithm determined $I_{K1}$ conductivity with 3±3% precision, $I_{Kr}$− 1.6 ±1.6%, $I_{Na}$− 4±4%, $I_{CaL}$—8±6% (vs. 17±17%, 11±10%, 16±15%, 86±94% error, respectively, compared to the original algorithm). Membrane currents that did not have profound effects on the AP waveform were less precise ($I_{Ks}$, $I_{to}$, $I_{NCX}$, $I_{NaK}$− 24±24%, 21±23%, 25±31%, 18±19% error, respectively), however, precision was still better then original algorithm (56±69%, 46 ±55%, 36±46%, 92±9%, respectively). Model parameters that did not affect AP waveform directly (RyR, SERCA and CAMKII) were also determined with an error less than 69%: 25 ±13% vs 25±32% for RyR, 25±22% vs 46±55% for SERCA, 14±16% vs 68±77% for CAMKII. As shown in Fig 6B the modified algorithm output intracellular concentration are also close to the input model precise values: for $[Na^+]_i$ the difference was 0.2±0.6 mmol/l, while for $[Ca^{2+}]_{NSR}$ the difference was 0.3±0.2 mmol/l.

### Input data requirements: Restitution information

In the results described above, input data was APs simulated at 7 different PCLs (see Methods section). However, in real experimental setting it is preferable to reduce the recording time including the number of PCLs at which steady state AP waveforms are recorded. We have tested GA performance, when input AP was simulated at limited number of pacing frequencies. Particular PCLs we used to compare algorithm performance are listed in Fig 7A. We observed that for some parameters (Ito, SERCA: S9E and S9J Fig) single AP waveform recorded at 1000 ms was sufficient (*i.e.* algorithm precision did not increase when more detailed restitution properties were used). Two extreme points on the restitution curve (217 ms and 2000 ms) were required to determine $I_{Kr}$ conductivity with 1.6±1.6% precision (S9B Fig). The parameters most sensitive to the AP restitution are shown in Fig 7B–7F. It is preferable to use full restitution curve (7 PCLs) for $I_{Ks}$, $I_{Na}$ and $I_{CaL}$. However, 4 points on restitution curve still result in acceptable accuracy for all other parameters. While using considerably different target parameter values would affect the results shown in this section, it provides an estimate of the amount of restitution information required to determine identifiable parameters. Consequently, we have recorded the AP waveform at 4 PCLs in the experiments described below.

### Input data requirements: Signal-to-noise ratio

The other inherent problem of experimental data that might spoil algorithm performance is noise. To test how noisy data would affect the algorithm precision we have added Gaussian noise to the input simulated AP. Sample AP waveforms with different signal to noise ratio (SNR) are depicted on Fig 8E and 8F and S10 Fig. As expected, noise amplitude heavily affected precision of the output parameters. As seen on Fig 8A–8D and S11 Fig 20 dB SNR completely breaks down algorithm: for example, error is up to 100% for $I_{Kr}$ and up to 227% for $I_{Ks}$ conductivities. 28 dB SNR, achievable in experimental setting gives better results: the observed error is 1±2% for $I_{Kr}$, 6±3% for $I_{Na}$, 17±9% for $I_{CaL}$, 23±11% for $I_{NaK}$. As also demonstrated by histogram and probability plot on S12 Fig experimental noise in the optical mapping setting is indeed close to Gaussian (see corresponding section of S1 Text for details).

### Experimental data

GA was also tested with optical APs as input data (see Fig 1B and "Methods" section). We observed some degree of APD heterogeneity in the most of human wedge preparations that we

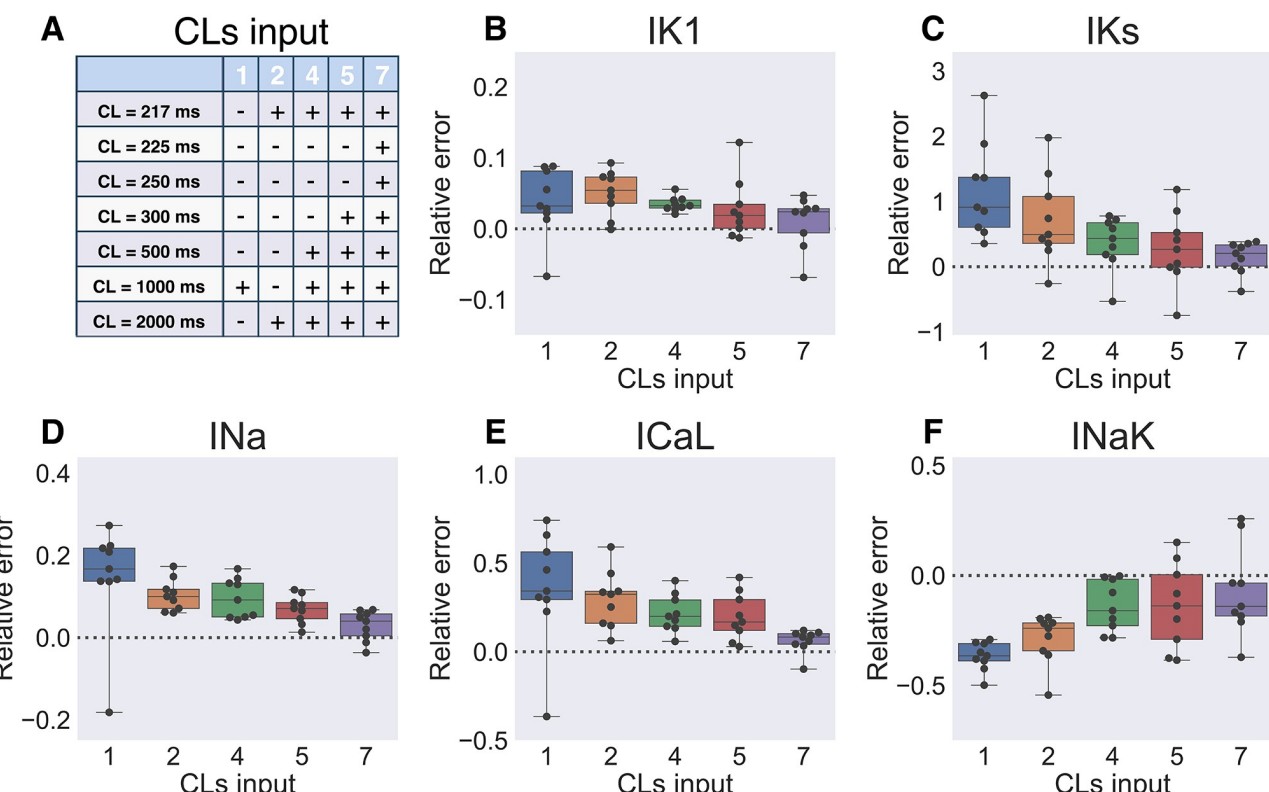

**Fig 7. Solution sensitivity to the number of input baselines.** (**A**) Input model AP was recorded at several PCLs listed in the table. Larger number of input baselines results in better algorithm performance. (**B**-**F**) Box-and-whiskers plots depict the parameters most sensitive to the changes in the number of input baselines ($I_{K1}$, $I_{Ks}$, $I_{Na}$, $I_{CaL}$, $I_{NaK}$) at generation 700 of 8 GA runs. Dashed line corresponds to the input model parameter value.

have optically mapped for this study. In particular, the *Patient 1* APD80 was 314±63 ms (APD map is shown on S13 Fig). One possible explanation of this heterogeneity is uneven perfusion of the sample resulting in mild ischemia that shortens AP duration because of activation of the ATP-dependent potassium channels $I_{K,ATP}$. Since ATP-dependent potassium current was not accounted for by the model, we have chosen an AP with the longest APD as and input to GA. Fig 9A compares GA output model with input data. The *Patient 1* model faithfully reproduced AP waveform dependence on the PCL, however, we observed some deviations between model and experiment. The deviations between input data and the output model are listed in Table 1. While RMSE is close to the noise level, and APD80 error did not exceed 14 ms, the difference in the depolarization phase is very pronounced: $(dV/dt)_{MAX}$ was approximately 20 V/s for input data, while for the output model it ranged from 55 to 80 V/s. This effect might be due to photon scattering in optical mapping recordings [14, 15]. Photons emitted by fluorescent dye undergo multiple scattering events and thus the recorded signal from a given pixel is actually an averaged signal from thousands of myocytes. This effect is known to distort AP waveform during depolarization when differences of membrane potential across the tissue are significant due to propagation of wavefront of excitation. Thus, in order to reduce the effect of this experimental artifact on the model, initial depolarization phase (below -20 mV) was removed from compared AP prior to fitness function calculation (see also Methods section).

As described above in the Methods section, in order to verify output parameter values the *Patient 1* model parameters were rescaled proportional to the ratio of mRNA expression levels between *Patient 2* and *Patient 1*. The technique is based on the assumption that ionic channels

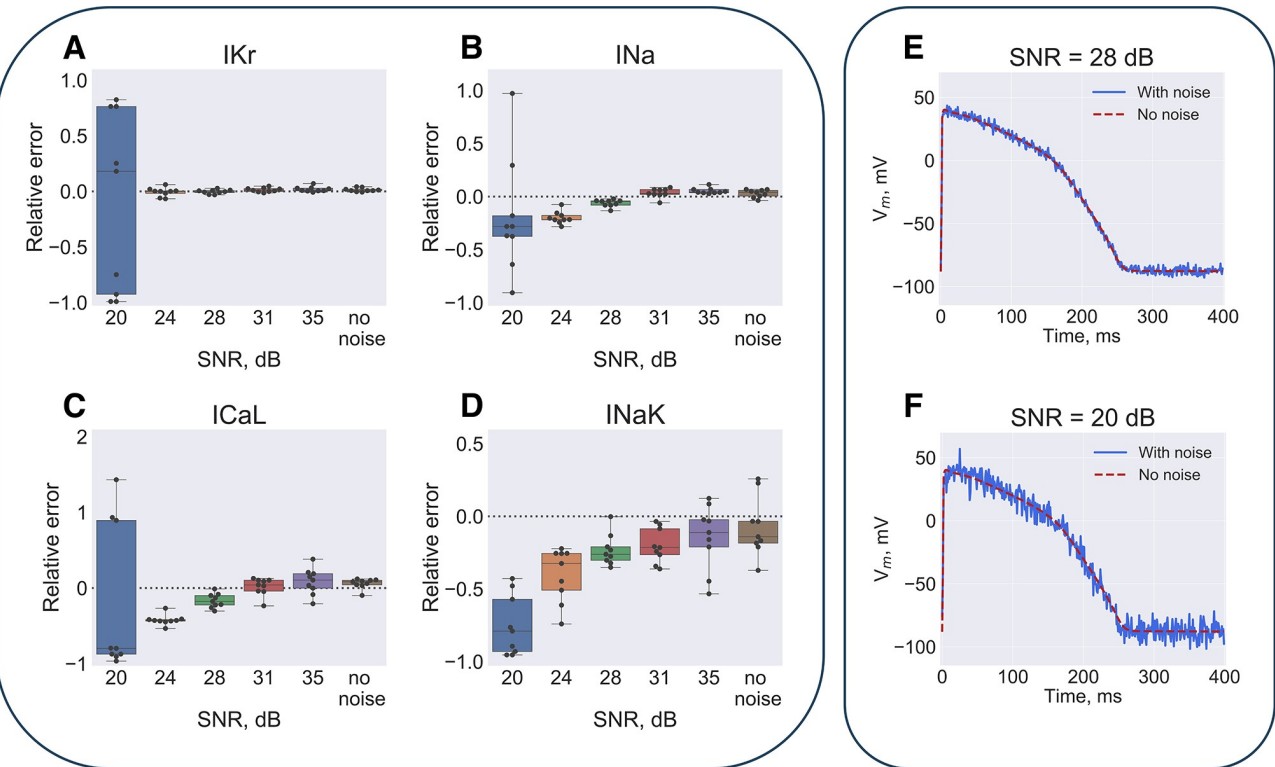

**Fig 8. Solution sensitivity to the input baselines signal-to-noise ratio (SNR).** The input model AP waveform was distorted by Gaussian noise. (**A**) Parameters dependence on the SNR is illustrated at generation 700 of 9 GA runs. Dashed line depicts the input model parameter value. (**B**) Sample AP waveform at 28 dB and 20 dB SNR.

conductivities are proportional to the mRNA counts (and, consequently, to the amount of protein being expressed) as measured by CAGE. If GA output parameters are close to actual conductivities in *ex vivo* tissue sample, then a model with parameters rescaled proportionally to the ratio of mRNA levels is going to reproduce another patients AP waveform. Thus, we have compared rescaled *Patient 2* model to the optical mapping data recorded from the corresponding heart. As shown on Fig 9B and Table 2 *Patient 2* model still faithfully reproduced AP waveform at every PCL, however, AP waveform error was more pronounced. Particular spots of AP recordings for Patient 1 and Patient 2 are shown in S13 Fig.

Using a similar technique, we have reconstructed personalized models for other 5 patients, AP waveform and restitution curves are depicted in Fig 9C and 9D. As noted above, functional data was not available for these particular patients, but the variability between the models is within physiological range [36]. *Patient 5* and *Patient 7* APD was too short (below 200 ms), which is explained by the fact that mRNA levels of expression had the most extreme deviations from median value in these patients. *Patient 5* was an outlier in terms of *ATP1A1*, *ATP2A2*, *ATP1B3*, *ATP2C1*, *KCNJ5*, *KCNK3* genes. *Patient 7* was an outlier in terms of *ATP1A1*, *ATP1B1*, *ATP1B4*, *CACNA1C*, *CACNA2D1*, *CACNA2D3*, *CACCB1*, *CACNB2*, *CALM1*, *CALM3*, *KCNH2*, *KCNJ11*, *KCNJ3*, *KCNK1*, *KCNK3*, *CAMK2B*, *CAMK2D* genes (see also S4 Fig). These deviations might indicate problems with heart preservation prior to tissue collection or undiagnosed heart diseases.

To further study the precision and limitations of mRNA-based rescaling, we have tested this technique on a number of RV preparations. We have to note here, that while apical RV

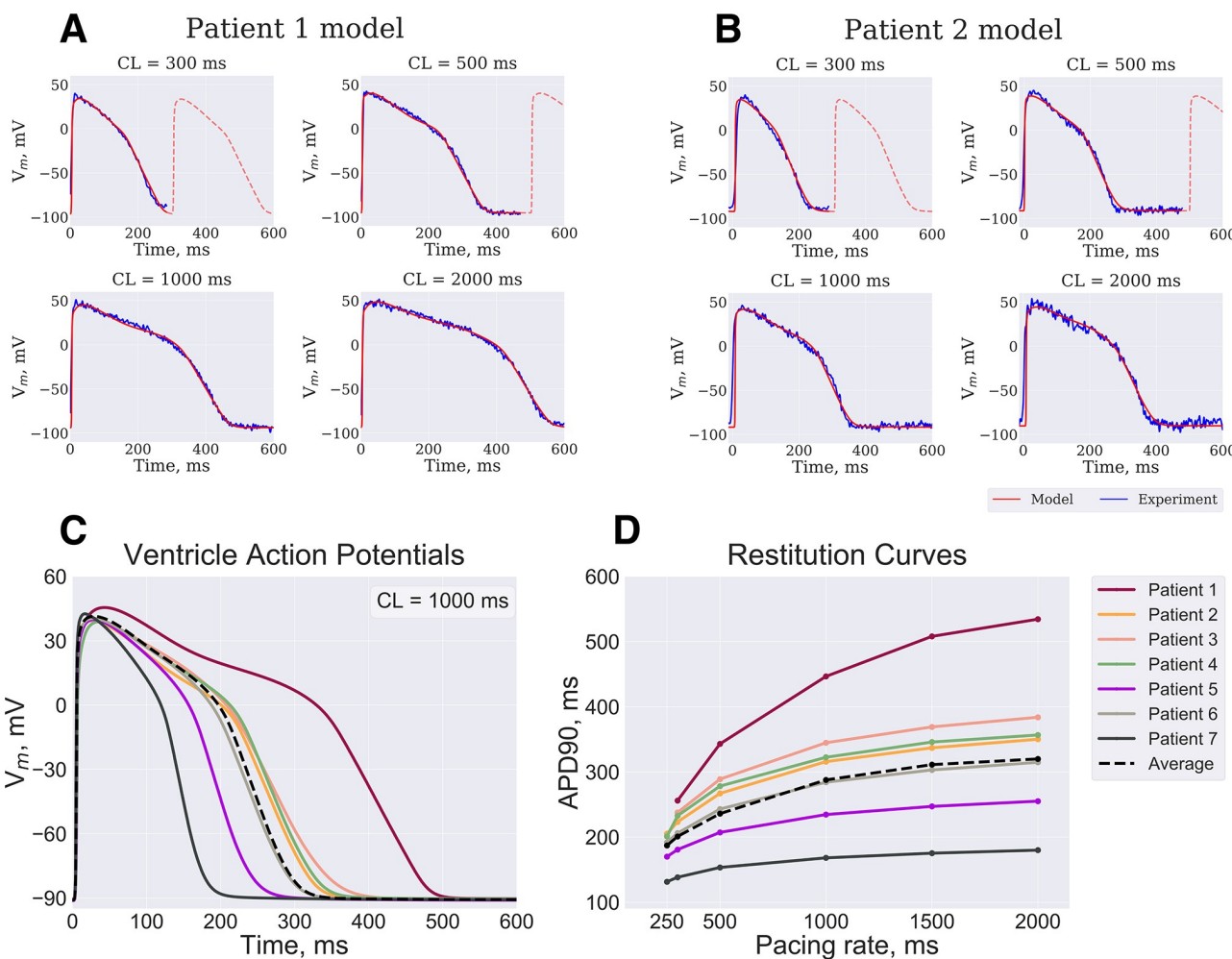

**Fig 9. Algorithm verification.** (**A**) Patient 2 (see text for details) model APs (red line) fitted by GA as compared to the optical APs (blue) recorded at CL of 2000 ms, 1000 ms, 500 ms and 300 ms. (**B**) Patient 1 model was rescaled using mRNA expression data and compared to experimental APs (see Fig 1B). Personalized models based on mRNA expression data from 7 donor hearts ventricles demonstrate different AP waveform (**C**) and restitution properties (**D**).

region was used for tissue collection and consequent transcriptome analysis, RVOT preparation was used for optical mapping studies. As seen on APD maps on Fig 10 in general we observed shorter APD close to the pulmonary artery (upper half of the preparation). Given the heterogeneity of the sample, we supposed the longer AP to represent typical RV tissue. Particular spot used for GA input is shown by red cross in Fig 10. As shown in Fig 11 and Table 3 the GA output model reproduced the AP waveform and restitution.

Similarly to the case described above, we have rescaled GA-optimized RV model parameters proportionally to mRNA expression ratios as measured *via* RNA-seq technique. Resulting

**Table 1. *Patient 1* GA-optimized model error.**

|  | CL = 300 ms | CL = 500 ms | CL = 1000 ms | CL = 2000 ms |
|---|---|---|---|---|
| $\Delta(dV/dt)_{MAX}$ | 38 V/s | 53 V/s | 34 V/s | 58 V/s |
| RMSE | 6.3 mV | 4.0 mV | 5.8 mV | 3.8 mV |
| $\Delta$APD80 | 14 ms | 3 ms | 8 ms | 3 ms |

**Table 2.** *Patient 2* CAGE-based model error.

|  | CL = 300 ms | CL = 500 ms | CL = 1000 ms | CL = 2000 ms |
|---|---|---|---|---|
| $\Delta(dV/dt)_{MAX}$ | 69 V/s | 64 V/s | 47 V/s | 36 V/s |
| RMSE | 8 mV | 12 mV | 11 mV | 16 mV |
| $\Delta APD80$ | 7 ms | 4 ms | 10 ms | 9 ms |

models did reproduce the experimental AP waveform and restitution. The comparison of resulting *Patients 9–11 AP* waveforms with experimental recordings are given in Fig 10 and Table 4.

As demonstrated by violin plots in Fig 11 the sample was very heterogeneous, thus mRNA-based model reproduced an experimental recording from a particular spot of the preparation (the particular pixel that was used for AP comparison is shown on APD maps in Fig 10). Violin plots demonstrate that in the cases of Patients *8*, *10 and 11* the distribution was bi-modal, while mRNA-based model AP represents the longer mode of distribution. Since the apical RV region was used for tissue collection, we hypothesize that the longer mode of the distribution represents typical RV tissue, while the shorter mode represents RVOT-specific tissue.

On the other hand, as shown in Fig 10 (red AP) the rescaled models failed to reproduce the experimental AP waveform for *Patients 12–14*. In the cases of *Patients 12 and 13*, the sample was most probably ischemic, since APD was below 300 ms. In the case of *Patient 14* the lower half of the sample, that we hypothesize to represent typical RV tissue, was mostly very noisy. Thus, the longer mode of the APD distribution might be obscured in this particular case.

Fig 10 also demonstrates GA-optimized models of *Patients 9* (blue line) and *11* (green line) as rescaled to all other RV preparations. Although GA-output models were relatively close to input AP waveform (RMSE was below 7.4 mV for every AP waveform), the mRNA-based rescaling failed to reproduce the other patients AP waveform. Table 5 lists the parameter values for three variants of *Patient 9* model: GA-output model, and two transcription profile-based models. It should be noted that in the case of GA-output *Patient 11* model the sodium current is particularly low, resulting in subthreshold depolarizations of corresponding mRNA-based *Patients 9* and *10* models. As mentioned above, this artifact is caused by slow depolarization phase of optical signal, which resulted in low $I_{Na}$ scaling parameter in this particular GA run. It is also interesting that, as seen on Fig 10, AP for two models of *Patient 9* are relatively close: *Patient 9* GA-output model (blue line) and *Patient 8* based model after parameters rescaling (red line). However, parameter sets shown in Table 5 are very different, in particular $I_{NCX}$, $I_{NaK}$ and SERCA conductivities are 1.5–3.5 times higher in the latter case, while $I_{Na}$ is 2.5 times lower. Surprisingly, in this case, the GA-output model actually performed worse than the mRNA-based model, the difference is most prominent at 250 ms PCL: in the former case (*Patient 9*-based model) APD80 error was 26 ms, in the latter (*Patient 8*-based model) APD80 error was 3 ms. On the other hand, this difference resulted only in slight $RMSE_{250ms}$ difference: 5.7 vs 5.0 mV correspondingly. This implies high sensitivity of GA to AP perturbations that was shown above by noise sensitivity analysis: GA-output model should follow experimental AP waveform closely and minor experimental artefacts might cause major change in parameter values. *Patient 8* based models reproduced AP waveforms for 4 hearts, which indicates model precision for this particular case. This is also in line with the fact that GA-output model total RMSE was lower for *Patient 8* than for *Patient 9*: *15.6 mV* against *24.5 mV*. It is also interesting that despite major differences in parameters, models behavior was surprisingly similar not only in the case of *Patient 9*, but also in a number of other cases. At least partially, this could be explained by similarities in expression profiles: for example, differences between

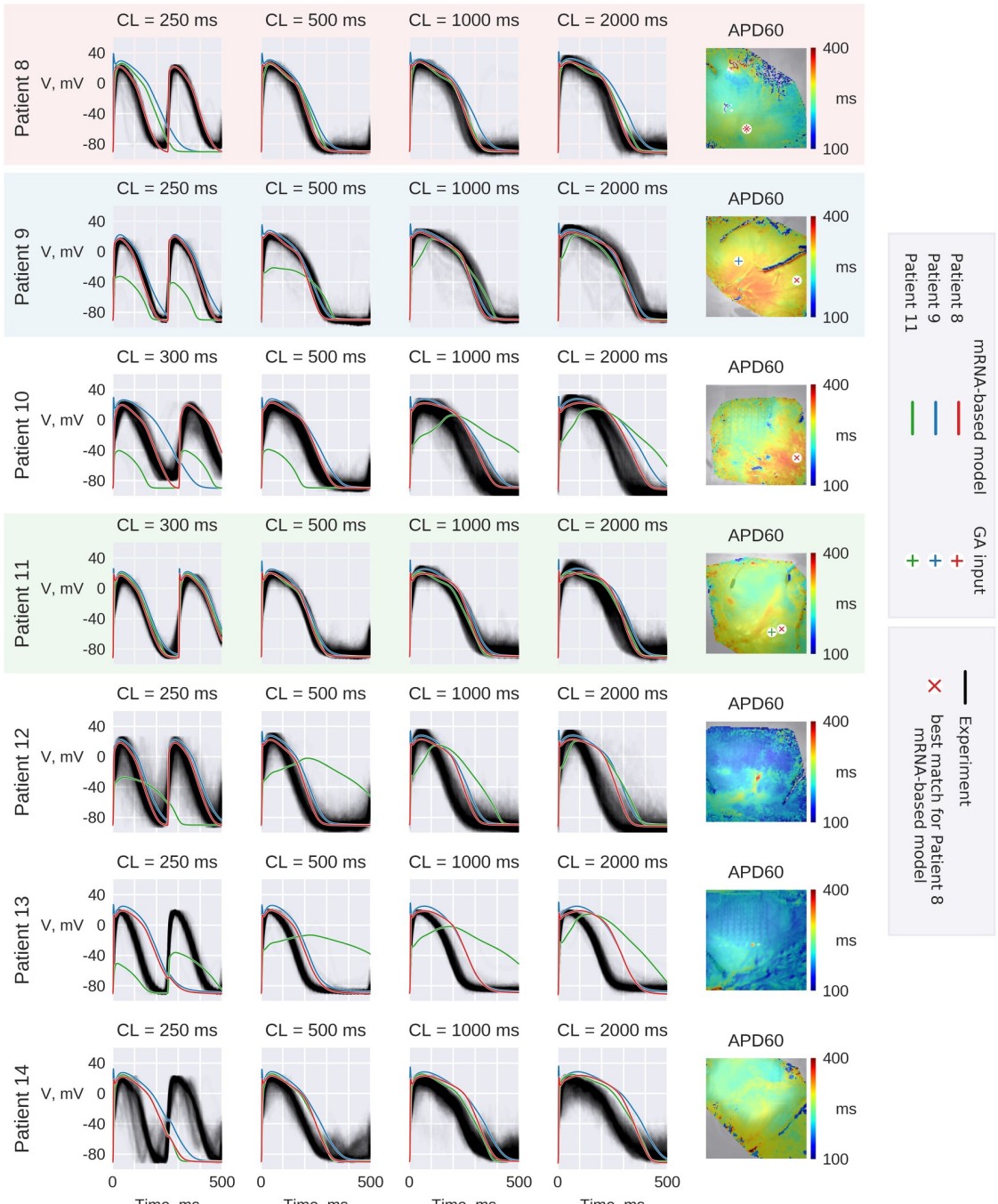

**Fig 10. RVOT experimental recordings.** Gray lines depict AP waveforms recorded from the wedge preparations. Experimental waveforms are aligned to match the time corresponding to $(dV/dt)_{MAX}$. Red lines correspond to *Patient 8* GA-output model with parameters rescaled according to *Patients 9–14* mRNA profiles. Similarly, blue and green lines correspond to *Patient 9* and *11* based models correspondingly. The pixels of AP waveform recording that was used as input to GA is labeled by "+" symbols on APD maps. The pixel of the recording that was used on Fig 11 to compare *Patient 8* based models with experimental AP is marked by red x on the APD maps.

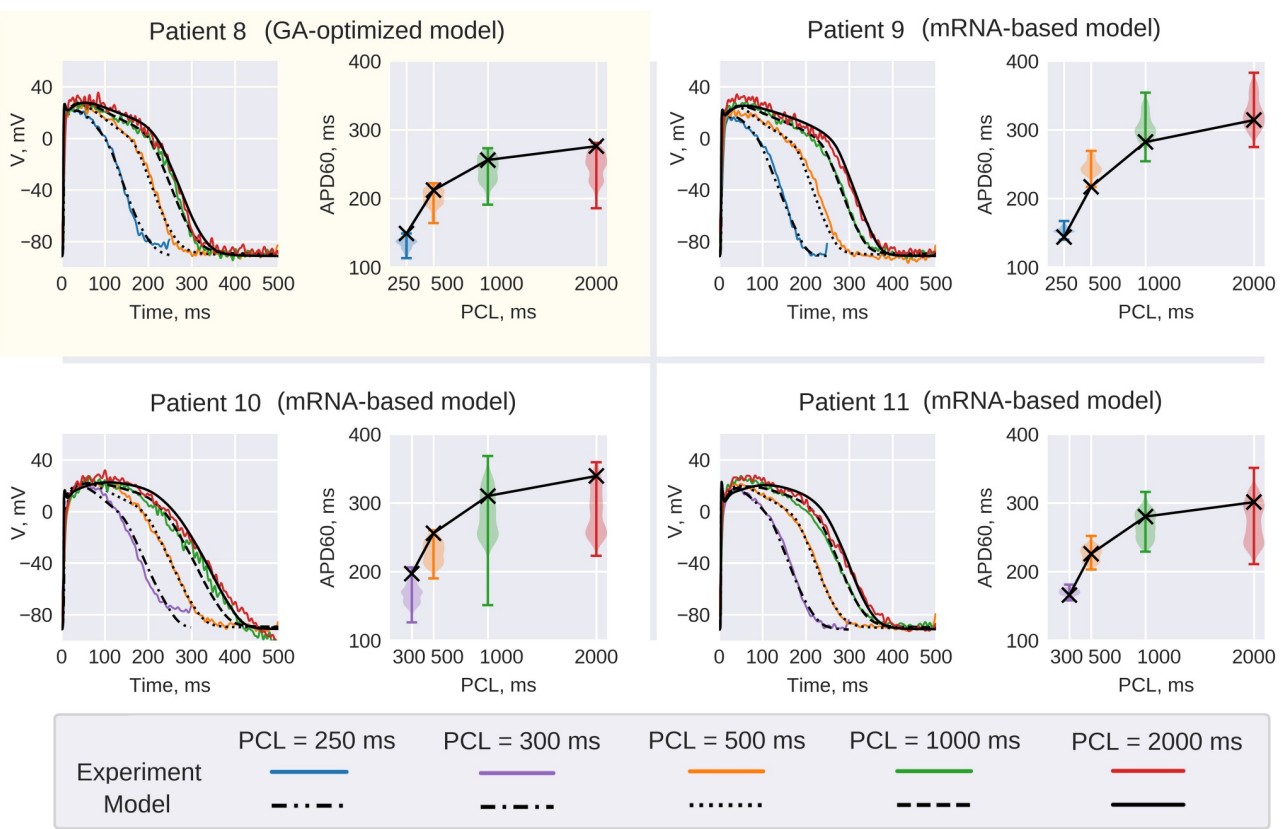

**Fig 11. *Patient 8* GA output model.** *Patient 8* GA-output model (top left corner) comparison with experimental AP recordings. Red, green yellow and blue lines correspond to AP recorded at 2000, 1000, 500 and 250 ms PCL correspondingly. The pixel of recording is shown on APD maps in Fig 10. Model restitution curve (black line) is superimposed on violin plots depicting APD60 distribution in the wedge preparation. Other panels use the same convention to compare *Patient 8* model rescaled according to RNA-seq data to *Patients 10–12* AP recordings.

*Patient 11* and *Patient 13* genes expression corresponding to $I_{Kr}$, $I_{Ks}$, $I_{to}$, $I_{CaL}$, RyR and CaMKII were less than 9% (see Table B in S1 Text).

## Discussion

In this study, firstly, we have introduced a novel GA modification allowing one to personalize cardiac electrophysiology models using steady-state AP recordings dependence on PCL as input data. The algorithm modification is based on the idea that parametric optimization and slow variables steady state search should be performed simultaneously for effective convergence. Secondly, we have tested the modifications we introduced on synthetic action potentials proving our modifications to be advantageous for overall algorithm performance. Finally, we have tested the algorithm performance against cardiac optical mapping experimental data and mRNA expression profile. The output parameters precision was confirmed by the observation

**Table 3. *Patient 8* GA-optimized model error.**

|           | CL = 250 ms | CL = 500 ms | CL = 1000 ms | CL = 2000 ms |
|-----------|-------------|-------------|--------------|--------------|
| RMSE      | 3.9 mV      | 3.2 mV      | 3.6 mV       | 5.0 mV       |
| ΔAPD80    | 10 ms       | 1 ms        | 7 ms         | 7 ms         |

**Table 4.** *Patient 9–11* RNA-seq based models error.

|  | CL = 250 ms | CL = 300 ms | CL = 500 ms | CL = 1000 ms | CL = 2000 ms |
|---|---|---|---|---|---|
| *Patient 9* | | | | | |
| RMSE | 4.9 mV |  | 5.2 mV | 3.7 mV | 5.3 mV |
| ΔAPD80 | 3 ms |  | 17 ms | 5 ms | 4 ms |
| *Patient 10* | | | | | |
| RMSE |  | 5.9 mV | 5.4 mV | 6.4 mV | 5.7 mV |
| ΔAPD80 |  | 19 ms | 4 ms | 30 ms | 13 ms |
| *Patient 11* | | | | | |
| RMSE |  | 4.9 mV | 4.5 mV | 4.3 mV | 5.5 mV |
| ΔAPD80 |  | 4 ms | 8 ms | 9 ms | 13 ms |

that mRNA-based models predict patients AP waveform and restitution. Essentially, a combination of GA with the mRNA expression measurements provides a novel technique for model personalization.

## The algorithm modification

Many different combinations of model parameters result in the same AP waveform. However, these solutions are mostly non-steady state: AP waveform diverges from experimental in the long run. Implicit steady state requirement allowed us to substantially narrow down the parameters range. As shown above (Fig 7, S9 Fig), using AP waveform dependence on PCL as input data further increased algorithm precision.

Reaching steady state for every organism is computationally very expensive. To solve the problem, parameter search and steady state search are performed at the same time. Model gets closer to steady state after a short simulation, after that final model state is saved, the fitness function is evaluated, and parameters are modified by genetic operators. Next generation simulations start from new initial state bringing the model closer to steady state. If given solutions

**Table 5.** Parameter values (relative to baseline O'Hara-Rudy model [12]) of *Patient 9* models derived via GA or via comparison of mRNA levels to reference patient.

| *Reference patient* | *Patient 8* (mRNA-rescaled model) | *Patient 9* (GA-optimized model) | *Patient 10* (mRNA-rescaled model) |
|---|---|---|---|
| IK1 | 0.457 | 0.283 | 0.618 |
| IKr | 1.226 | 1.002 | 0.806 |
| IKs | 0.149 | 0.143 | 0.306 |
| INa | 0.989 | 2.542 | 0.550 |
| Ito | 8.218 | 9.743 | 4.227 |
| ICaL | 0.606 | 0.830 | 0.405 |
| INCX | 2.991 | 1.772 | 1.459 |
| INaK | 4.136 | 1.220 | 3.353 |
| IpCa | 1.066 | 0.170 | 0.307 |
| SERCA | 4.690 | 2.227 | 3.436 |
| CaM | 10.112 | 3.762 | 1.739 |
| RyR | 0.825 | 0.690 | 0.470 |
| CaMKII | 0.163 | 0.211 | 0.902 |

of the optimization problem do not reproduce input data in steady state, then RMSE starts to increase as the algorithm goes and these solutions are going to be discarded by the algorithm eventually. Without further modifications this approach has two limitations: firstly, as demonstrated in Fig 2A–2D steady state is dependent on initial state, thus single steady state for the algorithm would fail to explore all the possibilities; secondly, the slowest variables still require very long run (many generations) to approach steady state. In order to address these issues, two slow variables: intracellular sodium concentration and sarcoplasmic reticulum calcium load are treated as model parameters by the algorithm, i.e. mutation and crossover operator are applied to these variables as well. As shown above beneficial byproduct of this approach is that intracellular ionic concentrations are determined by the algorithm with relatively high precision: for $[Na^+]_i$ the error in the test runs was 0.2±0.6 mmol/l, while for $[Ca^{2+}]_{NSR}$ the error was 0.3±0.2 mmol/l. We have to note here, that several groups of researchers did observe restitution hysteresis as well as alternans hysteresis [37, 38, 39]. Thus, we can hypothesize that multiple steady states is an inherent feature of a myocyte that might be personalized by the GA. On the other hand, living cardiomyocyte is a complicated dynamic system and ionic channel conductivities themselves could change over the time [40].

Albeit premature convergence to local minimum of RMSE hinders the solution of the optimization problem, it is preferable to *exploit* the vicinity of previously visited points once global minimum is localized. This is accomplished in this study by a large number of elite organisms. When *elite* organisms fall in some vicinity of RMSE minimum two scenarios are possible. If given solution is far from steady state they are going to be either discarded because of major variation of AP waveform in few generations (since these organisms are not susceptible to modification by *mutation*). If slow variables are close to steady state, *elite* organisms start to attract the population to the same vicinity. In the latter case, the whole population is going to *exploit* the vicinity of the optimization problem solution, thus working as a local optimizer. As seen from the comparison between S6C and S6D and S2B and S2C Figs intracellular concentrations and parametric convergence share similar dynamics when ratio of elite organisms is high. On the other hand, it might be more effective to use either classic gradient-based methods or Covariance Matrix Adaptation Evolution Strategy [41] once global minimum is localized (*i.e.* when a number of *elite* organisms converged to the same vicinity).

The convergence speed of the algorithm was improved by further modifications in the mutation operator. We found that mutating each parameter with a fixed probability (referred to as "*point mutation*" above) resulted in a single parameter being modified by mutation operator (Fig 3B). This kind of descent is very slow in 27-dimensional parameter space (parameters include intracellular concentrations at each PCL) and usually stopped convergence after about 100 generations (Fig 3C). Instead, we chose random direction in multidimensional parameter space and mutated parameter in this direction using Cauchy distribution, resulting in better algorithm performance (Figs 3 and 5).

It is worth noting that all these modifications to GA essentially make the algorithm similar to particle-based algorithms. For example, multivariate jumps are used by Particle Swarm [42], Covariance Matrix Adaptation Evolution Strategy [41] and Cuckoo Flight [43] algorithms. The last one uses "pathological" distribution similar to Cauchy mutation for Levy-flight walk. Although it would be very interesting to compare these algorithms performance with GA modification proposed in this study, some custom modifications to the aforementioned algorithms are still required: as was noted above same AP waveform could be reproduced by different parameter sets [8]. In order to narrow down the output parameters range, not steady state solutions should be somehow automatically discarded by the algorithm. This kind of extensive benchmarking is beyond the scope of the current study.

## The algorithm verification against synthetic data

Test runs indicate that resulting algorithm evaluates the most important model parameters ($I_{K1}$, $I_{Kr}$, $I_{Na}$ and $I_{CaL}$) with high precision (Fig 6). However, low amplitude ionic currents and parameters affecting calcium transients were much less precise. We further tested the algorithm sensitivity to noise to find that 20 dB SNR breaks the algorithm, while SNR above 28 dB has minor effects on algorithm performance (Fig 8). Another point should be discussed in this regard: while the algorithm is relatively stable, SNR requirements are quite strict. It was possible to achieve 28 dB level noise in *ex vivo* optical mapping experiment; however, it still required some post-processing: hum removal and ensemble averaging in particular. These strict requirements might hinder algorithm application to clinical recordings prone to artifacts distorting the signal. As discussed below a combination of GA with mRNA expression profile could help to solve the problem of model personalization using clinically-measurable data.

## The algorithm verification against experimental data

We did not measure ionic channels conductivities in this study, thus we could not directly verify algorithm output parameters precision against experimental data. Instead, we used an indirect approach based on the following strong assumptions. Firstly, we assumed that given ionic channel conductivity should be proportional to corresponding proteins mRNA expression level. One thing to note here is that actual ionic channel is assembled from several proteins. For example, cardiac sodium channel is assembled from pore-forming subunit encoded by SCN5A gene and auxiliary subunits encoded by SCN1B and SCN2B genes [44]. The situation is more complicated in the case of IK1, which is most likely a heterotetrameric complex assembled from proteins encoded by KCNJ2, KCNJ4 and KCNJ12 genes in any combination [45]. Thus, our second assumption was that a single pore-forming subunit level of expression could predict ionic channel conductivity. If these assumptions hold and if GA output parameters correspond to actual ionic channels conductivities in one patient, then model of another patient model could be reproduced by simple rescaling of the conductivities.

Indeed, two-patient comparison of LV preparations have shown that mRNA-rescaled model reproduced *Patient 2* AP waveform at every PCL (Figs 1B and 10). We have undertaken more extensive study on RV preparations that demonstrated that *Patient 8* GA output model reproduced AP waveform and restitution in three cases (*Patients 9–11*) after parameters rescaling (Fig 11), however it failed to reproduce the AP waveform for 3 other patients (*Patient 12–14*) (Fig 10). One possible reason standing behind this fact is poor perfusion of the wedge preparation resulting in the activation of ATP-dependent potassium current. Indeed APD80 was below 300 ms for *Patients 12* and *13*. Another possible reason is the heterogeneity within the right ventricle that we observed in all RVOT preparations (Fig 10). Since different regions of the ventricle were used for tissue collection and optical mapping, expression profile might not correspond to optically mapped RVOT region. This heterogeneity also explains why models shown in Fig 10 tend to overestimate the APD. In order to exclude possible effect of ATP-dependent shortening of AP, we have manually chosen the recording with the longest AP as GA-input. On the other hand, probable local tissue damage at the site of recording might result in an opposite effect, *i.e.* downregulation of K-channels [46] prolonging the APD.

In the Fig 10 and Table 5 we also demonstrate that minor changes in AP waveform may result in major changes of parameters values. Despite the differences in *Patients 9-11*-based models, in some cases the AP waveforms and restitution were very similar. Consequently, this imposes a limitation on the GA-output model: it has to follow experimental data very closely and perturbations of experimental AP waveform are likely to spoil algorithm performance. In

our opinion, this fact also implies that the modifications we introduced to GA were crucial to the successful gene expression-based prediction of AP waveform.

As was noted previously [47] modern cardiac models coalesce multiple studies performed on different species, using different experimental conditions. Moreover, given the complexity of modern models, it is possible to describe particular dataset using different parameters [8–10] (see also Table 5 and Fig 10). These facts leave us with questions: even if the model describes the particular dataset underlying it, what is the predictive capability of computer simulations? Is it possible to extrapolate model predictions to different clinical or experimental conditions? For example, it is possible that the model description of a particular ionic current is imprecise, but other model components are instead tuned to counterbalance this imprecision. In this case, the model components rescaling would most probably upset the balance and result in model failure to predict AP waveform. In this study, we have shown that it is possible to use computational model to map expression profile to cardiac function and predict actual AP waveform (Figs 9–11). This fact makes a point not only in favor of particular GA-output parameters set, but also indicates the underlying computational model precision.

Our results also indicate that it might be possible to reconstruct the personalized model of *in situ* heart using a similar technique, *i.e.* transcription profile in combination with GA. In some cases (for example, post heart transplant patients) ventricular biopsy could be justified, and tissue samples might be obtained from the patient's heart. In these cases model parameters could be recovered from an *ex vivo* heart experiment via GA, then another *in situ* personalized model could be rescaled using measured mRNA levels. Another possible implication of the provided technique is drug effects investigation. For example, ionic channel blockers often affect several ionic currents. Using GA to measure the effects of a drug provides a cheaper way to recover several ionic currents dose-response curve simultaneously then the patch-clamp technique.

## Limitations

The output conductivities of high-amplitude ionic currents were very precise; however, algorithm performance was much less accurate for important model parameters affecting calcium transients, RyR and SERCA in particular. Using multiparametric optical mapping [48] as input data could probably further increase algorithm precision, however, this question is beyond the scope of the current study.

We have observed a large number of factors that affect GA output values: strict SNR requirements, possible ischemia or heterogeneity of the preparation. Thus, additional measurements are still required to control the GA output parameters precision. Patient 11-model appears to underestimate the sodium current and consequently Patient 11-based models failed to depolarize in half of the cases. Patient 9-based model seem to overestimate APD and, consequently, in half of the cases it was impossible to pace the model at the fastest frequency (Fig 10). This implies the imprecision of Patients 9 and 11 GA-output parameter values.

We used a rough approach for mRNA-based personalized models: ionic channel conductivity was taken proportional to a single gene level of expression. Mostly we used the pore-forming protein for this purpose. More precise approach would require one to account for auxiliary subunits affecting ionic channels voltage-dependence.

## Supporting information

**S1 Fig. Tissue effects.** (A) Comparison of a single cell AP waveform (red line) and AP waveforms recorded from a central cell of a 1D string of cells, tissue size was varied, CV was ≈*27*

*cm/s* in tissue simulations. (B) Comparison of AP waveforms recorded from a central cell of a 100-cells long (*1 cm*) string of cells with variable gap junctions conductivity.
(TIFF)

**S2 Fig. Clusters characteristics: Mean Cluster Error (MCE) and Standard Distance (SDist).** (A) Mean cluster error (distance between the center of each cluster and reference value) and Standard Distance (plotted as a radius of dashed circle, measures the size of a distribution). Cluster mean centers are shown by numbers I (corresponding to 0% of elite organism, red points), II (0.3% of elite organism, blue points), III (3.3% of elite organisms, green points), IV (6.6% of elite organisms, purple points). (B) MCE dependence on generation number for each cluster. Purple and green clusters rapidly shift to the exact solution neighborhood and remain there until the GA termination, while red and blue clusters don't converge to the reference value. (C) SDist dependence on generation number for each cluster. Purple cluster size decreased approximately 8 times after a hundred of generations. Red and blue clusters size decreased 2.6 times after 500 generations.
(TIFF)

**S3 Fig. Polynomial and Cauchy mutations with different distribution parameters.** (A) Best organism RMSE dependence on the γ parameter of the Cauchy distribution on generation 700. (B) Cauchy distribution probability density function dependence on the γ parameter. (C) Best organism RMSE dependence on the η parameter of the polynomial distribution on generation 700. (D) Polynomial distribution probability density function dependence on the η parameter.
(TIFF)

**S4 Fig. CAGE measured mRNA-expression profiles for Patients 1–7.** The mRNA expression level measured in 7 donor hearts. Only genes used for rescaling model parameters are shown. Outliers were determined by IQR method. Colors correspond to APs and restitution curves shown in Fig 9.
(TIFF)

**S5 Fig. $[Na^+]i$ and $[Ca^{2+}]_{nsr}$ distance from steady state values dependence on number of beats per generation.**
(TIFF)

**S6 Fig. Dynamics of $[Na^+]_i$ and $[Ca^{2+}]_{nsr}$ concentration.** (A, B) Best organism intracellular $[Na^+]_i$ and $[Ca^{2+}]_{nsr}$ concentrations averaged over 9 GA runs plotted against generation number. Dashed line in both panels corresponds to input model concentration values. (C, D) Intracellular $[Na^+]_i$ and $[Ca^{2+}]_{nsr}$ concentrations taken from one of the GA runs.
(TIFF)

**S7 Fig. Convergence without crossover.** Best organism parameter values on generation 700 of GA runs with (blue boxes, n = 6) and without (red boxes, n = 6) crossover operator.
(TIFF)

**S8 Fig. RMSE dependence on the number of elite organisms.**
(TIFF)

**S9 Fig. Solution sensitivity to the number of input baselines.** (A-M) Box-and-whiskers plots depict the model parameters sensitivity to the number of input AP baselines. Input AP was simulated at several PCLs listed in the Fig 7A. Dashed line corresponds to the input model parameter value.
(TIFF)

**S10 Fig. Input baselines signal-to-noise ratio.** (A-E) APs waveforms (blue curves) for the different SNR values: 35 dB, 31 dB, 28 dB, 24 dB, 20 dB. Red dashed lines correspond to precise signal with CL = 1000 ms.
(TIFF)

**S11 Fig. Parameters dependence on the SNR.** (A-M) Optimized model parameters distribution depending on the SNR of input APs. Dashed line depicts input model parameter value.
(TIFF)

**S12 Fig. Gaussian noise.** (A) Experimental noise is reproduced the normal distribution with mean = 0.264 mV, and standard deviation = 6.039 mV. (B) Corresponding probability plot: quantiles of experimental noise amplitude distribution (blue) are plotted against quantiles of a theoretical normal distribution (red line).
(TIFF)

**S13 Fig. Heterogeneity of APD for Patient 1 and Patient 2.** Grey lines depict AP waveforms recorded from the wedge preparations. Experimental waveforms are aligned to match the time corresponding to $(dV/dt)_{MAX}$. Red lines correspond to Patient 1 GA-output model (top row) and Patient 2 mRNA-based model (bottom row). The pixels of AP waveform recording that was used as input to GA is marked by the "x" symbol on the top APD map. The pixel of the recording that was used on Fig 9 to compare Patient 2 model with experimental AP is marked by the "+" symbol on the bottom APD maps.
(TIFF)

**S1 Text. Supplemental results.**
(PDF)

## Author Contributions

**Conceptualization:** Roman Syunyaev, Igor R. Efimov.

**Data curation:** Roman Syunyaev, Ruslan Deviatiiarov, Oleg Gusev.

**Formal analysis:** Dmitrii Smirnov, Andrey Pikunov, Roman Syunyaev, Ruslan Deviatiiarov, Anna Gams.

**Funding acquisition:** Roman Syunyaev, Igor R. Efimov.

**Investigation:** Dmitrii Smirnov, Andrey Pikunov, Roman Syunyaev, Ruslan Deviatiiarov, Kedar Aras, Anna Gams, Aaron Koppel.

**Methodology:** Roman Syunyaev.

**Project administration:** Roman Syunyaev, Igor R. Efimov.

**Software:** Dmitrii Smirnov, Andrey Pikunov, Roman Syunyaev.

**Supervision:** Roman Syunyaev, Oleg Gusev, Igor R. Efimov.

**Validation:** Dmitrii Smirnov, Andrey Pikunov, Roman Syunyaev.

**Visualization:** Dmitrii Smirnov, Andrey Pikunov.

**Writing – original draft:** Dmitrii Smirnov, Roman Syunyaev, Ruslan Deviatiiarov, Kedar Aras, Aaron Koppel.

**Writing – review & editing:** Andrey Pikunov, Roman Syunyaev, Anna Gams, Igor R. Efimov.

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
