## [Decision Letter · Decision Letter 0]

26 Sep 2019

PONE-D-19-25321

Genetic algorithm-based personalized models of human cardiac action potential.

PLOS ONE

Dear Dr. Syunyaev,

Thank you for submitting your manuscript to PLOS ONE. After careful consideration, we feel that it has merit but does not fully meet PLOS ONE’s publication criteria as it currently stands. Therefore, we invite you to submit a revised version of the manuscript that addresses the points raised during the review process.

Two reviewers have provided very extensive and constructive feedback, and I encourage you to carefully consider and address their comments. 

To enhance the reproducibility of your results, we recommend that if applicable you deposit your laboratory protocols in protocols.io, where a protocol can be assigned its own identifier (DOI) such that it can be cited independently in the future. For instructions see: http://journals.plos.org/plosone/s/submission-guidelines#loc-laboratory-protocols

We look forward to receiving your revised manuscript.

Kind regards,

B. Rodríguez

Academic Editor

PLOS ONE

Journal Requirements:

Reviewers' comments:

Reviewer's Responses to Questions

**Comments to the Author**

1. Is the manuscript technically sound, and do the data support the conclusions?

Reviewer #1: Partly

Reviewer #2: Yes

2. Has the statistical analysis been performed appropriately and rigorously? 

Reviewer #1: N/A

Reviewer #2: Yes

3. Have the authors made all data underlying the findings in their manuscript fully available?

Reviewer #1: No

Reviewer #2: Yes

4. Is the manuscript presented in an intelligible fashion and written in standard English?

Reviewer #1: Yes

Reviewer #2: No

5. Review Comments to the Author

Reviewer #1: Genetic algorithm-based personalized models of human cardiac action potential

The authors propose and implement a genetic-algorithm approach to estimate maximal conductances (and similar parameters for e.g. J_up) an action potential model, based on optical mapping of voltage in human left-ventricular preparations.

Improvements to previous GA methods are presented, including a restitution based approach, and a novel mutation operator.

A synthetic data study is performed, after which the method is applied to real data, and validated by comparing the obtained parameters to mRNA expression levels.

The work is very interesting, the focus on clinically measurable signals is novel and important, and the data from donor hearts adds further value to this study.

However, the text needs major work.

The manuscript is not structured well, requiring the reader to skip back and forth between sections.

It would benefit from an introduction that clearly states the entire study's objectives (not just the GA improvement) and gives the reader a better idea of what is to come.

Many of the results are stated as fact in the methods (without making clear this is something the study will attempt to prove), which further adds to the confusion.

I believe this manuscript has the potential to be very interesting if the authors rewrite it significantly.

Some more work also needs to be done to show that the chosen strategy of varying part of the initial state in the optimisation still leads to a stable `steady-state'; especially in the light of the reduced number of pre-pacing beats and the interesting observation that there are multiple steady-states, with the exact state reached depending on the initial state).

Major points:

- Results section: It seems the text has been written to have the methods section between the introduction and the results. As it is now at the end of the paper, some rewriting is necessary. I am not sure why the methods are placed at the end, either, this does not seem to be the norm for PLOS ONE in the papers I checked.

- Lines 120-122: Please explain why this strategy, with only 9 stimulations at each PCL, is adequate, and show data to support this claim.

- Lines 122-124: After showing that the O'Hara model has a steady-state that depends on the initial conditions, the authors vary these initial conditions at every iteration of the optimisation, and then run a limited number of repeats. It is not clear to me at all how this would help bring the system into a steady-state faster, or why this should result in a better model. Please add a more detailed explanation, and show how your approach compares to a traditional approach with longer pre-pacing periods. If this is computationally infeasible, at least show the effect of adding more beats (e.g. 5, 9, 18, and 36). I would also be interested to know if the authors consider the existence of multiple "steady states" (technically periodic orbits) as a flaw in the model or if its a realistic feature that gets 'personalised' by the GA.

- The introduction states that modifications to GA are needed, but does not argue this very well. It would be good to mention that this is in fact one of the results shown later in the paper. Similarly, the methods section states "We have found that the original polynomial mutation tends to trap soution in the local minima", but doesn't mention that this will be shown in figure 4. The reasonining for introducing a Cauchy distribution is also not clear to me.

- Lines 172-173: "as shown by Andrews curves in Fig 4C the whole population is in the immediate vicinity of a single solution". It is not clear to me how Fig 4C shows this. Figures 4D and 4E also do not look like an optimiser getting stuck in local minima.

- Lines 181-182: "Moreover, the best organism is unchanged for a number of generations". Please explain how choosing a different distribution to sample new mutations from has caused this effect?

- Lines 181-182: "the best organism is unchanged for a number of generations, allowing it to reach steady state". I'm not sure how much this figure says about the Cauchy distribution being better, and how much it says about 9 beats being too short to reach steady state. Would a Cauchy distribution still be better if you had more beats? Presumably there is some sort of trade-off between the number of beats per simulation and the number of iterations of the GA. Please explain this in greater detail.

- Lines 299-313: This is very nice work and deserves to be mentioned much more in the paper, which is almost entirely about the GA and the syntetic data study.

- Lines 551-552: "Output model parameters ... between two patients.". Please explain this in much greater detail, here in the methods section, but also in the introduction or in the relevant part of the results section.

Minor points

- The authors present a novel adatation of a genetic algorithm, but do not present any arguments why genetic algorithms are suitable optimisers in this case. I would suggest adding either a comparison to other optimisers (e.g. CMA-ES), or at least provide some arguments why other methods are unsuitable and a custom GA is required.

- Line 88-89: Please add more references here, e.g. to the work by Sarkar or the subsequent discussion by the Weiss group.

- Line 137: The term "SBX crossover" has not been introduced.

- Figure 2: The resolution of the provided figure is too low to read the text.

- Figure 2: Several things mentioned in the caption to figure 2 (e.g. mRNA expression), and adapting the patient-1 model for patient 2) have not been mentioned yet. The text would benefit from an introduction of these important concepts in the Introduction section.

- Line 149: "coordinate descent" this term has not been introduced. Please rewrite to do so, or avoid using this term here instead.

- Line 160: "SD dynamics" this term has not been explained at this point in the text.

- Line 166: "As mentioned above", the methods section is no longer above this sentence.

- Line 168: "Although the best organism is ... to the steady state.". This means that running the simulations has now become part of your mutation operator. This should be explained more clearly in the methods.

- Lines 200-201: Please rewrite this. 6-10% seems a fair estimate, but it is not demonstrated all that clearly from the graph (for example, what happens at ~7.5%?).

- Lines 289-291: "The membrane potential differences are mostly confined to depolarization phase and are probably due to photon scattering". Please explain this in more detail, preferably with evidence to back it up.

- Figure 4: How much of the observed differences in convergence are due to the type of distribution, and how much due to the distribution's parameters? What would happen with a narrower Cauchy distribution? Or a wider polynomial one?

- Figure 6b: A central idea in genetic algorithms is that cross-breeding good solutions will yield improved solutions. The red trace in Fig 6B suggests that this is not the case for this problem. Does this imply that the crossover step is not beneficial for this problem? If so, it seems this is more a particle based search method rather than a genetic algorithm per se? It would be good if the authors could experiment with leaving out the crossover, or trying different optimisers on this problem.

- Figure 6: The resolution of the attached figure is too low to make this out clearly.

- Figure 6: These results are very interesting, and show that a GA with a number of elite organisms can act both as a global and a local optimiser. It would be good to discuss this further in the text. At the same time, the very fast convergence of elite organisms to something very close to the true solution suggests that it may be more efficient to perform a two-stage optimisation. Please comment.

- Figure 7: It is not clear from the caption how this comparison was set up. Did you re-implement the method by Bot et al. and apply it to your data?

- Line 391: "Computer simulations". Please provide more detail here. Why were 1D simulations preferred over single cell simulations, and why 1D instead of 2 or 3D? What integration method was used (and with what time step or tolerances?). How was this implemented? How many cells were stimulated, and from which cell or cells was an AP recorded?

- Line 397-399: Please explain this shifting in far greater detail. Which of the two signals was shifted and did this happen once or at every iteration? (In the latter case, how did you stop this method from introducing identifiability problems or local minima?)

- Line 407: This is not a standard deviation but a root-mean-squared error.

- Line 407: The choice of the symbol t^i_max for the number of samples (which is not a time) is confusing.

- Lines 419-420: "we have found odd number of stimulations helpful to exclude possible 1:1 alternans in the output model." This needs a lot more explanation.

- Line 430-431: "This is especially important in terms of intracellular ionic concentrations" Why?

- Line 430: Local minima are a problem _for_ optimisation methods, not _of_ them (they are a property of the score function being optimised).

- Lines 434-438: Please add more detail. What is a "half-width of the half-maximum of the distribution", and why is 0.18 times the original value a good choice?

- Line 441: "We have found that it has adverse effects on algorithm convergence". Please explain where/if this is shown.

- Lines 440-444: Please list the parameters that were varied.

- Lines 441-443: Please explain how applying a mutation to each parameter seperatly differs from your approach. Is the mutation in a parameter i now correlated to the mutation in parameter j somehow?

- Line 453: 1000s? Or 1000*PCL ms?

- Lines 456: The Andrews plots are cool, but don't offer much insight into which parameters vary. Did you try simpler representations, e.g. radar plots, instead?

- Line 466 m x n * p. Please add parentheses and use x or . instead of *.

- Lines 524-525 "Libaries in silico processing performed in Morai system.". Please rewrite this.

- Line 529: "python scripts: level1 and level2". What do these scripts do?

- Line 541: "(100 s)", 100s or a 100 beats?

- Line 558: "below". This is no longer below.

Reviewer #2: 1. Is the manuscript technically sound?

I believe it is. There were multiple moments reading through the paper where I began wondering if the authors had considered something I considered an issue, and then found that they had as I continued reading. I consider this a good sign that the application of the genetic algorithm technique to the problem of personalised cardiac model generation here has been considered in a good amount of depth. The methodology also proves quite successful.

Although there are some points below where I comment on potential issues with the methodology, I do not feel that they compromise the core material of the manuscript (application of a modified GA to achieve action potential model personalisation). The authors should, however, clarify what is going on with the question raised on Line 123.

I cannot comment on the experimental procedures (obtaining imaged voltage data or genetic expression levels).

2. Has the statistical analysis been performed appropriately and rigorously?

I believe so, but it would improve the paper if the results were also presented in a statistical way (i.e. in text, comment on spread of performance across runs, instead of using the worst run as the discussed figure).

3. Have the authors made all data underlying the findings in their manuscript fully available?

I missed where in the paper this was discussed, but am trusting the authors when they state on the online submission that data has been made available. If it is not yet mentioned in the paper, I suggest a very short (1-2 sentence) new section at the end of Methods that points readers to where data can be obtained.

4. Is the manuscript presented in an intelligible fashion and written in standard English?

I feel the work is rather clumsily written. There are many grammatical errors, but those are a separate issue. What I consider more important is that as a reader, I was given very little idea about how the paper sits within the literature. For example, a list of references of other works using the genetic algorithm in the cardiac context is provided, but then only one is further discussed (as the method there serves as the basis for the authors' modified method). Without going through all of those papers, a reader will have no idea whether those works used experimental/clinical data, or simply demonstrated an ability to recover chosen parameter values in the action potential model (i.e. fitting synthetic data). Flicking through the referenced papers, they do seem to work with experimental data, and so this is not a point of novelty.

Instead, a potential point of novelty is the consideration of steady state of the action potential model. The methodology in this paper has been specifically adjusted to recover the steady state as it goes, something which I suspect is novel. So then the questions are, is this indeed novel? If so, the authors should highlight it in the introduction, by pointing out other genetic algorithm applications in cardiac electrophysiology have not considered it (or considered it simply by incurring great computational cost). Additionally, if steady state is so important, the authors should highlight it. Figure 1 shows the effects of choosing different initial conditions, but that is a different issue - if anything, re-using the final state of an organism as the starting point for its offspring seems like it exacerbates this issue (more comments on this below).

Modifications are made to the genetic algorithm to improve its performance, but again it is not at all clear whether these are only novel in the field of cardiac electrophysiology, or novel ideas of the authors in the GA context. For example, using Cauchy distribution jumps is something I've seen in other optimisation (not GA) contexts (e.g. "Levy-flight" jumps in bio-inspired optimisation algorithms), and I assume has also been used in the GA context already. The authors should check up on this if they haven't, and provide an appropriate reference. Similarly, random vector jumps might indeed not be the norm in GA, but I'm sure they must've been used (many optimisation algorithms, including many that use multiple search agents as GA does, will be doing random multivariate Gaussian jumps, for example). Additionally, if these two modifications were so necessary to get good results here, one wonders how some of the other published GA papers in cardiac electrophysiology got away with not using them (or if they did use them, this should definitely be mentioned and emphasis on those aspects of the paper reduced!).

Instead, one major novelty of this work seems to be the success of a GA-fitted model to be used across different members of a population, simply by measuring differences in their relative gene expression levels. Although I don't have a great sense for how easy/hard this is to actually do clinically and so cannot comment too much on the impact level of this finding, this was certainly a unique (and interesting!) aspect of the paper... and one that isn't even mentioned in the introduction! Hence my comment about clumsy writing. As I state many times in this review, I feel this part of the work needs to be highlighted much more.

I also re-iterate that although the English is mostly understandable, there are many grammatical errors and that the manuscript could be improved significantly in this regard. I have highlighted some of the most glaring issues below.

Specific Issues:

Abstract: It feels a bit number-heavy, and the numbers quoted are not average (with standard deviation) error, but maximum error, which I don't think is typical. It also talks about things like "13mV accuracy" which doesn't make sense to me. Is that a maximum deviance of 13mV at any point along the action potential? A root mean square error of the voltage along the whole action potential? This is not clear. In general, I personally think the "story" of the paper could be much better told than just listing error numbers.

Line 50: There's no guarantee the parameters found by GA (with the benefit of multiple pacing frequencies) will be unique, but indeed use of multiple pacing frequencies helps a lot. If this wasn't explicitly considered, I recommend softening this language (similar to how it's expressed on line 97).

Line 98: Again, the intro doesn't mention use of CAGE at all. This seems like a huge mistake!

Line 110: It's nice to point out the issue with a sensitivity to initial condition in cardiac AP models. I believe this is an issue that has been commented in the literature, although I cannot think of a reference off the top of my head. One would be nice here (see also comment on Line 120, though).

Line 114: "Saving organisms state vectors" is not a good figure title, in my opinion. Most of the figure is discussing the behaviour of the action potential model. Speaking of, I feel the caption should mention the model used (O'Hara-Rudy), given the text hasn't yet, and the reader might get curious. I know I did.

Line 120: My issue here is, I don't see how the methodology introduced here actually helps with the issue just discussed, of different initial conditions giving different steady state behaviour. If anything, using the final state of an organism as the initial condition of its offspring seems like it will only cause the method to hone in on a certain behaviour, and fail to explore the range of possibilities represented by different initial conditions. I do not consider this a major failing of the work, as I do not think there are many works at all in cardiac electrophysiology that even consider, let alone deal with this issue. Instead, what I think should happen is that the issue is given less focus in the manuscript (if even brought up), and instead, Fig. 1A-C should highlight the importance of steady state (for example, by plotting an action potential at steady state, and one during the transient phase). Assuming steady state even is important, but one is lead to believe it is in this paper, given the focus on achieving it whilst also avoiding excessive computational cost. This change would make Fig. 1 contribute a lot more to the story of the paper.

Line 123: It was not originally clear to me how allowing intracellular ion concentrations to vary would speed up convergence. I now see that this is about convergence to steady state over the course of the genetic algorithm. Perhaps this could be reworded somehow to be more clear. Furthermore, I don't understand how [Na+]_i and [Ca2+]_nsr can be parameters chosen by the GA, given that they are variables in the model. Do the authors mean the *initial* value is allowed to vary within the GA? Or am I missing something? This needs to be made clear in the paper.

Line 125: "1.5mM precision" sounds not great for an intracellular ion concentration (well, maybe [K+]_i!). This seems to be the worst possible value, which is not a great way to report your error, in my opinion. Mean and standard deviation makes a lot more sense (and will look better!)

Line 132: SD is not defined yet as far as I saw.

Line 141: Patient 1 and patient 2 seem to be switched here (or in the figure). This was also an issue when the CAGE results were discussed later.

Line 154: Again, these error values seem to be worst-case.

Line 178: "Proper value" is mentioned here. Is that the value for synthetic data? I think this needs to be clearer.

Line 181: It's interesting that the best model organism not changing (implying stagnation of the algorithm) is here pointed out as a good thing. Can't argue with the results, though!

Line 215: The point here about "fine tuning" needs to be expressed better. I think I understand it now, but it took me a few reads. The idea is that with a small cluster, and offspring generated from the current population of organisms, the algorithm will only search locally and thus waste less time, right?

Line 304: This is not a clear way to discuss error, as commented on previously.

Discussion: I guess it is up to the editor and journal as to whether this numbered list of highlight points is suitable format for a discussion. It's not my personal preference.

Line 330/343/350: I'm not sure that the modifications to the GA are worth three separate discussion points. In particular, discussion points #3 and #4 should probably be combined and cut down a little, in my opinion.

Line 358: "algorithms" should be "organisms". There are many more grammatical issues, but this one is one that changes the meaning so I have commented on it specifically.

Line 376: Tacking on the CAGE results at the end here with a casual mention is not really doing them justice. Those results seem to me like a much bigger deal than the authors make them sound!

Line 466: This notation was unclear to me. Combining two different multiplication symbols leaves me guessing which is which. If I guessed right, I think you could just say m x np?

6. PLOS authors have the option to publish the peer review history of their article (what does this mean?). If published, this will include your full peer review and any attached files.

Reviewer #1: No

Reviewer #2: No

---

## [Author Response · Author response to Decision Letter 0]

12 Jan 2020

We sincerely thank both reviewers for the time and effort they put into very extensive reviews and insightful critiques to the manuscript. Their critiques served us as a welcome guide during extensive revisions and additional research. We apologize that it took us long time to address their concerns, but it was due to very extensive nature of the critiques. In reply to the reviewers’ questions we ran additional simulations and thoroughly revised the text. Both reviewers mentioned that RNA-based model verification deserves much more prominence in the text. We agree with this suggestion; however, the data that we have presented in the originally submitted manuscript was not sufficient to provide a reasonable discussion of the subject. Now, in reply to reviewers concern we were able to conduct much more extensive additional research of the mRNA-based models rescaling, and we are happy to report that our new significant data provides solid support to our original premise. 

We do hope that the new revision is much more comprehensible and interesting for potential reader then the initial submission. To facilitate review process, in the files attached to the submission and labeled as "Response to Reviewers", we have restated original critiques in green font and provided our response in black font and new manuscript text in blue font. 

Reviewer #1: Genetic algorithm-based personalized models of human cardiac action potential

The authors propose and implement a genetic-algorithm approach to estimate maximal conductances (and similar parameters for e.g. J_up) an action potential model, based on optical mapping of voltage in human left-ventricular preparations. Improvements to previous GA methods are presented, including a restitution based approach, and a novel mutation operator. A synthetic data study is performed, after which the method is applied to real data, and validated by comparing the obtained parameters to mRNA expression levels.

The work is very interesting, the focus on clinically measurable signals is novel and important, and the data from donor hearts adds further value to this study. However, the text needs major work. The manuscript is not structured well, requiring the reader to skip back and forth between sections.

Thank you very much for positive overall assessment of our work. 

It would benefit from an introduction that clearly states the entire study's objectives (not just the GA improvement) and gives the reader a better idea of what is to come. Many of the results are stated as fact in the methods (without making clear this is something the study will attempt to prove), which further adds to the confusion.

We thank the reviewer for bringing up this issue of the manuscript, in the revised version the manuscript introduction is extended, and states aims of the study more clearly. 

I believe this manuscript has the potential to be very interesting if the authors rewrite it significantly.

Some more work also needs to be done to show that the chosen strategy of varying part of the initial state in the optimisation still leads to a stable `steady-state'; especially in the light of the reduced number of pre-pacing beats and the interesting observation that there are multiple steady-states, with the exact state reached depending on the initial state).

Thank you, in reply to this helpful comment we have ran significant additional simulations and paced GA output model for 1000 s to compare steady state with GA output state. The difference is shown in new Fig. 2 and corresponding text. 

Major points:

- Results section: It seems the text has been written to have the methods section between the introduction and the results. As it is now at the end of the paper, some rewriting is necessary. I am not sure why the methods are placed at the end, either, this does not seem to be the norm for PLOS ONE in the papers I checked.

Thank you, as we have stated above the Introduction section is extended, while “Methods” section is placed in the beginning of the article.

- Lines 120-122: Please explain why this strategy, with only 9 stimulations at each PCL, is adequate, and show data to support this claim.

Thank you for this important comment. The strategy is explained in greater detail and a kind of proof that this technique still results in intracellular concentrations close to steady state is depicted in Fig.2 of the new version and corresponding lines 481-498 of the text. New text reads:

“Why would this technique eventually result in concentrations converging to a steady state? The rationale is the following: if given parameters vector results in an acceptable AP waveform (i.e. close to the input AP), but far from steady state, then this solution is going to be discarded eventually, since waveform is going to change after few beats. For example, if AP of particular organism on generation N minimizes RMSE, but corresponds to one of the dotted lines in Fig. 2B, then RMSE is going to be large for this particular organism on generation N+1 (one of the dashed lines). 

 In particular, we suggest including the initial intracellular sodium ([Na+]i) and calcium sarcoplasmic reticulum load ([Ca2+]NSR) at every PCL as model parameters. The [Ca2+]NSR concentration was chosen, because most of the calcium is stored within sarcoplasmic reticulum when the cell is at resting potential. We did not include potassium concentrations in the optimized parameters vector, since 10 mM K+i concentration changes results only in approximately 2% Nernst potential change, and thus a major concentration changes have a minor immediate effect on AP waveform. In order to test if this technique indeed results in a steady state we have run 9 GA simulations. After that, the output models on generation № 700 were paced for 1000 seconds. The observed difference between GA output state and steady state is depicted in Fig. 2E-F. The output [Na+]i was 0.016±0.012 mM from steady state, the [Ca2+]NSR was 0.08±0.07 mM from steady state.”

- Lines 122-124: After showing that the O'Hara model has a steady-state that depends on the initial conditions, the authors vary these initial conditions at every iteration of the optimisation, and then run a limited number of repeats. It is not clear to me at all how this would help bring the system into a steady-state faster, or why this should result in a better model. Please add a more detailed explanation, and show how your approach compares to a traditional approach with longer pre-pacing periods. If this is computationally infeasible, at least show the effect of adding more beats (e.g. 5, 9, 18, and 36). I would also be interested to know if the authors consider the existence of multiple "steady states" (technically periodic orbits) as a flaw in the model or if its a realistic feature that gets 'personalized' by the GA.

Thank you for this question! In part, we provided clarification in the reply to the previous question. Please also, find a clarification on bringing the system closer to steady state, while making “jumps” in slow variables space the lines 473-481 of the text: 

“The final state variables at each PCL are saved and reused as initial state at the next generation (Fig 1A). However, given that the set of parameters minimizing RMSE is itself a function of intracellular concentrations, this approach results in an optimizer solving essentially a new problem every generation. Moreover, as shown above (Fig. 2 B-D) “fixed” initial state for the organisms should result in a “fixed” steady state that might be different from input data. The solution that we propose in this study is to perform a simultaneous search in the parametric and slow variables space, i.e. the initial values of slow variables were added to parameters vector making them susceptible to mutation and crossover operators.”

The traditional approach in some sense, was provided in the Fig 7 of the manuscript, where every organism of the original Bot et. al. algorithm was paced for 50 beats. It is computationally very expensive to provide more than 50 beats. 

We have also ran additional simulations with our version of algorithm to investigate the effects of different number of beats. The results (distance from steady state) are presented on S10 Fig. 

We have added additional discussion of experimental evidence of multiple steady state in lines 778-773 of the text. There is some circumstantial evidence of multiple steady states: hysteresis of APD and alternans was observed by several research groups previously. However, we cannot be sure that this phenomenon occurs in living myocytes. For example, the number of some ionic channels (IKs in particular) might be dynamically adjusted by cell in response to different factors, such as calcium stress. Consequently, there are multiple ways to explain hysteresis phenomena.

New text reads: 

“We have to note here, that several groups of researchers did observe restitution hysteresis as well as alternans hysteresis [34, 35, 36]. Thus, we can hypothesize that multiple steady states is an inherent feature of a myocyte that might be personalized by the GA. On the other hand, living cardiomyocyte is a complicated dynamic system and ionic channel conductivities themselves could change over the time [37].” 

- The introduction states that modifications to GA are needed, but does not argue this very well. It would be good to mention that this is in fact one of the results shown later in the paper. Similarly, the methods section states "We have found that the original polynomial mutation tends to trap soution in the local minima", but doesn't mention that this will be shown in figure 4. The reasonining for introducing a Cauchy distribution is also not clear to me. 

Thank you, we have now mentioned this fact in the introduction. Please, see lines 113-119. New text reads:

“As we demonstrate below in the Final Algorithm subsection of the Results section, this approach results in a poor parameters convergence. We have identified two reasons behind this fact: firstly, intracellular concentrations require much more than 10 seconds to converge to a steady state; secondly, a model with the particular set of parameters may converge to different steady states depending on the initial conditions. In order to address these issues we implemented a modification of GA allowing one to perform parameters optimization and steady state search in the slow variables space simultaneously.” 

Regarding the Cauchy distribution: our initial observation was that modifying the concentrations by polynomial mutation might have the following effect. The whole population prematurely converges to a single non-steady state solution. After that, even though best organism gets closer to steady state every generation (increasing RMSE), some organism from the population mutates back to the previous concentration values. Cauchy distribution reduced the probability of premature convergence and did not reduce the convergence speed at the same time. In reply to reviewers’ questions we have ran additional simulations comparing Cauchy and polynomial distributions. Although, the former on average did result in better state variables, the distance from steady state was not statistically significant on the generation № 700. 

For Cauchy mutation the output [Na+]i on the generation № 700 was 0.016±0.012 mmol from steady state, the [Ca2+]NSR was 0.08±0.07 mmol from steady state.

For Polynomial mutation the output [Na+]i on the generation № 700 was 0.03±0.02 mmol from steady state, the [Ca2+]NSR was 0.10±0.12 mmol from steady state.

Therefore, we have modified the figure (Fig 6 in the new revision) and corresponding text in the lines 493-508 as follows:

“Cauchy distribution is a “pathological” distribution with infinite variance and expected value. As was noted previously [18] Cauchy mutation tends to generate offspring far away from its parent. Consequently, it prohibits algorithm stagnation in local minima and generally results in better convergence for multimodal functions. Fig 6A compares two sample runs of GA with polynomial and Cauchy mutation. In case of polynomial mutation by generation 300 the whole population converged to a vicinity of a single solution that is different from input model set of parameters (red line). We observed slow convergence on subsequent generations simultaneous with slow intracellular concentration changes (compare to S2 Fig plotting corresponding intracellular concentration changes of the best organism). In the case of Cauchy mutation every parameter except INaK converged to some vicinity of input model value by generation 300. At the same time, we did not observe algorithm stagnation: note, for example, the wide variation of IK1 and RyR parameters on generation 500. Consequently, when synthetic AP was used as input data, we observed more robust algorithm convergence in case of Cauchy mutation (Fig. 6B). For example, the error of model parameters was 14±24% vs 17±75% for IKs, 11±25% vs 45 ±23% for Ito, 6±6% vs 13 ±15% for ICaL in GA runs with Cauchy and polynomial mutation correspondingly.” 

- Lines 172-173: "as shown by Andrews curves in Fig 4C the whole population is in the immediate vicinity of a single solution". It is not clear to me how Fig 4C shows this. Figures 4D and 4E also do not look like an optimiser getting stuck in local minima.

- Lines 181-182: "Moreover, the best organism is unchanged for a number of generations". Please explain how choosing a different distribution to sample new mutations from has caused this effect?

- Lines 181-182: "the best organism is unchanged for a number of generations, allowing it to reach steady state". I'm not sure how much this figure says about the Cauchy distribution being better, and how much it says about 9 beats being too short to reach steady state. Would a Cauchy distribution still be better if you had more beats? Presumably there is some sort of trade-off between the number of beats per simulation and the number of iterations of the GA. Please explain this in greater detail.

Thank you very much for expressing these valid concerns. As mentioned in reply to previous question we have revised this part of the text, we hope that the new Fig 6 provides better arguments in favor of Cauchy distribution.

- Lines 299-313: This is very nice work and deserves to be mentioned much more in the paper, which is almost entirely about the GA and the syntetic data study. 

- Lines 551-552: "Output model parameters ... between two patients.". Please explain this in much greater detail, here in the methods section, but also in the introduction or in the relevant part of the results section.

Thank you for a nice comment! We have conducted additional experiments and extended the corresponding sections extensively to address your questions. 

Minor points

- The authors present a novel adatation of a genetic algorithm, but do not present any arguments why genetic algorithms are suitable optimisers in this case. I would suggest adding either a comparison to other optimisers (e.g. CMA-ES), or at least provide some arguments why other methods are unsuitable and a custom GA is required.

We have added additional discussion of the subject in the lines 806-815. Although it would be interesting to compare different optimizer, some custom modification would be required to find the steady-state solution. This kind of benchmarking would be very complicated and we believe it to be beyond the scope of current study. But we have modified the text as follows:

“It is worth noting that all these modifications to GA essentially make the algorithm similar to particle-based algorithms. For example, multivariate jumps are used by Particle Swarm [38], Covariance Matrix Adaptation Evolution Strategy [39] and Cuckoo Flight [40] algorithms. The last one uses “pathological” distribution similar to Cauchy mutation for Levy-flight walk. Although it would be very interesting to compare these algorithms performance with GA modification proposed in this study, some custom modifications to the aforementioned algorithms are still required: as was noted above same AP waveform could be reproduced by different parameter sets [8]. In order to narrow down the output parameters range, not steady state solutions should be somehow automatically discarded by the algorithm. This kind of extensive benchmarking is beyond the scope of the current study.”

- Line 88-89: Please add more references here, e.g. to the work by Sarkar or the subsequent discussion by the Weiss group.

Thank you very much indeed! We have provided more references in lines 94-95 in the revised manuscript. 

- Line 137: The term "SBX crossover" has not been introduced.

Thank you, the abbreviation is introduced in line 81 in the revised manuscript. 

- Figure 2: The resolution of the provided figure is too low to read the text.

Thank you for pointing out this important oversight! We believe that this was an image conversion issue on the PlosOne website, we will make sure to double check the resolution after the new submission is completed.

- Figure 2: Several things mentioned in the caption to figure 2 (e.g. mRNA expression), and adapting the patient-1 model for patient 2) have not been mentioned yet. The text would benefit from an introduction of these important concepts in the Introduction section.

Thank you for pointing out this lack of clarity in our original submission. The revised Introduction section was expanded as mentioned above. New text in the lines 122-133 now reads: 

“Finally, we have verified the algorithm against the experimental optical mapping AP recordings from the human heart. Since we could not measure ionic channel conductivities directly, we used the following assumption instead (Fig. 1B): these conductivities should be proportional to corresponding proteins mRNA level of expression as measured by Cap Analysis of Gene Expression (CAGE) [21] or RNA-seq. Thus, given that GA output model parameters represent actual ionic channels conductivities, the model rescaled in correspondence with mRNA expression profile differences between two patients, would reproduce AP restitution properties of both patients. Moreover, we have to note that this approach (i.e. combining GA with transcription profile) could be regarded as another technique of model personalization. As we show below, GA signal-to-noise ratio (SNR) requirements are rather strict and hard to accomplish in a clinical setting, while mRNA expression profile is possible to measure from tissue biopsy.” 

- Line 149: "coordinate descent" this term has not been introduced. Please rewrite to do so, or avoid using this term here instead. 

Thank you, we now provide a reference to the review paper, which describes coordinate descent algorithms on line 486 of the manuscript.

- Line 160: "SD dynamics" this term has not been explained at this point in the text. 

Thank you, the “RMSE dependence on generation number” is used throughout the new version of the manuscript instead of the “SD dynamics”. 

- Line 166: "As mentioned above", the methods section is no longer above this sentence. 

Thank you, the methods section was shifted to the beginning part of the article.

- Line 168: "Although the best organism is ... to the steady state.". This means that running the simulations has now become part of your mutation operator. This should be explained more clearly in the methods. 

Thank you for pointing out this lack of clarity in our original manuscript! We have now provided a more detailed explanation of elitism strategy in the Methods section. Please, see lines 232-238 of the text: 

“More precisely: elite organisms do participate in mutation and crossover as usual, but an “unspoiled” copy is saved to replace the worst organisms in every generation. However, final state of elite organisms on generation N are still reused as initial state on generation N+1, thus slow variables get closer to steady state, while AP waveform and RMSE changes correspondingly (see “save state variables” above). We have found that using a high number of elite organisms (about 6-10% of the whole population) is optimal for our GA modification.” 

- Lines 200-201: Please rewrite this. 6-10% seems a fair estimate, but it is not demonstrated all that clearly from the graph (for example, what happens at ~7.5%?). 

Because of the random nature of the algorithm, any combination of algorithm parameters might accidentally result in fast or slow convergence for a particular run. We believe that slow convergence of algorithm at 7.5 % is a random fluctuation. We have revised lines 553-558 accordingly, while new S8 Fig demonstrates RMSE dependence on the number of elite organisms on generations 100, 150, 300 and 400 giving better sense of convergence to a reader. 

“We observed slower algorithm convergence, when the number of elite organisms was below 4 % of the whole population. On the other hand, the increase of the proportion of elite organisms above 10 % typically resulted in algorithm stagnation after generation 100. Due to random nature of the algorithm convergence, the convergence speed is susceptible to fluctuations, but optimal proportion of elite organisms could be estimated as 6-10% of the whole population from S8 Fig.” 

- Lines 289-291: "The membrane potential differences are mostly confined to depolarization phase and are probably due to photon scattering". Please explain this in more detail, preferably with evidence to back it up. 

Thank you, more detailed explanation is added to lines 629-639, while (dV/dt)max values are provided to explain more clearly differences in depolarization. New revised text reads: 

“The deviations between input data and the output model are listed in Table 1. While RMSE is close to the noise level, and APD80 error did not exceed 14 ms, the difference in the depolarization phase is very pronounced: (dV/dt)MAX was approximately 20 V/s for input data, while for the output model it ranged from 55 to 80 V/s. This effect might be due to photon scattering in optical mapping recordings [10, 33]. Photons emitted by fluorescent dye undergo multiple scattering events and thus the recorded signal from a given pixel is actually an averaged signal from thousands of myocytes. This effect is known to distort AP waveform during depolarization when differences of membrane potential across the tissue are significant due to propagation of wavefront of excitation. Thus, in order to reduce the effect of this experimental artifact on the model, initial depolarization phase (below -20 mV) was removed from compared AP prior to fitness function calculation (see also Methods section).”

- Figure 4: How much of the observed differences in convergence are due to the type of distribution, and how much due to the distribution's parameters? What would happen with a narrower Cauchy distribution? Or a wider polynomial one?

We hope that new S9 Fig comparing RMSE with different distribution parameters provides the answer to this question. 

- Figure 6b: A central idea in genetic algorithms is that cross-breeding good solutions will yield improved solutions. The red trace in Fig 6B suggests that this is not the case for this problem. Does this imply that the crossover step is not beneficial for this problem? If so, it seems this is more a particle based search method rather than a genetic algorithm per se? It would be good if the authors could experiment with leaving out the crossover, or trying different optimisers on this problem. 

Thank you for this excellent suggestion! We have ran a number of additional simulations without crossover.S7 Fig depicts the results, while corresponding discussion is provided in lines 545-551 of the revised text: 

“The requirement of a high number of elite organisms indicates that interbreeding via crossover operator does not necessarily result in improved results. In order to test whether crossover operator is indeed essential to algorithm convergence we have ran 7 GA runs without the crossover. Indeed, as seen in S7 Fig., leaving out the crossover still results in decent convergence, however some output parameters are much less precise, in particular the error is 100±50% vs 19±23% for IKs, 93±54% vs 32±25% for SERCA, 200±140% vs 26±14% for RyR.” 

- Figure 6: The resolution of the attached figure is too low to make this out clearly.

Thank you, like in the case of Fig. 2 we will double check the resolution after conversion on PlosOne website. 

- Figure 6: These results are very interesting, and show that a GA with a number of elite organisms can act both as a global and a local optimiser. It would be good to discuss this further in the text. At the same time, the very fast convergence of elite organisms to something very close to the true solution suggests that it may be more efficient to perform a two-stage optimisation. Please comment. 

Thank you for the suggestion, this problem is discussed in lines 537-544 and 784-797 of the revised manuscript. 

“In other words, large number of elite organisms result in two-stage optimization: initially the whole parametric space is explored, but eventually a number of elite organisms converge to the same minimum attracting the whole population to its vicinity and resulting in effective local optimization. Although PCA plots did not account for intracellular concentrations (see Methods section) it could be seen from comparison of S2 Fig (Cauchy mutation) and Fig 5 (6.6 % elite) that actually convergence to global parametric minimum (MCE and SDist reduction) is simultaneous with concentration changes to some vicinity of input model steady state value.”

“Albeit premature convergence to local minimum of RMSE hinders the solution of the optimization problem, it is preferable to exploit the vicinity of previously visited points once global minimum is localized. This is accomplished in this study by a large number of elite organisms. When elite organisms fall in some vicinity of RMSE minimum two scenarios are possible. If given solution is far from steady state they are going to be either discarded because of major variation of AP waveform in few generations (since these organisms are not susceptible to modification by mutation). If slow variables are close to steady state, elite organisms start to attract the population to the same vicinity. In the latter case, the whole population is going to exploit the vicinity of the optimization problem solution, thus working as a local optimizer. As seen from the comparison between S2 Fig. C-D and Fig. 5B-C intracellular concentrations and parametric convergence share similar dynamics when ratio of elite organisms is high. On the other hand, it might be more effective to use either classic gradient-based methods or Covariance Matrix Adaptation Evolution Strategy [39] once global minimum is localized (i.e. when a number of elite organisms converged to the same vicinity).”

- Figure 7: It is not clear from the caption how this comparison was set up. Did you re-implement the method by Bot et al. and apply it to your data?

Thank you, we have provided a more detailed description in the lines 561-567 of the revised text:

“We have re-implemented the original algorithm by Bot et.al. using Sastry toolbox [32] with the following modifications to it. Firstly, each organism was paced for 50 stimulations at every PCL, i.e. quasi-steady-state was used because it was computationally very expensive to reach actual steady state. Secondly, the normalized AP waveform at 7 PCLs was used as input data. Finally, the least squares technique was used to renormalize input data prior to fitness function evaluation as described in the Methods section.” 

- Line 391: "Computer simulations". Please provide more detail here. Why were 1D simulations preferred over single cell simulations, and why 1D instead of 2 or 3D? What integration method was used (and with what time step or tolerances?). How was this implemented? How many cells were stimulated, and from which cell or cells was an AP recorded? 

Thank you for this question. We provided detailed explanations on lines 200-206 of the revised text: 

“1D model simulations, while being less computationally expensive than 2D or 3D models, correspond to a plane wave propagating in a 3D tissue at a significant distance from the pacing electrode. Moreover, given that in a wide range of conductivities (S1B Fig) exact AP waveform was essentially independent from gap junctions conductivity, we can conclude that in case of a minor 2D or 3D wavefront curvature, additional perturbations by diffusion operator would not affect AP waveform as well.”

- Line 397-399: Please explain this shifting in far greater detail. Which of the two signals was shifted and did this happen once or at every iteration? (In the latter case, how did you stop this method from introducing identifiability problems or local minima?)

Thank you, a more detailed explanation is provided on lines 178-188. One particular trick that we used is assigning large RMSE values to subthreshold depolarizations. AP waveform rescaling was actually performed for every organism. Please also find a brief discussion on the identifiability problem in lines 102-121 of the new manuscript.

“The algorithm is optimized for optical mapping recordings, where absolute transmembrane potential (TP) values are not known, and thus input data (both synthetic and experimental) was normalized by the algorithm. The least-squares technique is used to find scaling and shift coefficients of AP waveform. For every organism, before fitness function evaluation, input AP was shifted along the time axis to superimpose half-maximum depolarization of compared waveforms, after that experimental AP is rescaled and shifted to minimize the deviation between the experiment and the model: , where and coefficients are determined by the least-squares technique. In order to discard subthreshold depolarizations some large error value was assigned to low-amplitude APs (see below “Fitness function” subsection).”

“To the best of our knowledge, this is a first study providing a technique suitable for optical mapping recordings, where only normalized AP waveform is known, but not the exact transmembrane potential values. Arbitrary rescaling and shift of input AP waveform introduces a new dimension to parameters identifiability problem mentioned above. A possible approach to solve this problem is to utilize so-called restitution property, which is AP dependence on heart rate or pacing cycle length (PCL). For example, the sodium current reduction would result not only in the reduction of the amplitude of AP, but also in change of the steady state intracellular ionic concentrations and consequently, in the restitution curve change. Previously, several publications [2,5] utilized restitution property for GA-based cardiac model optimization. For example, Syed et. al. [2] have used atrial AP waveform at several PCLs as input for GA and paced every organism for 10 seconds before fitness function evaluation.” 

- Line 407: This is not a standard deviation but a root-mean-squared error.

Thank you, SD was changed to RMSE in text and figures.

- Line 407: The choice of the symbol t^i_max for the number of samples (which is not a time) is confusing. 

Thank you, the symbol was changed to N. 

- Lines 419-420: "we have found odd number of stimulations helpful to exclude possible 1:1 alternans in the output model." This needs a lot more explanation. 

Thank you, we have provided additional explanations on lines 223-226: 

“in the case of alternating APs, the waveform (and, consequently, RMSE as well) was different every other generation. Thus, if alternating AP have a low RMSE value on generation N, the RMSE is going to increase on generation N+1.” 

- Line 430-431: "This is especially important in terms of intracellular ionic concentrations" Why? 

As already mentioned above in this reply to reviewers we have substantially revised the section of the article describing the Cauchy mutation. 

- Line 430: Local minima are a problem _for_ optimisation methods, not _of_ them (they are a property of the score function being optimised). 

Thank you, the error was fixed, however these lines were removed in the revised manuscript. 

- Lines 434-438: Please add more detail. What is a "half-width of the half-maximum of the distribution", and why is 0.18 times the original value a good choice? 

The explanation is provided in line 246. “(i.e. PDF is half the maximum value, when )” We have also provided additional S9_Fig demonstrating convergence dependence on γ value. 

- Line 441: "We have found that it has adverse effects on algorithm convergence". Please explain where/if this is shown. 

Thank you a reference to corresponding Results section is provided on line 259 of the revised manuscript: “See also Fig. 3 and the corresponding Results section.” 

- Lines 440-444: Please list the parameters that were varied. 

Thank you. The list of parameters was provided in lines 237-240 of the revised text: 

“Thus, when input data included 4 pacing frequencies the full list of parameters was: gNa, gKr, gK1, gKs, PCaL, gto, gNaK, gNCX, gpCa, Jrel, Jup, CMDN, CaMKII, [Na+]300 ms, [Na+]500 ms, [Na+]1000 ms, [Na+]2000 ms, [Ca2+]300 ms, [Ca2+]500 ms, [Ca2+]1000 ms, [Ca2+]2000 ms.” 

- Lines 441-443: Please explain how applying a mutation to each parameter seperatly differs from your approach. Is the mutation in a parameter i now correlated to the mutation in parameter j somehow? 

Additional explanations were provided on lines 257-259: 

“For example, in the case of the two-parameter problem: if (,) unit vector was chosen, then both parameters are going to be increased by the same amount after mutation. See also Fig. 3 and the corresponding Results section.” 

- Line 453: 1000s? Or 1000*PCL ms? 

1000 seconds. We have changed “s” to “seconds” in the text. 

- Lines 456: The Andrews plots are cool, but don't offer much insight into which parameters vary. Did you try simpler representations, e.g. radar plots, instead?

Thank you for this suggestion. Radar plots are provided in the revised Fig. 6. 

- Line 466 m x n * p. Please add parentheses and use x or . instead of *. 

Thank you, we have changed the formula to mxnp on the line 289. 

- Lines 524-525 "Libaries in silico processing performed in Morai system.". Please rewrite this.

- Line 529: "python scripts: level1 and level2". What do these scripts do?

Thank you: the CAGE data processing was modified on lines 371-372 and 377-380: 

“In silico processing of sequenced CAGE tags was performed by using Moirai system [24].”

“The first script generates CTSS (CAGE tag starting sites) files, where 5' end of the mapped CAGE reads are counted at a single base pair resolution. The second script performs signal clustering on CTSS files with a minimum 10 TPM (tags per million) in at least one sample and minimum distance between clusters of 20 base pairs.” 

- Line 541: "(100 s)", 100s or a 100 beats? 

We have changed it to 100 seconds, see line 401. 

- Line 558: "below". This is no longer below. 

Thank you, the Methods section was shifted to initial part of the article.

We would like to thank Reviewer #1 for most thorough and helpful review, which significantly improved our manuscript!

Reviewer #2: 1. Is the manuscript technically sound? 

I believe it is. There were multiple moments reading through the paper where I began wondering if the authors had considered something I considered an issue, and then found that they had as I continued reading. I consider this a good sign that the application of the genetic algorithm technique to the problem of personalised cardiac model generation here has been considered in a good amount of depth. The methodology also proves quite successful. 

Although there are some points below where I comment on potential issues with the methodology, I do not feel that they compromise the core material of the manuscript (application of a modified GA to achieve action potential model personalisation). The authors should, however, clarify what is going on with the question raised on Line 123. 

I cannot comment on the experimental procedures (obtaining imaged voltage data or genetic expression levels).

Thank you for positive assessment of our work. We have thoroughly revised our manuscript to address your concerns and concerns of other reviewers. 

2. Has the statistical analysis been performed appropriately and rigorously? 

I believe so, but it would improve the paper if the results were also presented in a statistical way (i.e. in text, comment on spread of performance across runs, instead of using the worst run as the discussed figure). 

Thank you! Mean and SD values are now provided in the revised manuscript.

3. Have the authors made all data underlying the findings in their manuscript fully available?

I missed where in the paper this was discussed, but am trusting the authors when they state on the online submission that data has been made available. If it is not yet mentioned in the paper, I suggest a very short (1-2 sentence) new section at the end of Methods that points readers to where data can be obtained.

Yes, we have uploaded the data to datadryad.org. The data is going to be available upon article publication with the following DOI https://doi.org/10.5061/dryad.stqjq2c09 . Meanwhile the data is available to reviewers and editors via the following link https://datadryad.org/stash/share/LZQuhX_jA7dq003OcNniS7B8ngFfIy3J6hEY4644x5o .

4. Is the manuscript presented in an intelligible fashion and written in standard English?

I feel the work is rather clumsily written. There are many grammatical errors, but those are a separate issue. What I consider more important is that as a reader, I was given very little idea about how the paper sits within the literature. For example, a list of references of other works using the genetic algorithm in the cardiac context is provided, but then only one is further discussed (as the method there serves as the basis for the authors' modified method). Without going through all of those papers, a reader will have no idea whether those works used experimental/clinical data, or simply demonstrated an ability to recover chosen parameter values in the action potential model (i.e. fitting synthetic data). Flicking through the referenced papers, they do seem to work with experimental data, and so this is not a point of novelty. 

Thank you for this important critique! We have extended introduction section and hope that new revision gives a better sense of novelty and reviews previous literature in the field. These are the main points of our motivation: 

Firstly, to the best of our knowledge this is a first study discussing GA application to optical mapping recordings of human action potentials, where only a normalized AP waveform is available, but not the exact voltage values. Secondly, in the previous reports which relied on experimental data to unambiguously determine computer model parameters, only single-cell voltage-clamp recordings were utilized. That limits the scope of the GA method application. Most importantly, since it is almost impossible to measure all ionic channels conductivities none of the previous works verified output parameters of GA as applied to experimental data. Please, find new text in the lines 94-133 of the manuscript: 

“Another limitation of optimization algorithms as applied to electrophysiological models is the absence of a unique solution. As was noted previously [8–10] same AP waveforms could be reproduced by computer models with different sets of parameters, in other words, model parameters are unidentifiable from the AP. Also, techniques combining stochastic pacing and complicated voltage-clamp protocols have been recently proposed to overcome the problem [4,8]. However, these techniques are limited to single-cell voltage-clamp recordings, which are not feasible in clinical electrophysiology or whole heart and cardiac tissue measurements. The aim of the current study was to develop a technique that would allow one to find a unique solution using optical mapping, microelectrode, or monophasic AP recordings from cardiac tissue or whole heart. To the best of our knowledge, this is a first study providing a technique suitable for optical mapping recordings, where only normalized AP waveform is known, but not the exact transmembrane potential values. Arbitrary rescaling and shift of input AP waveform introduces a new dimension to parameters identifiability problem mentioned above. A possible approach to solve this problem is to utilize so-called restitution property, which is AP dependence on heart rate or pacing cycle length (PCL). For example, the sodium current reduction would result not only in the reduction of the amplitude of AP, but also in change of the steady state intracellular ionic concentrations and consequently, in the restitution curve change. Previously, several publications [2,5] utilized restitution property for GA-based cardiac model optimization. For example, Syed et. al. [2] have used atrial AP waveform at several PCLs as input for GA and paced every organism for 10 seconds before fitness function evaluation. As we demonstrate below in the Final Algorithm subsection of the Results section, this approach results in a poor parameters convergence. We have identified two reasons behind this fact: firstly, intracellular concentrations require much more than 10 seconds to converge to a steady state; secondly, a model with the particular set of parameters may converge to different steady states depending on the initial conditions. In order to address these issues we implemented a modification of GA allowing one to perform parameters optimization and steady state search in the slow variables space simultaneously. After each short simulation variables are saved, modified by genetic operators and reused as initial states for a new generation.

Finally, we have verified the algorithm against the experimental optical mapping AP recordings from the human heart. Since we could not measure ionic channel conductivities directly, we used the following assumption instead (Fig. 1B): these conductivities should be proportional to corresponding proteins mRNA level of expression as measured by Cap Analysis of Gene Expression (CAGE) [21] or RNA-seq. Thus, given that GA output model parameters represent actual ionic channels conductivities, the model rescaled in correspondence with mRNA expression profile differences between two patients, would reproduce AP restitution properties of both patients. Moreover, we have to note that this approach (i.e. combining GA with transcription profile) could be regarded as another technique of model personalization. As we show below, GA signal-to-noise ratio (SNR) requirements are rather strict and hard to accomplish in a clinical setting, while mRNA expression profile is possible to measure from tissue biopsy.”

Instead, a potential point of novelty is the consideration of steady state of the action potential model. The methodology in this paper has been specifically adjusted to recover the steady state as it goes, something which I suspect is novel. So then the questions are, is this indeed novel? If so, the authors should highlight it in the introduction, by pointing out other genetic algorithm applications in cardiac electrophysiology have not considered it (or considered it simply by incurring great computational cost). Additionally, if steady state is so important, the authors should highlight it. Figure 1 shows the effects of choosing different initial conditions, but that is a different issue - if anything, re-using the final state of an organism as the starting point for its offspring seems like it exacerbates this issue (more comments on this below).

Thank you, and yes – this is novel. We have modified the figure (Fig 2 in the revision) and corresponding text according to your recommendation. See text in lines 456-462 of the revised text:

“Why would this technique eventually result in concentrations converging to a steady state? The rationale is the following: if given parameters vector results in an acceptable AP waveform (i.e. close to the input AP), but far from steady state, then this solution is going to be discarded eventually, since waveform is going to change after few beats. For example, if AP of particular organism on generation N minimizes RMSE, but corresponds to one of the dotted lines in Fig. 2B, then RMSE is going to be large for this particular organism on generation N+1 (one of the dashed lines). “

Modifications are made to the genetic algorithm to improve its performance, but again it is not at all clear whether these are only novel in the field of cardiac electrophysiology, or novel ideas of the authors in the GA context. For example, using Cauchy distribution jumps is something I've seen in other optimisation (not GA) contexts (e.g. "Levy-flight" jumps in bio-inspired optimisation algorithms), and I assume has also been used in the GA context already. The authors should check up on this if they haven't, and provide an appropriate reference. Similarly, random vector jumps might indeed not be the norm in GA, but I'm sure they must've been used (many optimisation algorithms, including many that use multiple search agents as GA does, will be doing random multivariate Gaussian jumps, for example). Additionally, if these two modifications were so necessary to get good results here, one wonders how some of the other published GA papers in cardiac electrophysiology got away with not using them (or if they did use them, this should definitely be mentioned and emphasis on those aspects of the paper reduced!).

Thank you for this important point, which we missed! A brief discussion of the subject is now provided in the Discussion section of the revised manuscript as follows: 

“It is worth noting that all these modifications to GA essentially make the algorithm similar to particle-based algorithms. For example, multivariate jumps are used by Particle Swarm [38], Covariance Matrix Adaptation Evolution Strategy [39] and Cuckoo Flight [40] algorithms. The last one uses “pathological” distribution similar to Cauchy mutation for Levy-flight walk. Although it would be very interesting to compare these algorithms performance with GA modification proposed in this study, some custom modifications to the aforementioned algorithms are still required: as was noted above same AP waveform could be reproduced by different parameter sets [8]. In order to narrow down the output parameters range, not steady state solutions should be somehow automatically discarded by the algorithm. This kind of extensive benchmarking is beyond the scope of the current study.”

Instead, one major novelty of this work seems to be the success of a GA-fitted model to be used across different members of a population, simply by measuring differences in their relative gene expression levels. Although I don't have a great sense for how easy/hard this is to actually do clinically and so cannot comment too much on the impact level of this finding, this was certainly a unique (and interesting!) aspect of the paper... and one that isn't even mentioned in the introduction! Hence my comment about clumsy writing. As I state many times in this review, I feel this part of the work needs to be highlighted much more. 

Thank you very much for this recommendation! We agree completely! The reason we have not emphasized this point earlier was that previous data was not extensive enough to provide well-grounded discussion of the subject. Now during revisions, we were able to collect additional data in address reviewers’ concerns. Please see new Figs 11-12 and corresponding sections of the text. Introduction, Methods, Results and Discussion were all extensively revised to cover this subject. 

I also re-iterate that although the English is mostly understandable, there are many grammatical errors and that the manuscript could be improved significantly in this regard. I have highlighted some of the most glaring issues below. 

Thank you very much for your numerous helpful recommendations that allowed us to improve the paper significantly, both scientifically and stylistically/grammatically. 

Specific Issues:

Abstract: It feels a bit number-heavy, and the numbers quoted are not average (with standard deviation) error, but maximum error, which I don't think is typical. It also talks about things like "13mV accuracy" which doesn't make sense to me. Is that a maximum deviance of 13mV at any point along the action potential? A root mean square error of the voltage along the whole action potential? This is not clear. In general, I personally think the "story" of the paper could be much better told than just listing error numbers.

Thank you, we have modified the abstract accordingly. 

Line 50: There's no guarantee the parameters found by GA (with the benefit of multiple pacing frequencies) will be unique, but indeed use of multiple pacing frequencies helps a lot. If this wasn't explicitly considered, I recommend softening this language (similar to how it's expressed on line 97). 

Thank you, we have modified the text as you recommend. New lines 50-51 read: 

“In order to find the set of model parameters we use steady-state action potential waveform dependence on heart rate, known as restitution property.” 

Line 98: Again, the intro doesn't mention use of CAGE at all. This seems like a huge mistake! 

Thank you, we have provided the discussion of the subject in the lines 122-133 of the revised text: 

“Finally, we have verified the algorithm against the experimental optical mapping AP recordings from the human heart. Since we could not measure ionic channel conductivities directly, we used the following assumption instead (Fig. 1B): these conductivities should be proportional to corresponding proteins mRNA level of expression as measured by Cap Analysis of Gene Expression (CAGE) [21] or RNA-seq. Thus, given that GA output model parameters represent actual ionic channels conductivities, the model rescaled in correspondence with mRNA expression profile differences between two patients, would reproduce AP restitution properties of both patients. Moreover, we have to note that this approach (i.e. combining GA with transcription profile) could be regarded as another technique of model personalization. As we show below, GA signal-to-noise ratio (SNR) requirements are rather strict and hard to accomplish in a clinical setting, while mRNA expression profile is possible to measure from tissue biopsy.” 

Line 110: It's nice to point out the issue with a sensitivity to initial condition in cardiac AP models. I believe this is an issue that has been commented in the literature, although I cannot think of a reference off the top of my head. One would be nice here (see also comment on Line 120, though). 

Thank you, although this issue is rarely considered, we have provided a reference to an important Rudy lab’s paper addressing the subject. Please, see lines 445-446 of the revised text:

“We have to note here that, while sensitivity of cardiac models to initial conditions is rarely considered, is was discussed previously [30].” 

Line 114: "Saving organisms state vectors" is not a good figure title, in my opinion. Most of the figure is discussing the behaviour of the action potential model. Speaking of, I feel the caption should mention the model used (O'Hara-Rudy), given the text hasn't yet, and the reader might get curious. I know I did.

Thank you for noticing this glaring omission on our part! The figure caption was modified accordingly. 

Line 120: My issue here is, I don't see how the methodology introduced here actually helps with the issue just discussed, of different initial conditions giving different steady state behaviour. If anything, using the final state of an organism as the initial condition of its offspring seems like it will only cause the method to hone in on a certain behaviour, and fail to explore the range of possibilities represented by different initial conditions. I do not consider this a major failing of the work, as I do not think there are many works at all in cardiac electrophysiology that even consider, let alone deal with this issue. Instead, what I think should happen is that the issue is given less focus in the manuscript (if even brought up), and instead, Fig. 1A-C should highlight the importance of steady state (for example, by plotting an action potential at steady state, and one during the transient phase). Assuming steady state even is important, but one is lead to believe it is in this paper, given the focus on achieving it whilst also avoiding excessive computational cost. This change would make Fig. 1 contribute a lot more to the story of the paper. 

Thank you! Former Fig. 1 (Fig 2 in the revisions) was modified accordingly. New figure depicts initial and intermediate AP waveform in addition to the steady state. Revised text was also provided in the lines 449-456. Briefly, the offspring initial state is also inherited from the final state of the parent, in between the fitness function evaluations (model runs) the concentrations are modified by mutation and crossover operators, prohibiting certain behavior of the model. Revised text now reads: 

“However, given that the set of parameters minimizing RMSE is itself a function of intracellular concentrations, this approach results in an optimizer solving essentially a new problem every generation. Moreover, as shown above (Fig. 2 B-D) “fixed” initial state for the organisms should result in a “fixed” steady state that might be different from input data. The solution that we propose in this study is to perform a simultaneous search in the parametric and slow variables space, i.e. the initial values of slow variables were added to parameters vector making them susceptible to mutation and crossover operators.” 

Line 123: It was not originally clear to me how allowing intracellular ion concentrations to vary would speed up convergence. I now see that this is about convergence to steady state over the course of the genetic algorithm. Perhaps this could be reworded somehow to be more clear. Furthermore, I don't understand how [Na+]_i and [Ca2+]_nsr can be parameters chosen by the GA, given that they are variables in the model. Do the authors mean the *initial* value is allowed to vary within the GA? Or am I missing something? This needs to be made clear in the paper. 

Thank you, we hope new lines 456-469 discuss the issue much more clearly. Revised text reads: 

“Why would this technique eventually result in concentrations converging to a steady state? The rationale is the following: if given parameters vector results in an acceptable AP waveform (i.e. close to the input AP), but far from steady state, then this solution is going to be discarded eventually, since waveform is going to change after few beats. For example, if AP of particular organism on generation N minimizes RMSE, but corresponds to one of the dotted lines in Fig. 2B, then RMSE is going to be large for this particular organism on generation N+1 (one of the dashed lines). 

 In particular, we suggest including the initial intracellular sodium ([Na+]i) and calcium sarcoplasmic reticulum load ([Ca2+]NSR) at every PCL as model parameters. The [Ca2+]NSR concentration was chosen, because most of the calcium is stored within sarcoplasmic reticulum when the cell is at resting potential. We did not include potassium concentrations in the optimized parameters vector, since 10 mM [K+]i concentration changes results only in approximately 2% Nernst potential change, and thus a major concentration changes have a minor immediate effect on AP waveform.” 

Line 125: "1.5mM precision" sounds not great for an intracellular ion concentration (well, maybe [K+]_i!). This seems to be the worst possible value, which is not a great way to report your error, in my opinion. Mean and standard deviation makes a lot more sense (and will look better!) 

Thank you, we have modified maximum values to mean and standard deviation throughout the text. 

Line 132: SD is not defined yet as far as I saw. 

Thank you, the SD was changed to RMSE in new version of the text, while the methods section moved to initial part of the article. 

Line 141: Patient 1 and patient 2 seem to be switched here (or in the figure). This was also an issue when the CAGE results were discussed later.

Thank you for pointing out this error! The Patient 1\\2 error was fixed in Fig.1 caption as well as 644-646 of the text. 

Line 154: Again, these error values seem to be worst-case.

Thank you, the issue was fixed. Please see lines 506-508 of the manuscript: 

“For example, the error of model parameters was 14±24% vs 17±75% for IKs, 11±25% vs 45 ±23% for Ito, 6±6% vs 13 ±15% for ICaL in GA runs with Cauchy and polynomial mutation correspondingly.” 

Line 178: "Proper value" is mentioned here. Is that the value for synthetic data? I think this needs to be clearer.

Thank you, the issue was fixed. See lines 504-506 of the text.

“Consequently, when synthetic AP was used as input data, we observed more robust algorithm convergence in case of Cauchy mutation (Fig. 6B).” 

Line 181: It's interesting that the best model organism not changing (implying stagnation of the algorithm) is here pointed out as a good thing. Can't argue with the results, though! 

Actually, the stagnation of the algorithm itself was not pointed out as a good thing. Instead, we were drawing attention of the reader to the fact that in this example, it is clear that mutation did not spoil the convergence to the steady state. However, we have ran additional tests and noticed that although Cauchy mutation in general resulted in an output state closer to steady-state, the difference was not statistically significant at the higher numbers of generations. Therefore, Cauchy mutation section was modified substantially in the new version. 

Line 215: The point here about "fine tuning" needs to be expressed better. I think I understand it now, but it took me a few reads. The idea is that with a small cluster, and offspring generated from the current population of organisms, the algorithm will only search locally and thus waste less time, right? 

Yes, this is exactly what we meant; we have modified the text to express it more clearly. See lines lines 524-528 and 537-540 of the Results section, and lines 784-797 of the Discussion section:

“While wide exploration of parametric space is required at the initial stage of algorithm convergence, it is more effective to exploit the global minimum once it was localized (see [31] for discussion on exploration and exploitation). This could be achieved by a large number of elite organisms passing their parameters via crossover operator to siblings after clustering around the same minimum.”

“In other words, large number of elite organisms result in two-stage optimization: initially the whole parametric space is explored, but eventually a number of elite organisms converge to the same minimum attracting the whole population to its vicinity and resulting in effective local optimization.”

“Albeit premature convergence to local minimum of RMSE hinders the solution of the optimization problem, it is preferable to exploit the vicinity of previously visited points once global minimum is localized. This is accomplished in this study by a large number of elite organisms. When elite organisms fall in some vicinity of RMSE minimum two scenarios are possible. If given solution is far from steady state they are going to be either discarded because of major variation of AP waveform in few generations (since these organisms are not susceptible to modification by mutation). If slow variables are close to steady state, elite organisms start to attract the population to the same vicinity. In the latter case, the whole population is going to exploit the vicinity of the optimization problem solution, thus working as a local optimizer. As seen from the comparison between S2 Fig. C-D and Fig. 5B-C intracellular concentrations and parametric convergence share similar dynamics when ratio of elite organisms is high. On the other hand, it might be more effective to use either classic gradient-based methods or Covariance Matrix Adaptation Evolution Strategy [39] once global minimum is localized (i.e. when a number of elite organisms converged to the same vicinity).” 

Line 304: This is not a clear way to discuss error, as commented on previously. 

Thank you, the issue was fixed. 

Discussion: I guess it is up to the editor and journal as to whether this numbered list of highlight points is suitable format for a discussion. It's not my personal preference. 

We have modified the discussion, accordingly. 

Line 330/343/350: I'm not sure that the modifications to the GA are worth three separate discussion points. In particular, discussion points #3 and #4 should probably be combined and cut down a little, in my opinion. 

Thank you. We have removed some parts of the text from discussion, however, after considering reviewers questions, we have decided that some points on algorithm modification have to be discussed more clearly. Thus, the discussion was cut down and extended once again. Please see lines 758-815 of the text. 

Line 358: "algorithms" should be "organisms". There are many more grammatical issues, but this one is one that changes the meaning so I have commented on it specifically. 

This part of the text was removed according to previous critiques. 

Line 376: Tacking on the CAGE results at the end here with a casual mention is not really doing them justice. Those results seem to me like a much bigger deal than the authors make them sound! 

Thank you, as mentioned above, we have performed additional experiments and extended corresponding sections. 

Line 466: This notation was unclear to me. Combining two different multiplication symbols leaves me guessing which is which. If I guessed right, I think you could just say m x np?

Thank you for the suggestion, we have fixed the text accordingly. Please, see line 289 of the new version of the manuscript.

We would like to thank Reviewer #2 for a thorough and friendly review, which served us a welcome guide during revisions. We hope you find it improved.

---

## [Decision Letter · Decision Letter 1]

5 Feb 2020

PONE-D-19-25321R1

Genetic algorithm-based personalized models of human cardiac action potential.

PLOS ONE

Dear Dr. Syunyaev,

Thank you for submitting your manuscript to PLOS ONE. After careful consideration, we feel that it has merit and has improved significantly, but still does not fully meet PLOS ONE’s publication criteria as it currently stands. Therefore, we invite you to submit a revised version of the manuscript that addresses the points raised during the review process.

Please take the opportunity to further improve the manuscript, following the constructive feedback from Reviewer 2. 

We would appreciate receiving your revised manuscript by Mar 21 2020 11:59PM. To enhance the reproducibility of your results, we recommend that if applicable you deposit your laboratory protocols in protocols.io, where a protocol can be assigned its own identifier (DOI) such that it can be cited independently in the future. For instructions see: http://journals.plos.org/plosone/s/submission-guidelines#loc-laboratory-protocols

We look forward to receiving your revised manuscript.

Kind regards,

B. Rodríguez

Academic Editor

PLOS ONE

Reviewers' comments:

Reviewer's Responses to Questions

**Comments to the Author**

1. If the authors have adequately addressed your comments raised in a previous round of review and you feel that this manuscript is now acceptable for publication, you may indicate that here to bypass the “Comments to the Author” section, enter your conflict of interest statement in the “Confidential to Editor” section, and submit your "Accept" recommendation.

Reviewer #1: (No Response)

Reviewer #2: (No Response)

2. Is the manuscript technically sound, and do the data support the conclusions?

Reviewer #1: Yes

Reviewer #2: Yes

3. Has the statistical analysis been performed appropriately and rigorously? 

Reviewer #1: Yes

Reviewer #2: Yes

4. Have the authors made all data underlying the findings in their manuscript fully available?

Reviewer #1: Yes

Reviewer #2: Yes

5. Is the manuscript presented in an intelligible fashion and written in standard English?

Reviewer #1: Yes

Reviewer #2: No

6. Review Comments to the Author

Reviewer #1: I'd like to congratulate the authors on a much improved manuscript and a very interesting study.

A few minor concerns remain, which I am confident the authors can address:

- The caption to figure 2 has not been updated. (B,C) should now be (C,D), (D,E) should become (E,F), and the new B panel should be described.

- The first and last sentence of lines 220-229 appear almost contradictory, please rewrite.

- The statement on line 625-626 needs clarification

Finally, time permitting, the authors might consider adding a comparison of GA-obtained parameters with measured RNA levels (e.g. show if/how they correlate).

But as I didn't mention this in the initial review I'm happy to let this go!

Reviewer #2: Firstly I thank the authors for their very thorough response to my comments, and I believe that they have addressed the points I raised very well. The introduction reads a lot better now, in particular, and the discussion is also much improved. The improvements made to figures really help, also!

However, the paper has now also added a lot more results and I feel this has introduced some new issues into the work. I actually wonder if there was a misunderstanding of my (and the other reviewer's) comments - from what I see both of us asked for more emphasis to be placed on the CAGE results, but this meant (certainly in my case anyway) to give these results a proper discussion, mention them in the introduction, and so on. Instead, the authors have now included many more results further testing out the idea of using CAGE and also RNA-seq data to attempt to generate personalised models. This is great! But still actually feels underemphasised in terms of the text (things like discussion of the implications is a bit lacking). I would like to iterate how impressive this result is! I feel the impact of the paper could be greatly improved if more of a focus was given to those results. That said, PLoS One is about publishing correct science without all the emphasis on impact, and so I am not considering this a negative so much as a comment as to how I think the paper could potentially be further improved.

I have selected a recommendation of Minor Revisions. Although I have perhaps a lot of comments below, I would like to express the fact that I think the paper is much improved and with some cleaning up absolutely fit for publication.

I have some general comments, and then specific comments that I have sorted into important, and less important.

General Comments

* I think the referencing of the supplementary figures is out of order. Not a huge deal, but suboptimal. Additionally, the style of referencing figures is not consistent throughout the paper. This would need to be fixed up in review or proof stage.

* The paper was originally written to have the methods at the back, and this was now changed for this version. However, some of the text in the Results feels like it's there because it's a holdover from the previous revision (or reworked/improved text from the previous version). This has resulted in text in the Results that feels more like justification for choices made in Methods and I think it would improve the paper to move it back into Methods, now that Methods comes before Results.

* The English is on the whole understandable and again I think the writing was improved, things I picked up on last time were given a much better explanation. That said, I still believe there is considerable room to improve the grammar, which would help in making the paper more readable, but also in selling the results and conclusions to the reader. I still consider it the journal's call as to what standard of grammar is acceptable.

For now, I have marked the criteria for "presented in an intelligible fashion and written in standard English" as a No. This is less because of the grammar and more because there are some specific issues (covered in my comments) that I think need to be fixed first. However, I do think the paper is understandable on the whole and I would mark it "Yes" if not for those specific issues.

* My other major comment relates to the arbitrarity of the results that are/are not displayed for the CAGE and RNA-seq model personalisation sections. Some of this might have been me missing something, but why is only Patient 2 compared to data, and not Patients 3-7? Was analysis run using one of the other patients as the baseline (as done for the RNA-seq section?). If so, transparency as to how that went would be good, and if not, this should be stated so that the reader isn't left wondering if Patient 1 was cherrypicked as the best.

Similarly, in the RNA-seq section, why are only patients 8, 9 and 11 used as baselines to create the other models? I appreciate that too many lines in Figure 11 would make it unreadable, but did the authors try using the other patients as baselines also? It would be good to have this at least commented on. Whether a sentence admitting that the three shown were the best three, or that "performance using the other patients as baselines was mixed, and not as good as patients 8 and 9", or whatever the exact case is here. If the analysis wasn't run for those, then say that, so that it doesn't just look like cherrypicking.

* Figure 11 is super low quality in the version I have here. Not sure if it was corrupted as part of the submission process, or if the figure just needs to be created in higher quality for upload. As it stands it's very hard to visually make out some of what it shows (for example, + and x are indistinguishable, and the lines of different colours are hard to tell apart, all can sometimes just appear grey due to image artifacts).

* Table 5 and surrounding discussion need work. See the relevant specific comments below.

Specific Comments (Important)

Line 39: This final sentence could be written in a far more impactful way. It also still mentions two patients, even though far more were used in the current results.

Line 65: It is simply not true that single-cell models "remain non-personalized". Line 71 is reviewing other papers who have used GA to recapture an experimental AP.

Line 91: Weird phrasing. The goal of the study is not to "develop a GA implementation suitable for cardiac models", because those already exist and are referenced. I just think this wording is a little strong and also not very specific to the very real benefits your GA approach brings.

Line 171: (B,C) in caption should be (C,D) I think

Line 172: (D,E) in caption should be (E,F) I think

Line 175: This mentions green-tinted boxes which were in Fig. 1A, not 2A. Fix numbering.

Line 221: Use something like "infeasible" instead of "impossible". Something is not "computationally impossible" just because it takes a long time.

Line 425: I was first left wondering why there were essentially two groups, patients 1-7 and then 8-14. I eventually realised that this was because one group was using CAGE, the other RNA-seq. This could really have been expressed much more cleanly at some point around here.

Line 445: I appreciate adding in a reference discussing the issue of sensitivity to initial conditions as I suggested. However the sentence that does so reads very awkwardly and reads very much like one of those "here because reviewer wanted it" sentences. Why not delete this sentence, and instead just add this reference on to the end of the sentence on line 438-439 where you're actually naturally bringing up this point about initial conditions in the first place? Something like:

"Moreover, given different initial states, intracellular concentrations can converge to different steady state values [30], although this issue is very often neglected in cardiac electrophysiology studies."

Line 462: From what I saw in the figure, moving from dot to dashed line is not "one generation" as the text implies here.

Line 463: "We suggest" isn't great wording. The section later goes on to say things like "we did not include potassium concentrations..." so why not stick to that sort of language? "We included the initial intracellular sodium..."

Line 463: The whole first half of this paragraph feels far more like methods than results. The authors talk about an extra modification made to the GA (intracellular ion concentration initial conditions also allowed to vary as part of the GA), but results without this modification are not shown. If results without were shown, and then commented upon, informing this modification that then led to improved results, I could understand putting this content in the Results section. As it is though, the easiest fix seems to be just to move it into the Methods (which also avoids having to generate any new Results).

Line 504: "Consequently, when synthetic AP was used as input data..." This makes it sound like the authors were using something else previously, then switched to synthetic AP data. I know that's not the case, but I just mean this wording is very awkward. Rephrase as something like:

"We observed more robust algorithm convergence in the case of Cauchy mutation, for model fitting to synthetic AP data (Fig. 6B)"

Line 521: "a" and "spoil" are in the wrong order.

(Already commented above as a general point there, but kept here for posterity.)

Line 666: Why are the other patients, 3-7 only having their APs displayed to indicate the extent and character of the variability, as opposed to patient 2 which features explicit comparison between the data and the CAGE-weighted prediction? This feels very arbitrary, essentially like the results worked well for patient 2 and not for any of the other patients. Honestly including patients 3-7 but not properly including them only makes this whole part feel weaker. Does the concept of adjusting the GA-tuned parameters according to CAGE fail for those patients, or was it just not performed for some reason? Is it that APs were not actually measured, only CAGE data, so this is the best that can be done? If so, make this super clear! If it's in here already and I just missed it (possible!), emphasise it, or otherwise, make sure to mention this! Otherwise, the extra patient data here just fills my mind with more questions, as opposed to strengthening the result.

(Already commented above as a general point there, but kept here for posterity.)

Line 685: Not sure if it's just the version I have on .pdf as a reviewer, but Figure 11 (one of the most important figures!) does not look great. There are serious image artifacts and it's actually a struggle for me to make out the lines plotting the AP curves against the data in some cases. The red and blue lines often blur in with one another and I do not have colourblindness, it is simply a result of the artifacting, plus their thinness. Much of this might just be limited to the version I'm looking at, but please see if this can be improved.

Line 705: In contrast to the comment for line 666 the treatment of patients 8-14 is far more transparent. The successes and failures of the approach are shown clearly, there's no hiding. This makes me feel a lot more comfortable, and the fact that the method works at all (even if not always) is still really nice. Seeing changes in DNA map directly through to changes in parameters that actually do a decent job of predicting the patient's AP is a HUGE point in favour of the underlying AP model doing the right thing (something which is always a little in question because there's many competing models that all give different predictions). "which indirectly indicates GA-output parameters precision for this particular case" is seriously understating what is actually a really cool result! I think this part needs way more/better discussion, including both a better emphasis on the good result,and some more insight into why the authors believe other cases may have failed. I know that some thoughts along these lines are given for why the patient 11-based models weren't great, but another point that could be made is that the models always seem to overestimate the APD when they fail to fit the data super well. If the authors have any thinking as to why, that'd be nice to add. Also, the models based on patient 8 and patient 9 seem very similar for most of the patients, but as pointed out, the parameter values themselves are very different. Worth commenting on that everpresent issue of multiple parameter sets giving similar behaviours, and that actually you've found two sets of parameters that also behave pretty much the same across different pacing frequencies.

Line 738: "The difference in APD80 is below 15ms for CL 1000ms". So what? Why pick out one biomarker, at one pacing frequency, and only talk about it? Is this supposed to be picking out the biggest point of discrepancy between the two curves, because otherwise their agreement is very good? If so, please communicate this (and also feel to use stronger words than "relatively close" on line 736, because although it's hard to make out on the low-quality figure, I'd call them a lot more than "relatively close" if I'm seeing them correctly!)

Line 745: Table 5 was an enigma to me before multiple re-reads. Are these multipliers? Compared to what baseline? The text says that RyR is 1.5-3.5 times higher for the Patient 8 based model for Patient 9 (as compared to the direct GA fit to patient 9 data) and yet the RyR values are 0.690 and 0.825, respectively, so this is not true! The caption for Table 5 really needs more information, and/or the text does. A reword would help make it so much more understable, too.

For example, if I'm indeed understanding it right, something like:

"Table 5: Parameter values (relative to baseline) selected for models for Patient 9 derived via GA, or via comparison of mRNA levels to reference patient"

And then the headings are essentially what you have, but make it clear that those are the reference patient. As it is, it's too easy for me to glance at the table and think the first column is results for Patient 8, and the last column for Patient 10. When it's not, all columns show results for Patient 9, but with models derived in different ways (as the caption tries to say, but I feel my wording just provided achieves this much better).

Line 748: The opening sentence of the discussion is very disagreeable to me. The paper hasn't really introduced a GA-based technique that allows personalization of AP models using their dependence on the pacing rate, despite claiming this. Refs [2,5] in the paper are already credited earlier in the paper as doing that. What this paper *does* do is bring improvements to the methodology. And then the CAGE and RNA-seq stuff, which is really striking. The discussion should tie these results together. "Look, our GA fits using our new improved methodology are so good, that we can even go and scale them based on DNA/RNA readings and go and recapture intersubject variability!" Perhaps that claim is a little strong, as poorer fits from older methods might still achieve much the same and this wasn't explored, but I think that is the direction you should be phrasing things in.

Line 770: "This approach has two disadvantages:" is not the way this should be worded. Why not instead something like "There remained two issues to be addressed:". The two things that are discussed are not so much disadvantages of your approach, as just challenges for any method trying to find steady state quickly insides of a GA (or other optimisation method). So my wording here takes out the unnecessary badmouthing of your method.

Line 869: Presuming I'm interpreting the figure right (low quality makes it hard to see), I think the Patient 9-based model did a much better job than the authors give it credit for, here. Patient 8 did the best, but Patient 9 actually fits well to more patient+pacing rate combinations than it fails to fit to. This is pretty good performance for a parameter tuning that is coming from indirect measurements, and as I mentioned earlier, actually has implications regarding how much confidence to give our AP model predictions (in a good way!). I don't think it should just be lumped in with the Patient 11-based model as "failing to fit".

Specific Comments (Less Important / Positives)

Lines 93-133: This was a big improvement over the previous version. Well written.

Line 168: Figure 2 is much improved!

Line 181: The explanation starting here was a bit hard to understand. Not a major point but I think it could be expressed more clearly.

Line 268: Maybe put the i's and NSR's on these concentrations, though I appreciate it's uglier when you also want to use the subscript to mark the pacing frequency. Not a major issue to me either way.

Line 286: I agree with the other reviewer's first round comments, I think that the principal components are a much nicer way to visualise this!

Line 302: The point is labelled here R, and another later A, but then R and A are not referred to again from what I saw. Labelling them like this just made it more confusing to me, (x0,y0) and (xc,yc) would be fine.

Line 316: Figure 5 could probably be moved to the supplement, in my opinion. The point that large numbers of elite particles performs better is already made in Figure 4.

Line 456: This is a nice explanation of why the method presented in the paper could be expected to work. I feel like it fits better where the method is presented, instead of in results, but this isn't major.

Line 584: The section "Input data requirements" talks first about the amount of restitution information required for different parameters to be correctly identified, and then the requirements in terms of noise. I would actually just break this up into two separate sections, with more informative headings on each. "Input data requirements" doesn't immediately make me think of restitution/requirements on number of pacing frequencies. Also, it seems to me like other target values for the parameters could likely change how many different frequencies are required to identify those different parameters. So talking about this like a requirement on the input data has been identified feels a little wrong.

Line 604: On the other hand, while it might be true that different APs also showed a different sensitivity to noise, I'm far more comfortable with this section being expressed in terms of the "required signal-to-noise ratio". I also appreciate its inclusion, it was interesting and I think it's valuable.

7. PLOS authors have the option to publish the peer review history of their article (what does this mean?). If published, this will include your full peer review and any attached files.

Reviewer #1: Yes: Michael Clerx

Reviewer #2: No

---

## [Author Response · Author response to Decision Letter 1]

20 Mar 2020

For the convenience of reviewers the text below is attached as a separate file.

We are grateful to the editor and reviewers for the opportunity to revise and resubmit our manuscript. In order to expedite the review process we restated reviewers’ questions in green font and our provided our answers in blue font. 

Reviewer #1: I'd like to congratulate the authors on a much improved manuscript and a very interesting study.

We thank the Reviewer very much for the positive assessment of the revised manuscript. 

A few minor concerns remain, which I am confident the authors can address:

- The caption to figure 2 has not been updated. (B,C) should now be (C,D), (D,E) should become (E,F), and the new B panel should be described.

Thank you, we have fixed the error in figure caption.

- The first and last sentence of lines 218-228 appear almost contradictory, please rewrite.

Thank you, we believe that we have expressed the idea more clearly in the new revision, new text reads as follows:

“Computational cost of pacing every organism at every generation until reaching steady state during a GA run is prohibitively high. Thus, we saved each AP after a short simulation… As a result, each organism approaches closer to steady state variables with every generation.”

- The statement on line 634-636 needs clarification

Thank you, we have included a sentence explaining the statement:

“One possible explanation of this heterogeneity is uneven perfusion of the sample resulting in a mild ischemia that shortens AP duration because of activation of the ATP-dependent potassium channels IK,ATP.”

Finally, time permitting, the authors might consider adding a comparison of GA-obtained parameters with measured RNA levels (e.g. show if/how they correlate).

But as I didn't mention this in the initial review I'm happy to let this go!

Indeed, we have considered the correlation between output parameters and mRNA levels. However, given the limited amount of experimental data, we concluded that correlation is not too meaningful. As we mentioned several times throughout the text, we believe that at least in some cases experimental tissue was exposed to ischemic conditions. We also demonstrate in the article that Patients 9 and 11 models performance are not perfect: the former underestimated the sodium current, while the latter overestimated the APD. Moreover, as we demonstrate in Fig. 6, the precision of some parameters, such as low-amplitude ionic currents, was relatively imprecise. We thus concluded that, given the limited amount of data, model rescaling is the best way to verify the algorithm precision. For example, low-amplitude currents would spoil the correlation, but are less likely to affect AP waveform after the parameters rescaling. However, we will address this important issue in the future studies. 

Reviewer #2: Firstly I thank the authors for their very thorough response to my comments, and I believe that they have addressed the points I raised very well. The introduction reads a lot better now, in particular, and the discussion is also much improved. The improvements made to figures really help, also!

However, the paper has now also added a lot more results and I feel this has introduced some new issues into the work. I actually wonder if there was a misunderstanding of my (and the other reviewer's) comments - from what I see both of us asked for more emphasis to be placed on the CAGE results, but this meant (certainly in my case anyway) to give these results a proper discussion, mention them in the introduction, and so on. Instead, the authors have now included many more results further testing out the idea of using CAGE and also RNA-seq data to attempt to generate personalised models. This is great! But still actually feels underemphasised in terms of the text (things like discussion of the implications is a bit lacking). I would like to iterate how impressive this result is! I feel the impact of the paper could be greatly improved if more of a focus was given to those results. That said, PLoS One is about publishing correct science without all the emphasis on impact, and so I am not considering this a negative so much as a comment as to how I think the paper could potentially be further improved.

I have selected a recommendation of Minor Revisions. Although I have perhaps a lot of comments below, I would like to express the fact that I think the paper is much improved and with some cleaning up absolutely fit for publication.

We thank the reviewer for this positive evaluation of the new revision and new valuable comments and recommendations. We have carefully considered the reviewer’s recommendations and modified the manuscript accordingly.

I have some general comments, and then specific comments that I have sorted into important, and less important.

General Comments

* I think the referencing of the supplementary figures is out of order. Not a huge deal, but suboptimal. Additionally, the style of referencing figures is not consistent throughout the paper. This would need to be fixed up in review or proof stage.

Thank you, we have fixed the order and formatting of the references.

* The paper was originally written to have the methods at the back, and this was now changed for this version. However, some of the text in the Results feels like it's there because it's a holdover from the previous revision (or reworked/improved text from the previous version). This has resulted in text in the Results that feels more like justification for choices made in Methods and I think it would improve the paper to move it back into Methods, now that Methods comes before Results.

Thank you, we have moved some portions of the text to methods, in particular the description of the slow variables mutations was shifted to the “Methods” section as recommended by the reviewer. While we agree that on many occasions “Results” section repeats some of the information already mentioned in the “Methods” section, we have to note, that half of the figures is indeed the validation of the GA modifications. Although, mRNA-related study is probably much more impressive, we believe that these modifications were crucial to the successful model rescaling, and thus this validation was necessary. In our opinion, in most of the cases repeating some information from the “Methods” was necessary to spare the reader from flipping pages back and forth.

* The English is on the whole understandable and again I think the writing was improved, things I picked up on last time were given a much better explanation. That said, I still believe there is considerable room to improve the grammar, which would help in making the paper more readable, but also in selling the results and conclusions to the reader. I still consider it the journal's call as to what standard of grammar is acceptable.

For now, I have marked the criteria for "presented in an intelligible fashion and written in standard English" as a No. This is less because of the grammar and more because there are some specific issues (covered in my comments) that I think need to be fixed first. However, I do think the paper is understandable on the whole and I would mark it "Yes" if not for those specific issues.

* My other major comment relates to the arbitrarity of the results that are/are not displayed for the CAGE and RNA-seq model personalisation sections. Some of this might have been me missing something, but why is only Patient 2 compared to data, and not Patients 3-7? Was analysis run using one of the other patients as the baseline (as done for the RNA-seq section?). If so, transparency as to how that went would be good, and if not, this should be stated so that the reader isn't left wondering if Patient 1 was cherrypicked as the best.

Similarly, in the RNA-seq section, why are only patients 8, 9 and 11 used as baselines to create the other models? I appreciate that too many lines in Figure 11 would make it unreadable, but did the authors try using the other patients as baselines also? It would be good to have this at least commented on. Whether a sentence admitting that the three shown were the best three, or that "performance using the other patients as baselines was mixed, and not as good as patients 8 and 9", or whatever the exact case is here. If the analysis wasn't run for those, then say that, so that it doesn't just look like cherrypicking.

We thank the reviewer for pointing this omission on our part; we have revised the Methods section to avoid misinterpretation of the actual results, new text in the lines 411-413 and 421-427 reads:

“In the case of CAGE group of patients, functional data was not available for Patients 3-7, thus only Patients 1 and 2 were used for algorithm verification.”

“Due to computational limitations, after preliminary analysis several patients available were excluded and not used as input to GA. In particular, Patient 10 was excluded because of a very long depolarization time (see “Experimental data” subsection of the “Results” section for a brief discussion of optical mapping artefacts affecting depolarization phase). Patients 12 and 13 were excluded because of the very short APD (below 300 ms, which indicated ischemia of the preparation). Patients 12 and 14 were excluded due to low signal-to-noise ratio, while at high frequencies alternans was also observed for Patient 14.”

* Figure 11 is super low quality in the version I have here. Not sure if it was corrupted as part of the submission process, or if the figure just needs to be created in higher quality for upload. As it stands it's very hard to visually make out some of what it shows (for example, + and x are indistinguishable, and the lines of different colours are hard to tell apart, all can sometimes just appear grey due to image artifacts).

We are sorry, but actually, we believe the Figure 11 (Figure 10 in the new revision) resolution to be very high. Although we cannot control the conversion from TIFF to the PDF on the PLOS One site, we suggest downloading the high definition figure via the link available from the PDF above the figure. However, we are not sure if the link works for reviewers.

* Table 5 and surrounding discussion need work. See the relevant specific comments below.

Thank you we have modified the table and the text according to suggestion.

Specific Comments (Important)

Line 39: This final sentence could be written in a far more impactful way. It also still mentions two patients, even though far more were used in the current results.

We have modified the abstract according to suggestion, new version reads:

“We have demonstrated that mRNA-based models predict the AP waveform restitution with high precision. The latter also provides a novel technique of model personalization that makes it possible to map gene expression profile to cardiac function.”

Line 65: It is simply not true that single-cell models "remain non-personalized". Line 71 is reviewing other papers who have used GA to recapture an experimental AP.

Thank you for pointing out this inaccuracy, we have removed the sentence from the text.

Line 91: Weird phrasing. The goal of the study is not to "develop a GA implementation suitable for cardiac models", because those already exist and are referenced. I just think this wording is a little strong and also not very specific to the very real benefits your GA approach brings.

Thank you we have changed the text in the lines 89-90:

“One of the goals of the current study was to develop robust GA implementation making it possible to find the set of cardiac electrophysiology model parameters without premature convergence to sub-optimal solution.”

Line 171: (B,C) in caption should be (C,D) I think

Line 172: (D,E) in caption should be (E,F) I think

Thank you! The error in the figure caption was corrected.

Line 175: This mentions green-tinted boxes which were in Fig. 1A, not 2A. Fix numbering.

Thank you, the numbering was fixed.

Line 221: Use something like "infeasible" instead of "impossible". Something is not "computationally impossible" just because it takes a long time.

The text was fixed according to suggestion. Please, see line 216 in the new revision.

Line 425: I was first left wondering why there were essentially two groups, patients 1-7 and then 8-14. I eventually realised that this was because one group was using CAGE, the other RNA-seq. This could really have been expressed much more cleanly at some point around here.

Thank you, the clarification was added to the methods section, lines 402-404:

“Genome-wide transcription profile was measured via CAGE (for Patients 1-7) or RNA-Seq (for Patients 8-14) techniques as described above.”

Line 445: I appreciate adding in a reference discussing the issue of sensitivity to initial conditions as I suggested. However the sentence that does so reads very awkwardly and reads very much like one of those "here because reviewer wanted it" sentences. Why not delete this sentence, and instead just add this reference on to the end of the sentence on line 438-439 where you're actually naturally bringing up this point about initial conditions in the first place? Something like:

"Moreover, given different initial states, intracellular concentrations can converge to different steady state values [30], although this issue is very often neglected in cardiac electrophysiology studies."

Thank you, we have changed the text according to your recommendation.

Line 462: From what I saw in the figure, moving from dot to dashed line is not "one generation" as the text implies here.

Actually, since 9 stimuli were used for every generation, moving from dotted to dashed line is a single generation indeed. We have added a brief explanation that we hope will clarify this issue (line 460): 

“For example, if AP of particular organism on generation N minimizes RMSE, but corresponds to one of the dotted lines in Fig. 2B, then RMSE is going to be large for this particular organism on generation N+1 (i.e. after 9 beats, one of the dashed lines).”

The text above (lines 445-446) reads

“Instead of the direct approach, we have evaluated the fitness function after a short run (9 stimulations at each PCL).” 

Line 463: "We suggest" isn't great wording. The section later goes on to say things like "we did not include potassium concentrations..." so why not stick to that sort of language? "We included the initial intracellular sodium..."

The corresponding text was removed from the new version of the manuscript. Thank you. 

Line 463: The whole first half of this paragraph feels far more like methods than results. The authors talk about an extra modification made to the GA (intracellular ion concentration initial conditions also allowed to vary as part of the GA), but results without this modification are not shown. If results without were shown, and then commented upon, informing this modification that then led to improved results, I could understand putting this content in the Results section. As it is though, the easiest fix seems to be just to move it into the Methods (which also avoids having to generate any new Results).

Thank you, the text was moved to the lines 258-263.

Line 504: "Consequently, when synthetic AP was used as input data..." This makes it sound like the authors were using something else previously, then switched to synthetic AP data. I know that's not the case, but I just mean this wording is very awkward. Rephrase as something like:

"We observed more robust algorithm convergence in the case of Cauchy mutation, for model fitting to synthetic AP data (Fig. 6B)"

Thank you, the text was modified according to the recommendations. Please see lines 496-497 in the new revision. 

Line 521: "a" and "spoil" are in the wrong order.

Thank you, the error was fixed.

(Already commented above as a general point there, but kept here for posterity.)

Line 666: Why are the other patients, 3-7 only having their APs displayed to indicate the extent and character of the variability, as opposed to patient 2 which features explicit comparison between the data and the CAGE-weighted prediction? This feels very arbitrary, essentially like the results worked well for patient 2 and not for any of the other patients. Honestly including patients 3-7 but not properly including them only makes this whole part feel weaker. Does the concept of adjusting the GA-tuned parameters according to CAGE fail for those patients, or was it just not performed for some reason? Is it that APs were not actually measured, only CAGE data, so this is the best that can be done? If so, make this super clear! If it's in here already and I just missed it (possible!), emphasise it, or otherwise, make sure to mention this! Otherwise, the extra patient data here just fills my mind with more questions, as opposed to strengthening the result.

We thank the reviewer for pointing this out, as mentioned above, the Methods section was changed accordingly to clarify this issue. Moreover, short explanation was added to the lines 664-666: 

“As noted above, functional data was not available for these particular patients, but the variability between the models is within physiological range [11].”

(Already commented above as a general point there, but kept here for posterity.)

Line 685: Not sure if it's just the version I have on .pdf as a reviewer, but Figure 11 (one of the most important figures!) does not look great. There are serious image artifacts and it's actually a struggle for me to make out the lines plotting the AP curves against the data in some cases. The red and blue lines often blur in with one another and I do not have colourblindness, it is simply a result of the artifacting, plus their thinness. Much of this might just be limited to the version I'm looking at, but please see if this can be improved.

As we answered above, we think that the only solution of the problem is downloading high resolution figures using the link, but we are not sure if the link from PDF document works for reviewers. We have double-checked that the resolution of the figure is high and the high resolution images are available for download to us. 

Line 705: In contrast to the comment for line 666 the treatment of patients 8-14 is far more transparent. The successes and failures of the approach are shown clearly, there's no hiding. This makes me feel a lot more comfortable, and the fact that the method works at all (even if not always) is still really nice. Seeing changes in DNA map directly through to changes in parameters that actually do a decent job of predicting the patient's AP is a HUGE point in favour of the underlying AP model doing the right thing (something which is always a little in question because there's many competing models that all give different predictions). "which indirectly indicates GA-output parameters precision for this particular case" is seriously understating what is actually a really cool result! I think this part needs way more/better discussion, including both a better emphasis on the good result,and some more insight into why the authors believe other cases may have failed. I know that some thoughts along these lines are given for why the patient 11-based models weren't great, but another point that could be made is that the models always seem to overestimate the APD when they fail to fit the data super well. If the authors have any thinking as to why, that'd be nice to add. Also, the models based on patient 8 and patient 9 seem very similar for most of the patients, but as pointed out, the parameter values themselves are very different. Worth commenting on that everpresent issue of multiple parameter sets giving similar behaviours, and that actually you've found two sets of parameters that also behave pretty much the same across different pacing frequencies.

We thank the reviewer for positive evaluation of our results. We do hope that the new discussion is more comprehensive and impactful. Several clarifications were added to the Results section, but, most importantly, new lines 865-889 read:

“This heterogeneity also explains why models shown in Fig. 10 tend to overestimate the APD. In order to exclude possible effect of ATP-dependent shortening of AP, we have manually chosen the recording with the longest AP as GA-input. On the other hand, probable local tissue damage at the site of recording might result in an opposite effect, i.e. downregulation of K-channels [48] prolonging the APD.

 In the Fig. 10 and Table 5 we also demonstrate that minor changes in AP waveform may result in major changes of parameters values. Despite the differences in Patients 9-11-based models, in some cases the AP waveforms and restitution were very similar. Consequently, this imposes a limitation on the GA-output model: it has to follow experimental data very closely and perturbations of experimental AP waveform are likely to spoil algorithm performance. In our opinion, this fact also implies that the modifications we introduced to GA were crucial to the successful gene expression-based prediction of AP waveform. 

 As was noted previously [49] modern cardiac models coalesce multiple studies performed on different species, using different experimental conditions. Moreover, given the complexity of modern models, it is possible to describe particular dataset using different parameters [8-10] (see also Table 5 and Fig. 10). These facts leave us with questions: even if the model describes the particular dataset underlying it, what is the predictive capability of computer simulations? Is it possible to extrapolate model predictions to different clinical or experimental conditions? For example, it is possible that the model description of a particular ionic current is imprecise, but other model components are instead tuned to counterbalance this imprecision. In this case, the model components rescaling would most probably upset the balance and result in model failure to predict AP waveform. In this study, we have shown that it is possible to use computational model to map expression profile to cardiac function and predict actual AP waveform (Figs 9-11). This fact makes a point not only in favor of particular GA-output parameters set, but also indicates the underlying computational model precision. ”

Lines 895-899:

“Another possible implication of the provided technique is drug effects investigation. For example, ionic channel blockers often affect several ionic currents. Using GA to measure the effects of a drug provides a cheaper way to recover several ionic currents dose-response curve simultaneously then the patch-clamp technique.”

Lines 747-752 in the result section:

“It is also interesting that despite major differences in parameters, models behavior was surprisingly similar not only in the case of Patient 9, but also in a number of other cases. At least partially, this could be explained by similarities in expression profiles: for example, differences between Patient 11 and Patient 13 genes expression corresponding to IKr, IKs, Ito, ICaL, RyR and CaMKII were less than 9 % (see Table C in S1 text). ”

Lines 908-912 in the Limitations section:

“Patient 11-model appears to underestimate the sodium current and consequently Patient 11-based models failed to depolarize in half of the cases. Patient 9-based model seem to overestimate APD and, consequently, in half of the cases it was impossible to pace the model at the fastest frequency (Fig. 10). This implies the imprecision of Patients 9 and 11 GA-output parameter values.”

Line 738: "The difference in APD80 is below 15ms for CL 1000ms". So what? Why pick out one biomarker, at one pacing frequency, and only talk about it? Is this supposed to be picking out the biggest point of discrepancy between the two curves, because otherwise their agreement is very good? If so, please communicate this (and also feel to use stronger words than "relatively close" on line 736, because although it's hard to make out on the low-quality figure, I'd call them a lot more than "relatively close" if I'm seeing them correctly!)

Thank you, we have removed the sentence and instead inserted a portion of text that, we believe, communicates the idea better:

“Surprisingly, in this case, the GA-output model actually performed worse than the mRNA-based model, the difference is most prominent at 250 ms PCL: in the former case (Patient 9-based model) APD80 error was 26 ms, in the latter (Patient 8-based model) APD80 error was 3 ms. On the other hand, this difference resulted only in slight RMSE250ms difference: 5.7 vs 5.0 mV correspondingly. This implies high sensitivity of GA to AP perturbations that was shown above by noise sensitivity analysis: GA-output model should follow experimental AP waveform closely and minor experimental artefacts might cause major change in parameter values”

Line 745: Table 5 was an enigma to me before multiple re-reads. Are these multipliers? Compared to what baseline? The text says that RyR is 1.5-3.5 times higher for the Patient 8 based model for Patient 9 (as compared to the direct GA fit to patient 9 data) and yet the RyR values are 0.690 and 0.825, respectively, so this is not true! The caption for Table 5 really needs more information, and/or the text does. A reword would help make it so much more understable, too.

For example, if I'm indeed understanding it right, something like: "Table 5: Parameter values (relative to baseline) selected for models for Patient 9 derived via GA, or via comparison of mRNA levels to reference patient". And then the headings are essentially what you have, but make it clear that those are the reference patient. As it is, it's too easy for me to glance at the table and think the first column is results for Patient 8, and the last column for Patient 10. When it's not, all columns show results for Patient 9, but with models derived in different ways (as the caption tries to say, but I feel my wording just provided achieves this much better).

We thank the reviewer for pointing the issue out; the Table 5 was very hard to understand indeed. Regarding the RyR, we apologize, because it is an error on our side. Explanation in the text was modified. Lines 728-730: “Table 5 lists the parameter values for three variants of Patient 9 model: GA-output model, and two transcription profile-based models.”

Table 5 caption:

“Table 5: Parameter values (relative to baseline O’Hara-Rudy model [15]) of Patient 9 models derived via GA or via comparison of mRNA levels to reference patient.”

The headings also mention “Reference patient” now. 

Line 748: The opening sentence of the discussion is very disagreeable to me. The paper hasn't really introduced a GA-based technique that allows personalization of AP models using their dependence on the pacing rate, despite claiming this. Refs [2,5] in the paper are already credited earlier in the paper as doing that. What this paper *does* do is bring improvements to the methodology. And then the CAGE and RNA-seq stuff, which is really striking. The discussion should tie these results together. "Look, our GA fits using our new improved methodology are so good, that we can even go and scale them based on DNA/RNA readings and go and recapture intersubject variability!" Perhaps that claim is a little strong, as poorer fits from older methods might still achieve much the same and this wasn't explored, but I think that is the direction you should be phrasing things in.

Thank you. We agree. We have revised the opening sentence and added a few sentences here:

“In this study, firstly, we have introduced a novel GA modification allowing one to personalize cardiac electrophysiology models using steady-state AP recordings dependence on PCL as input data. The algorithm modification is based on the idea that parametric optimization and slow variables steady state search should be performed simultaneously for effective convergence. Secondly, we have tested the modifications we introduced on synthetic action potentials proving our modifications to be advantageous for overall algorithm performance. Finally, we have tested the algorithm performance against cardiac optical mapping experimental data and mRNA expression profile. The output parameters precision was confirmed by the observation that mRNA-based models predict patients AP waveform and restitution. Essentially, a combination of GA with the mRNA expression measurements provides a novel technique for model personalization.”

Line 770: "This approach has two disadvantages:" is not the way this should be worded. Why not instead something like "There remained two issues to be addressed:". The two things that are discussed are not so much disadvantages of your approach, as just challenges for any method trying to find steady state quickly insides of a GA (or other optimization method). So my wording here takes out the unnecessary badmouthing of your method.

Thank you for pointing out this unclear text in our manuscript. Actually, we intended, unsuccessfully, to discuss and justify the mutation of slow variables. We revised the text to express it hopefully more clearly. Please see lines 782-788:

“Without further modifications this approach has two limitations:… In order to address these issues, two slow variables: intracellular sodium concentration and sarcoplasmic reticulum calcium load are treated as model parameters by the algorithm”. 

Line 869: Presuming I'm interpreting the figure right (low quality makes it hard to see), I think the Patient 9-based model did a much better job than the authors give it credit for, here. Patient 8 did the best, but Patient 9 actually fits well to more patient+pacing rate combinations than it fails to fit to. This is pretty good performance for a parameter tuning that is coming from indirect measurements, and as I mentioned earlier, actually has implications regarding how much confidence to give our AP model predictions (in a good way!). I don't think it should just be lumped in with the Patient 11-based model as "failing to fit".

We thank the reviewer for this suggestion. However, we do believe Patient 9 and Patient 11 models to be relatively imprecise. The text was modified to communicate this message:

“Patient 11-model appears to underestimate the sodium current and consequently Patient 11-based models failed to depolarize in half of the cases. Patient 9-based model seem to overestimate APD and, consequently, in half of the cases it was impossible to pace the model at the fastest frequency (Fig. 10). This implies the imprecision of Patients 9 and 11 GA-output parameter values.”

Specific Comments (Less Important / Positives)

Lines 93-133: This was a big improvement over the previous version. Well written.

Line 168: Figure 2 is much improved!

Thank you!

Line 181: The explanation starting here was a bit hard to understand. Not a major point but I think it could be expressed more clearly.

We hope that new text is clearer:

“The algorithm is optimized for optical AP recordings, where absolute transmembrane potential (TP) values are not known, and thus input data (both synthetic and experimental) is renormalized by the algorithm prior to every fitness function calculation. The following technique was used for renormalization. Firstly, input AP waveform is shifted along the time axis in order to superimpose compared waveforms; in particular, half-maximum depolarization of the waveforms to be compared is used as a reference point. After that, input AP is rescaled: , where and coefficients are determined by the least-squares technique to minimize the deviation between input and output AP.”

Line 268: Maybe put the i's and NSR's on these concentrations, though I appreciate it's uglier when you also want to use the subscript to mark the pacing frequency. Not a major issue to me either way.

We have added the subscripts.

Line 286: I agree with the other reviewer's first round comments, I think that the principal components are a much nicer way to visualise this!

Thank you!

Line 302: The point is labelled here R, and another later A, but then R and A are not referred to again from what I saw. Labelling them like this just made it more confusing to me, (x0,y0) and (xc,yc) would be fine.

R and A were removed from the text.

Line 316: Figure 5 could probably be moved to the supplement, in my opinion. The point that large numbers of elite particles performs better is already made in Figure 4.

We have shifted the figure to the supplement.

Line 456: This is a nice explanation of why the method presented in the paper could be expected to work. I feel like it fits better where the method is presented, instead of in results, but this isn't major.

Thank you, we have shifted the text partially, but the text below also provides the proof that output variables are steady-state indeed, thus we believe that the manuscript is more comprehensible with the explanation in the results section.

Line 584: The section "Input data requirements" talks first about the amount of restitution information required for different parameters to be correctly identified, and then the requirements in terms of noise. I would actually just break this up into two separate sections, with more informative headings on each. "Input data requirements" doesn't immediately make me think of restitution/requirements on number of pacing frequencies. Also, it seems to me like other target values for the parameters could likely change how many different frequencies are required to identify those different parameters. So talking about this like a requirement on the input data has been identified feels a little wrong.

Line 604: On the other hand, while it might be true that different APs also showed a different sensitivity to noise, I'm far more comfortable with this section being expressed in terms of the "required signal-to-noise ratio". I also appreciate its inclusion, it was interesting and I think it's valuable.

We have divided the text in two subsections (“Input data requirements: restitution information.” and “Input data requirements: signal-to-noise ratio.”). Indeed changing the balance between ionic currents considerably (for example increasing IKs, while decreasing IKr) would affect the algorithm sensitivity to the amount of restitution information. We have acknowledged it in the former subsection, lines 587-590:

“While using considerably different target parameter values would affect the results shown in this section, it provides an estimate of the amount of restitution information required to determine identifiable parameters. Consequently, we have recorded the AP waveform at 4 PCLs in the experiments described below.”

---

## [Editor Report · Decision Letter 2]

31 Mar 2020

Genetic algorithm-based personalized models of human cardiac action potential.

PONE-D-19-25321R2

Dear Dr. Syunyaev,

We are pleased to inform you that your manuscript has been judged scientifically suitable for publication and will be formally accepted for publication once it complies with all outstanding technical requirements.

With kind regards,

B. Rodríguez

Academic Editor

PLOS ONE

Additional Editor Comments (optional):

Please revise grammar and typos throughout the manuscript. There are still minor language errors.
---

## [Editor Report · Acceptance letter]

21 Apr 2020

PONE-D-19-25321R2 

Genetic algorithm-based personalized models of human cardiac action potential. 

Dear Dr. Syunyaev:

I am pleased to inform you that your manuscript has been deemed suitable for publication in PLOS ONE. Congratulations! Your manuscript is now with our production department. 

With kind regards,

on behalf of

Prof B. Rodríguez 

Academic Editor

PLOS ONE